# Smoothed Online Learning for Prediction in Piecewise Affine Systems

**Adam Block**
Department of Mathematics
MIT
ablock@mit.edu

**Max Simchowitz**
CSAIL
MIT
msimchow@csail.mit.edu

**Russ Tedrake**
CSAIL
MIT

## Abstract

The problem of piecewise affine (PWA) regression and planning is of foundational importance to the study of online learning, control, and robotics, where it provides a theoretically and empirically tractable setting to study systems undergoing sharp changes in the dynamics. Unfortunately, due to the discontinuities that arise when crossing into different "pieces," learning in general sequential settings is impossible and practical algorithms are forced to resort to heuristic approaches. This paper builds on the recently developed *smoothed online learning* framework and provides the first algorithms for prediction and simulation in PWA systems whose regret is polynomial in all relevant problem parameters under a weak smoothness assumption; moreover, our algorithms are efficient in the number of calls to an optimization oracle. We further apply our results to the problems of one-step prediction and multi-step simulation regret in piecewise affine dynamical systems, where the learner is tasked with simulating trajectories and regret is measured in terms of the Wasserstein distance between simulated and true data. Along the way, we develop several technical tools of more general interest.

## 1 Introduction

A central problem in the fields of online learning, control, and robotics is how to efficiently plan through piecewise-affine (PWA) systems. Such systems are described by a finite set of disjoint regions in state-space, within which the dynamics are affine. In this paper, we consider the related problem of learning to make predictions in PWA systems when the dynamics are unknown. While recent years have seen considerable progress around our understanding of linear control systems, the vast majority of dynamics encountered in practical settings involve nonlinearities. Learning and predicting in nonlinear systems is typically far more challenging because, unlike their linear counterparts, the dynamics in a local region of a nonlinear system do not determine its global dynamics.

PWA systems allow for discontinuities across the separate regions ("pieces"), and are thus a simplified way of modeling rich phenomena that arise in numerous robotic manipulation and locomotion tasks [41, 64, 40], such as modeling dynamics involving contact. In addition, deep neural networks with ReLU activation [24] are PWA systems, providing further motiviation for their study. Already, there is a computational challenge simply in optimizing an objective over these systems, which is the subject of much of the previous literature. Here, we take a statistical perspective, assuming that we have access to effective heuristic algorithms for this optimization task, as is common in online learning [29, 9, 28]. Uniformly accurate learning of the dynamics across all pieces is typically impossible

37th Conference on Neural Information Processing Systems (NeurIPS 2023).

because some regions of space require onerous exploration to locate; thus, we instead consider an iterative prediction setting, where at each time $t$, the learner predicts the subsequent system state. From this, we extend to prediction of entire trajectories: over episodes $t = 1, \ldots, T$, a learner suggests a policy $\pi_t$, and the learner attempts to learn the dynamics so as to minimize prediction error along the trajectory $\pi_\tau$ induces. This is motivated by iterative learning control, where our notion of error is an upper bound on the discrepancy between the planner's estimate of policy performance and the ground truth.

Our iterative formulation of the learning problem is equivalent to the regret metric favored by the online learning community, and which has seen application to online/adaptive control and filtering problems in recent work. Critically, regret does not require uniform identification of system parameters, which is typically impossible. The key pathology is that policies can oscillate across the boundary of two regions, accruing significant prediction error due to sharp discontinuities of the dynamics between said regions, a problem well-known to the online learning community [36]. Moreover, this pathology is not merely a theoretical artifact: discontinuities and non-smoothness pose significant challenges to planning and modeling contact-rich dynamics such as those encountered in robotic manipulation [58, 59, 46].

Our solution takes inspiration from, and establishes a connection between, two rather different fields. Recent work in the robotics and control communities has relied on *randomized smoothing* to improve planning across discontinuous and non-smooth dynamics [58, 59, 35, 44]. Additionally, the online learning community has studied *smoothed online learning* [49, 27, 28, 9, 7, 6], which circumvents the threshold-effect pathologies described above. We show that, if the dynamics and the control inputs are subject to randomized smoothing, low regret becomes achievable. We note that the randomized smoothing approach is in some sense canonical in mitigating the aforementioned pathology of policies that oscillate across the boundaries of two regions; smoothing prohibits this pathology by ensuring that the system is generally far from these boundaries. More importantly, our proposed no-regret algorithm is *efficient* in terms of the number of calls to an optimization oracle, a popular notion of computational efficiency in the online learning community [29, 34, 9, 28]. In our setting, the optimization oracle required finds the *best-fit* PWA system to a batch of given data. Though this problem is intractable [5], there is a rich literature of popular heuristics [23, 45, 19]. Unlike those works, we examine the *statistical challenge* of generalization to novel data given such an oracle. We remark that in many practical cases, our smoothness assumption comes for free, in the sense that the Gaussian noise used to smooth gradients in Suh et al. [58] is already sufficient for our results to hold.

**Contributions.** A formal description of our setting is deferred to Section 2. Informally, we consider the problem of prediction in a PWA system over a horizon of $T$ steps, where the regions are determined by intersections of halfspaces; we obtain prediction in PWA dynamical systems as a special case. We aim to achieve sublinear-in-$T$ excess square-loss error (*regret*) of $O\left(T^{1-\alpha}\right)$ with respect to the optimal predictor which knows the system dynamics, where $\alpha > 0$ is constant and the prefactor on the regret is at most polynomial in all of the problem parameters. Our result is derived from a general guarantee for online piecewise affine regression, which subsumes the online PWA identification setting. We show that, when the dynamics and the control inputs are subject to randomized smoothing satisfying the $\sigma_{\mathrm{dir}}$-directional smoothness condition introduced by [7], our regret bound is polynomial in $\sigma_{\mathrm{dir}}$, dimension, and other natural problem parameters. While the exact dependence on horizon may be far from optimal, we emphasize that our algorithm is the first to achieve regret even polynomial in all parameters in the difficult setting of PWA systems. Moreover, our work provides the first regret bound with polynomial dependence in dimension for oracle-efficient smoothed online learning in the sense of Hazan and Koren [29] of a noncontinuous function class without a realizability assumption and allowing for process noise. As a further application, we adapt our algorithm to make predictions, at each time $t$, of the trajectory comprising the next $H$ steps in the evolution of the dynamical system. We bound the *simulation regret*, the cumulative error in Wasserstein distance between the distribution of our proposed trajectory and that of the actual trajectory. Assuming a Lyapunov condition, we demonstrate that our modified algorithm achieves simulation regret $O\left(\mathsf{poly}(H) \cdot T^{1-\alpha}\right)$, which allows for efficient simulation.

**Key challenges.** As noted above, vanishing regret in our setting is impossible without directional smoothness due to discontinuity in the regressors between different regions. The key challenge is to leverage smoothness to mitigate the effect of these discontinuities. Note that, if the learner were to observe the region in addition to the state, regression would be easy by decomposing the problem into

$K$ separate affine regression instances; the difficulty is in minimicking this approach *without explicit labels indicating which points are in which regions.* We remark that were we to care only about low regret without regard to efficiency, a simple discretization scheme of size exponential in dimension coupled with a standard learning from experts algorithm [12] can achieve low regret; realizing an oracle-efficient algorithm is significantly more challenging.

We adopt a natural, epoch-based approach by calling the optimization oracle between epochs, estimating the underlying system, and using this estimate to assign points to regions in the subsequent epoch. This introduces three new challenges, each of which requires substantial technical effort to overcome. First, we need to control the performance of our oracle on the regression task, which we term *parameter recovery* below. Second, we must enforce consistency between regions learned in different epochs, so that we can predict the correct regions for subsequent covariates; doing this necessitates an intricate analysis of how the estimated parameters evolve between epochs. Finally, we need to modify the output of our oracle to enforce low regret with respect to the classification problem of correctly predicting which mode we are in on as yet unseen data.

**Our techniques.**   Our algorithm proceeds in epochs, at the end of each of which we compute a best-fitting piecewise affine approximation to available data by calling the Empirical Risk Minimization (ERM) oracle. The analysis of this best-fit shows that we recover the parameters and decision boundaries of the associated regions ("pieces") frequently visited during that epoch. By the pigeonhole principle, this ultimately covers all regions visited a non-neglible number of times $t$ over the horizon $T$. Here, we leverage a careful analysis based on two modern martingale-least squares techniques: the self-normalized concentration bound [2] for establishing recovery in a norm induced by within-region empirical covariance matrices, and the block-martingale small-ball method [54] to lower-bound the smallest eigenvalues of empirical covariances with high probability. In addition, we introduce and bound a new complexity parameter, the *disagreement cover*, in order to make our statements uniform over the set of possible decision boundaries.

After the estimation phase, each epoch uses this estimate to make predictions on new data as it arrives. The key challenge is classifying the region to which each newly observed data point belongs. Without smoothing, this can be particularly treacherous, as points on or near to the boundary of regions can easily be misclassified and the correct predictions are discontinous. To leverage smoothing, we propose a reduction to online convex optimization on the hinge loss with lazy updates and show, through a careful decomposition of the loss, that our classification error can be controlled in terms of the assumed directional smoothness. Combining this bound on the classification error with the parameter recovery described in the preceding paragraph yields our sublinear regret guarantee.

We instantiate one-step prediction as a special case of the more general online linear regression problem, showing that the approach described above immediately applies to the first problem under consideration. Finally, in order to simulate trajectories, we use the learned parameters in the epoch to predict the next $H$ steps of the evolution. We then use the fact that our main algorithm recovers the parameters defining the PWA system to show that simulation regret is bounded, which solves the second main problem and allows for simulation in such systems.

## 1.1   Related Work

**System Identification.**   System-identification refers to estimating the governing equations of a dynamical system. The special case of linear dynamical systems is classical [37, 38, 16], and more recent work has provided finite-sample statistical analyses in many regimes of interest [54, 15, 43, 61, 63]. Estimation for non-linear systems has proven challenging, and has been studied most in the setting where the dynamics involve a known and smooth nonlinearity [39, 51, 20], or where only the observation model is nonlinear [14, 42]. [52] study *Markov jump systems*, where the system dynamics alternate between one of a finite number of linear systems ("modes"), with switches between modes governed by a Markov chain. PWA systems are thus an attractive intermediate step between the simple linear setting and the intractable general case. To our knowledge, ours is the first work to tackle piecewise affine dynamics in full generality, where the system mode is determined by the system state.

Despite its worst-case computational intractability [5], there is a rich literature of heuristic algorithms for computing the best-fit PWA system to data; see the surveys [23] and [45], which compare various approaches, notably the clustering method due to [19]. These references focus strictly on

the *algorithmic* facets of computing a best-fit given an existing data set. We instead abstract the computation of a best-fit into an ERM oracle in order to focus on the *statistical* considerations.

**Online Learning and Smoothing.** The study of online learning is now classical, with a small sample of early works including Littlestone [36], Freund and Schapire [22, 21]. More recently, a general theory of online learning matching that which we have in the statistical learning setting was built up in Rakhlin et al. [49], Rakhlin and Sridharan [47], Rakhlin et al. [50], Block et al. [8], with Rakhlin and Sridharan [48] characterizing the minimax regret of online regression. Due to the robustness and generality of this adversarial model, many problems that are trivial to learn in the statistical setting become impossible in the adversarial regime. Thus, some work [49, 27, 28, 9, 6] has sought to constrain the adversary to be *smooth*, in the sense that the adversary is unable to concentrate its actions on worst-case points. In addition, Block and Simchowitz [7], which introduces the notion of directional smoothness to control the regret in a realizable, online classification problem; realizability, here, means that there is no process noise in the sense that there exists some piecewise affine function that perfectly predicts all targets given the covariates. We also note that the algorithm in that paper is highly tailored to this realizable setting and thus is not at all robust to the noise considered in the present paper. Of particular note is the concurrent work of Block et al. [10], which solves a similar problem with a stronger oracle. Note that in that paper, the oracle assumption is strong enough to break the lower bound of Hazan and Koren [29], while in this work we use the much weaker oracle assumption found in Hazan and Koren [29], Haghtalab et al. [28], Block et al. [9]. This distinction is important because even in the standard (fully adversarial) online learning setting, there is a computational separation between these two oracle models in general, as per Agarwal et al. [4]; thus the fact that this separation does not apply to PWA systems given smoothness may be of independent interest.

**Online Learning for Control.** In addition to the recent advances in finite-sample system identification, a vast body of work has studied control tasks from the perspective of regret [1, 3, 56, 53, 13]. While we do not consider a control objective in this work, we share with these works the performance metric of regret. Our work is more similar in spirit to online prediction in control settings [31, 62], in that we also consider the task of next-step predictions under a system subject to control inputs.

**Smoothing in RL and Control.** One interpretation of the smooth online learning setting is that well-conditioned random noise is injected into the adversary's decisions. It is well known that such noise injection can be viewed as a form of regularization [17], and recent recent work in the robotics literature has shown that randomized smoothing improves the landscape in various robotic planning and control tasks [58, 59, 35, 44]. More broadly, randomization has been popular for computation of zeroth-order estimates of gradients, notably via the acclaimed policy-gradient computation [66].

## 2 Setting

In this section, we formally describe the problem setting of online PWA regression. We suppose there is a ground truth classifier $g_\star : \mathbb{R}^d \to [K]$ and matrices $\left\{ \boldsymbol{\Theta}_i^\star \mid i \in [K] \text{ and } \boldsymbol{\Theta}_i^\star \in \mathbb{R}^{m \times (d+1)} \right\}$ such that

$$\mathbf{y}_t = \boldsymbol{\Theta}_{i_t}^\star \cdot \bar{\mathbf{x}}_t + \mathbf{e}_t + \boldsymbol{\delta}_t, \quad i_t = g_\star(\bar{\mathbf{x}}_t), \quad \bar{\mathbf{x}}_t = [\mathbf{x}_t^\top | 1]^\top, \quad ||\boldsymbol{\delta}_t|| \leq \varepsilon_{\text{crp}}, \tag{2.1}$$

where, $\mathbf{x}_t \in \mathbb{R}^d$ are covariates, $\mathbf{y}_t \in \mathbb{R}^m$ are responses, $\mathbf{e}_t \in \mathbb{R}^m$ are zero-mean noise vectors in $\mathbb{R}^m$, $\boldsymbol{\delta}_t$ are (small) non-stochastic corruptions with norm at most $\varepsilon_{\text{crp}} \ll 1$, and $i_t \in [K]$ are the regression modes, which depend on the covariates $\mathbf{x}_t$. The extension to $\bar{\mathbf{x}}_t$ accomodates affine regressors. We suppose that the learner has access to pairs $(\mathbf{x}_t, \mathbf{y}_t)$ but, critically, does not know the regression modes $i_t$. We do however suppose that $g_\star$ is an *affine classifer*; that is $g^\star \in \mathcal{G} = \left\{ \mathbf{x} \mapsto \arg\max_{1 \leq i \leq K} \langle \mathbf{w}_i, \mathbf{x} \rangle + b_i \right\}$, where $\mathbf{w}_i \in S^{d-1}$ is the unit sphere and $b_i \in [-B, B]$. It can be shown that many natural physical systems are indeed modeled as PWA with affine boundaries [40], and the closed loop dynamics for model-predictive control (MPC) in these cases is also a PWA system [11]. We will assume throughout access to an empirical risk minimization oracle, ERMORACLE, which satisfies the following guarantee:

**Assumption 1** (ERMORACLE guarantee). *Given data* $(\bar{\mathbf{x}}_{1:s}, \mathbf{y}_{1:s})$, ERMORACLE *returns* $\left\{ \widehat{\mathbf{\Theta}}_i | i \in [K] \right\}, \widehat{g}$ *satisfying*

$$\sum_{t=1}^{s} \left\| \widehat{\mathbf{\Theta}}_{\widehat{g}(\mathbf{x}_t)} \bar{\mathbf{x}}_t - \mathbf{y}_t \right\|^2 - \inf_{g \in \mathcal{G}, \{\mathbf{\Theta}_i | i \in [K]\}} \sum_{t=1}^{s} \left\| \mathbf{\Theta}_{g(\mathbf{x}_t)} \bar{\mathbf{x}}_t - \mathbf{y}_t \right\|^2 \leq \varepsilon_{\mathrm{orac}}. \tag{2.2}$$

While such an oracle is not computationally efficient in the worst case [5], there are many popular heuristics used in practical situations [19, 23]. The learner aims to construct an algorithm, efficient with respect to the above oracle, that at each time step $t$ produces $\widehat{\mathbf{y}}_t$ such that

$$\mathrm{Reg}_T := \sum_{t=1}^{T} \|\widehat{\mathbf{y}}_t - \mathbf{\Theta}^\star_{g_\star(\bar{\mathbf{x}}_t)} \bar{\mathbf{x}}_t\|^2 = o(T). \tag{2.3}$$

This notion of regret is standard in the online learning community and, as we shall see, immediately leads to bounds on prediction error.

**Directional Smoothness.** As previously mentioned, without further restriction, sublinear regret is infeasible, as formalized in the following proposition:

**Proposition 1.** *In the above setting, there exists an adversary with $m = d = 1$, K=2, that chooses $\mathbf{\Theta}^\star$ and $g_\star$, as well as $\mathbf{x}_1, \ldots, \mathbf{x}_T$ such that any learner experiences $\mathbb{E}\left[\mathrm{Reg}_T\right] \geq \frac{T}{2}$.*

Proposition 1 follows from a construction of a system where there is a discontinuity across a linear boundary, across which the states $\mathbf{x}_t$ oscillate. The bound is then a consequence of the classical fact that online classification with linear thresholds suffers linear regret; see Appendix B.1 for a formal proof, included for the sake of completeness. Crucially, a discontinuity in the dynamics necessitates an $\Omega(1)$ contribution to regret each time the decision boundary is incorrectly determined. To avoid this pathology, we suppose that the contexts are smooth; however, because standard smoothness [26] leads to regret that is exponential in dimension [7], we instead consider a related condition, "directional smoothness."[1]

**Definition 2** (Definition 52 from Block and Simchowitz [7]). *Given a scalar $\sigma_{\mathrm{dir}} > 0$, a random vector $\mathbf{x}$ is called $\sigma_{\mathrm{dir}}$-directionally smooth if for all $\mathbf{w} \in S^{d-1} := \{w \in \mathbb{R}^d : \|w\| = 1\}$, $c \in \mathbb{R}$, and $\delta > 0$, it holds that $\mathbb{P}\left(|\langle \mathbf{w}, \mathbf{x} \rangle - c| \leq \delta\right) \leq \delta/\sigma_{\mathrm{dir}}$.*

As an example, for any vector $\mathbf{z}_t$, if $\mathbf{w}_t \sim \mathcal{N}(0, \sigma_{\mathrm{dir}}^2 \mathbf{I})$, then $\mathbf{z}_t + \mathbf{w}_t$ is $\frac{\sigma_{\mathrm{dir}}}{\sqrt{2\pi}}$-directionally smooth; see Appendix C.1 for a proof and examples of directional smoothness for other noise distributions. Crucially, directional smoothness is *dimension-independent*, in contradistinction to standard smoothness, where we would only have $\Theta\left(\sigma_{\mathrm{dir}}^d\right)$-smoothness (in the standard sense) in the previous example. We remark that directional smoothness is a weak condition that holds in many natural settings. Indeed, whenever noise is injected into the dynamics, directional smoothness will come for free. Such noise injection can occur when there is uncertainty in position, as is common in robotic applications; in this case, we can interpret this uncertainty itself as stochastic, which was the original motivation for smoothed analysis of algorithms [57]. Further, note that while directional smoothness ensures the system spends little time close to a boundary, due to the discreteness in time, there is no restriction on the number of mode switches, which can be $\Theta(T)$ in general.

We defer further discussion of the assumption to Appendix C.1. We require our smoothness condition to hold conditionally:

**Assumption 2.** *Let $\mathscr{F}_t$ denote the filtration generated by $\mathbf{x}_1, \ldots, \mathbf{x}_t$. For all times $t$, $\mathbf{x}_t$ conditional on $\mathscr{F}_{t-1}$ is $\sigma_{\mathrm{dir}}$-directionally smooth.*

**Further Assumptions.** Next, we make two standard assumption for sequential regression, one controlling the tail of the noise and the other the magnitude of the parameters.

**Assumption 3.** *Let $\mathscr{F}_t^y$ denote the filtration generated by $\mathscr{F}_t$ and $\mathbf{e}_1, \ldots, \mathbf{e}_{t-1}$ (equivalently, by $\mathscr{F}_t$ and $\mathbf{y}_t, \ldots, \mathbf{y}_{t-1}$). For all $t$, it holds that $\mathbf{e}_t$ is $\nu^2$-sub-Gaussian conditioned on $\mathscr{F}_t^y$; in particular, $\mathbf{e}_t$ is conditionally zero mean.*

**Assumption 4.** *We suppose that for all $1 \leq t \leq T$, $\|\bar{\mathbf{x}}_t\| \leq B$ and, furthermore, $\|\mathbf{\Theta}_i^\star\|_{\mathrm{F}} \leq R$ for all $i$. We will further assume that ERMORACLE always returns $\widehat{\mathbf{\Theta}}_i$ such that $\|\widehat{\mathbf{\Theta}}_i\|_{\mathrm{F}} \leq R$.*

---

[1]For a comparison between directional and standard smoothness, consult [7].

**Algorithm 1** Main Algorithm
---
1: **Initialize** Epoch length $E$, Classifiers $\mathbf{w}_{1:K}^0 = (\mathbf{0}, \ldots, \mathbf{0})$, margin parameter $\gamma > 0$, learning rate $\eta > 0$
2: **for** $\tau = 1, 2, \ldots, T/E$ **do** (`% iterate over epochs`)
3:    $(\widehat{\boldsymbol{\Theta}}_{\tau,i})_{1 \leq i \leq K}, \widehat{g}_\tau \leftarrow \text{ERMORACLE}(\bar{\mathbf{x}}_{1:\tau E}, \mathbf{y}_{1:\tau E})$
4:    $\widehat{g}_\tau \leftarrow \text{Reorder}\left(\widehat{g}_\tau, (\widehat{\boldsymbol{\Theta}}_{\tau,i})_{1 \leq i \leq K}, (\widehat{\boldsymbol{\Theta}}_{\tau-1,i})_{1 \leq i \leq K}\right)$
5:    $\mathbf{w}_{1:K}^{\tau E} \leftarrow \text{OGD}(\mathbf{w}_{1:K}^{(\tau-1)E}, \bar{\mathbf{x}}_{(\tau-1)E:\tau E}; \widehat{g}_\tau, \gamma, \eta)$
6:    Let $\tilde{g}_\tau(\bar{\mathbf{x}}) = \arg\max_{1 \leq i \leq K} \langle \mathbf{w}_i^{\tau E}, \bar{\mathbf{x}} \rangle$ denote classifier induced by $\mathbf{w}_{1:K}^{\tau E}$.
7:    **for** $t = \tau E, \ldots, (\tau+1)E - 1$ **do**
8:       **Receive** $\bar{\mathbf{x}}_t$, and **predict** $\hat{\mathbf{y}}_t = \widehat{\boldsymbol{\Theta}}_{\tau, \tilde{g}_\tau(\bar{\mathbf{x}}_t)} \cdot \bar{\mathbf{x}}_t$
9:       **Receive** $\mathbf{y}_t$
---

Finally, we assume that the true affine parameters are well-separated:

**Assumption 5.** *There is some $\Delta_{\text{sep}} > 0$ such that for all $1 \leq i < j \leq K$, it holds that $\|\boldsymbol{\Theta}_i^\star - \boldsymbol{\Theta}_j^\star\|_{\text{F}} \geq \Delta_{\text{sep}}$.*

Note that Assumption 5 is, in a sense, generic, as explained in Remark 4 We defer the formal setting of one-step prediction and simulation regret to Appendix G.1 and section 3.1.

## 3   Algorithm and Guarantees

We propose Algorithm 1, an oracle-efficient protocol with provably sublinear regret and polynomial dependence on all problem parameters. Algorithm 1 runs in epochs: at the beginning of epoch $\tau$, the learner calls ERMORACLE on the past data to produce a linear classifier $\widehat{g}_\tau$ and estimates of the regression matrices in each mode, $\widehat{\boldsymbol{\Theta}}_{\tau,i}$. A major challenge is identifying the same affine regions between epochs. To this end, the learner permutes the labels of $\widehat{g}_\tau$ to preserve consistency in labels across epochs, as explained informally in Section 4.2 and in greater detail in Appendix E.

The learner then runs online gradient descent (Algorithm 2) on the hinge loss to produce a modified classifier $\tilde{g}_\tau$; finally, throughout the epoch, the learner uses $\widehat{\boldsymbol{\Theta}}_{\tau,i}$ and $\tilde{g}_\tau$ to predict $\hat{\mathbf{y}}_t$ before repeating the process in the next epoch. The reason for using a secondary algorithm to transform $\widehat{g}_\tau$ into $\tilde{g}_\tau$ is to enforce stability of the predictor across epochs, which is necessary to bound the regret. Again, we suppose that Assumption 1 holds throughout.

**Theorem 3** (Regret Bound). *Suppose we are in the setting of online PWA regression (2.1) and Assumptions 2-5 hold. Then, if the parameters $E, \gamma, \eta$ are set as in (F.6), it holds that with probability at least $1 - \delta$, $\text{Reg}_T \leq \text{poly}(\overline{\text{par}}, \log(1/\delta)) \cdot T^{35/36} + \varepsilon_{\text{orac}} \cdot T + K^2 \cdot \sum_{t=1}^T \|\boldsymbol{\delta}_t\|$, for $\overline{\text{par}} := \max\{d, m, \sigma_{\text{dir}}^{-1}, \Delta_{\text{sep}}^{-1}, B, R, K, \nu\}$, where the polynomial dependence is given in (F.7).*

**Remark 1.** *While Theorem 3 requires that we are correctly setting the parameters of the algorithm, aggregation algorithms allow us to tune these parameters in an online fashion (see, e.g. [30]).*

**Remark 2.** *The misspecification errors $\boldsymbol{\delta}_t$ can be indicators corresponding to the $\bar{\mathbf{x}}_t$ being in small, rarely visited regions; thus, we could run an underspecified ERMORACLE, with $K' \ll K$ modes and recover a regret bound depending only on frequently visited modes.*

When $\varepsilon_{\text{crp}} = \varepsilon_{\text{orac}} = 0$, we obtain exactly the polynomial, no-regret rates promised in the introduction. Along the way to proving our regret bound, we establish that ERMORACLE correctly recovers the linear parameters within frequently visited modes.

**Theorem 4** (Parameter Recovery). *Suppose that Assumptions 2-4 hold and let $\left\{\widehat{\boldsymbol{\Theta}}_i | i \in [K]\right\}, \widehat{g}$ denote the output of $\text{ERMORACLE}(\bar{\mathbf{x}}_{1:T}, \mathbf{y}_{1:T})$ and $I_{ij}(\widehat{g})$ denote the set of times $t$ such that $\widehat{g}(\bar{\mathbf{x}}_t) = i$ and $g_\star(\bar{\mathbf{x}}_t) = j$. Then with probability at least $1 - \delta$, for all $1 \leq i, j \leq K$ such that $|I_{ij}(\widehat{g})| \geq \text{poly}\left(\overline{\text{par}}, \log\left(\frac{1}{\delta}\right)\right) \cdot T^{1-\Omega(1)}$ it holds that $\|\widehat{\boldsymbol{\Theta}}_i - \boldsymbol{\Theta}_j^\star\|_{\text{F}}^2$ is bounded by $\frac{\sqrt{T}}{|I_{ij}(\widehat{g})|} \cdot \text{poly}\left(\overline{\text{par}}, \log\left(\frac{1}{\delta}\right)\right)$.*

Above we have given the result with the assumption that $\varepsilon_{\text{crp}} = \varepsilon_{\text{orac}} = 0$ for the sake of presentation; in the formal statement, the general case is stated. Theorem 4 implies that directional smoothness

is sufficient to guarantee parameter recovery at a parametric rate, in marked distinction to most adversarial learning regimes. Before providing more detail on the algorithm and its analysis in Section 4, we discuss an application to $H$-step prediction. An application to one-step prediction with learner-provided controls can be found in Appendix G.1

### 3.1 Guarantees for Simulation Regret

We now describe an application of our results to $H$-step prediction; more formal description is detailed in Appendix G.3. The learning process occurs in *episodes* $t = 1, 2, \ldots, T$, consisting of steps $h = 1, 2, \ldots, H$. In each episode, an external planner provides the learner a policy $\pi_t$. For simplicity, we assume that $\pi_t = (\mathbf{u}_{t,1}, \ldots, \mathbf{u}_{t,H})$ is *open loop* (state-independent) and stochastic, minimicking situations such as model predictive control; extensions are mentioned in Appendix G.6. With each episode, the true dynamics are given by

$$\mathbf{z}_{t,h+1} = \mathbf{A}^\star_{i_{t,h}} \mathbf{z}_t + \mathbf{B}^\star_{i_{t,h}} \mathbf{u}_{t,h} + \mathbf{m}^\star_{i_{t,h}} + \mathbf{e}_{t,h}, \quad i_{t,h} = g^\star(\mathbf{z}_{t,h}, \mathbf{u}_{t,h}). \tag{3.1}$$

We assume that the process noise $\mathbf{e}_{t,h}$ are independently drawn from a known distribution $\mathcal{D}$, and that $\mathbf{z}_{t,1}$ is sampled from an arbitrary $\sigma_{\mathrm{dir}}$-smooth distribution. At the start of episode $t$, the learner then produces estimates $\hat{\mathbf{A}}_t, \hat{\mathbf{B}}_t, \hat{g}_t$ and simulates the plan $\pi_t$ using the plug in estimate of the dynamics, i.e., $\hat{\mathbf{z}}_{t;h+1} = \hat{\mathbf{A}}_{t,\hat{i}_{t,h}} \hat{\mathbf{z}}_{t,h} + \hat{\mathbf{B}}_{t,\hat{i}_{t,h}} \mathbf{u}_{t,h} + \hat{\mathbf{m}}_{t,\hat{i}_{t,h}} \hat{\mathbf{e}}_{t,h}, \hat{i}_{t,h} = \hat{g}_t(\hat{\mathbf{z}}_{t,h}, \hat{\mathbf{u}}_{t,h})$, where $\hat{\mathbf{e}}_{t,h} \overset{\mathrm{i.i.d.}}{\sim} \mathcal{D}$, and $\hat{\mathbf{u}}_{t,h}$ are an i.i.d. draw from the stochastic, open-loop policy $\pi_t$. Because the simulated and true processes use a different noise sequence, we consider the following notion of error, which measures the distance betwen the two *distributions* induced by the dynamics, as opposed to the regret, which measure the distance between the realizations thereof. We choose the Wasserstein metric $\mathcal{W}_2$, as it upper bounds the difference in expected value of any Lipschitz reward function between the true and simulated trajectories (see Appendix G.3 for a formal definition and more explanation).

**Definition 5** (Simulation Regret). *Let $\mathcal{W}_2(\cdot, \cdot)$ denote the $L_2$-Wasserstein distance, define the con-catenations $\mathbf{x}_{t,h} = (\mathbf{z}_{t,h}, \mathbf{u}_{t,h})$ and $\hat{\mathbf{x}}_{t,h} = (\hat{\mathbf{z}}_{t,h}, \hat{\mathbf{u}}_{t,h})$, and let $\bar{\mathscr{F}}_t$ be the filtration generated by $\{\mathbf{x}_{s,h}\}_{1 \le s \le t, 1 \le h \le H}$. We define the $T$-episode, $H$-step simulation regret as $\mathrm{SimReg}_{T,H} := \sum_{t=1}^{T} \mathcal{W}_2 \left( \mathbf{x}_{t,1:H}, \hat{\mathbf{x}}_{t,1:H} \mid \bar{\mathscr{F}}_{t-1} \right)^2$.*

Our goal is to achieve $\mathrm{SimReg}_{T,H} \lesssim \mathrm{poly}(H) \cdot T^{1-\Omega(1)}$, but polynomial-in-$H$ dependence may be challenging for arbitrary open-loop policies and unstable dynamics of the pieces. Thus, in the interest of simplicity, we decouple the problems of linear stability from the challenge of error compounding due to discontinuity of the dynamics by adopting the following strong assumption.

**Assumption 6.** *There exists a* known, *positive definite Lyapunov matrix* $\mathbf{P} \in \mathbb{R}^{d_z \times d_z}$ *that satisfies* $(\mathbf{A}_i^\star)^\top \mathbf{P}(\mathbf{A}^\star)_i \preceq \mathbf{P}$ *for all modes* $i \in [K]$.

Using a minor modification of Algorithm 1, detailed in Algorithm 4, we show that we can get vanishing simulation regret, summarized in the following result:

**Theorem 6.** *Suppose that we are in the setting of* (3.1) *and that Assumptions 2-5 and 6 hold. If we run a variant of Algorithm 1 (see Algorithm 4 in Appendix G.4), then with probability at least $1 - \delta$, it holds that* $\mathrm{SimReg}_{T,H} \le \mathsf{poly}\left(\overline{\mathrm{par}}, H\right) \cdot T^{35/36}$.

The proof of Theorem 6 with the exact polynomial dependence on the parameters can be found in Appendix G.3 and rests on a lemma showing that under Assumption 6, simulation regret at a given time $t$ can be bounded as $\mathrm{poly}(H)$ multiplied by the maximum one-step prediction error for times $t, t+1, \ldots, t+H$, which is bounded in Theorem 3. We provide an exact recovery guarantee in the case $H = 1$ and without requiring Assumption 6 in Appendix G.1.

## 4 Analysis

In this section, we present a sketch of the proof of Theorem 15. There are two primary sources of regret: that which is due to poorly estimating the linear parameters within a mode and that which is due to misclassification of the mode. We analyze each source separately, beginning with the parameter recovery result of Theorem 4.

---
**Algorithm 2** Online Gradient Descent (Single Epoch)
---
1: **Initialize** Data $\bar{\mathbf{x}}_{1:E}$, clasifiers $\mathbf{w}_{1:K}^0$, margin parameter $\gamma > 0$, learning rate $\eta > 0$, classifier $\widehat{g} : \mathcal{X} \to [K]$
2: **for** $s = 1, 2, \dots, E$ **do**    % $\widetilde{\ell}$ defined in (4.1)
3:      **Receive** $\bar{\mathbf{x}}_i$ and **update** $\mathbf{w}_{1:K}^s \leftarrow \Pi_{(\mathcal{B}^{d-1})^{\times K}} \left( \mathbf{w}_{1:K}^{s-1} - \eta \nabla \widetilde{\ell}_{\gamma, i, \widehat{g}}(\mathbf{w}^{s-1}) \right)$
4: **Return** $\mathbf{w}_{1:K}^m$
---

## 4.1 Proof Sketch of Theorem 4

We break the proof into three parts: first, we show that the regret of $\widehat{g}$ and the $\widehat{\Theta}_i$, when restricted to times $t \in I_{ij}(\widehat{g})$, is small; second, we relate the prediction error $\mathcal{Q}_{ij}(\widehat{g}) = \sum_{t \in I_{ij}(\widehat{g})} \|(\widehat{\Theta}_i - \Theta_j^\star)\bar{\mathbf{x}}_t\|^2$. to the regret on $I_{ij}(\widehat{g})$; third, demonstrate that $\Sigma_{ij}(\widehat{g}) = \sum_{t \in I_{ij}(\widehat{g})} \bar{\mathbf{x}}_t \bar{\mathbf{x}}_t^T$ has minimal eigenvalue bounded below by some constant with high probability. If all three of these claims hold, then, noting that $\|\widehat{\Theta}_i - \Theta_j^\star\|_{\mathrm{F}}^2 \leq \|\Sigma_{ij}(\widehat{g})^{-1}\|_{\mathrm{op}} \cdot \sum_{t \in I_{ij}(\widehat{g})} \|(\widehat{\Theta}_i - \Theta_j^\star)\bar{\mathbf{x}}_t\|^2$,

we conclude. Each of these claims requires significant technical effort in its own right. We introduce *disagreement covers*, a stronger analogue of $\varepsilon$-nets adapted to our problem and allowing us to turn statements proven for a single $g \in \mathcal{G}$ into ones uniform over $\mathcal{G}$. The first step is to bound the size of disagreement covers for the class $\mathcal{G}$ of interest; we then prove each of the three claims for some fixed $g \in \mathcal{G}$ and lift the result to apply to the data-dependent $\widehat{g}$. This is done in Appendix D.1.

We now turn to our three claims. For the first, proved in Lemma D.15, we introduce the *excess risk* $\mathcal{R}_{ij}(g, \Theta) = \sum_{t \in I_{ij}(g)} \|\Theta_i \bar{\mathbf{x}}_t - \mathbf{y}_t\|^2 - \|\mathbf{e}_t + \boldsymbol{\delta}_t\|^2$, for each pair $(i, j)$, and note that the cumulative empirical excess risk $\sum_{i,j} \mathcal{R}_{ij}(\widehat{g}, \widehat{\Theta})$ of predicting using $\widehat{g}$ and $\widehat{\Theta}$ returned by the ERM oracle is bounded by $\varepsilon_{\mathrm{orac}}$ because $\mathcal{R}_{ij}(g_\star, \Theta_\star) = 0$ due to Equation (2.1). Thus, showing that $\mathcal{R}_{ij}(\widehat{g}, \widehat{\Theta})$ is small can be done by showing that for no $i', j'$ is $\mathcal{R}_{i'j'}(\widehat{g}, \widehat{\Theta}) \ll 0$. This can be accomplished by a concentration argument for a single $g, \Theta_0$, coupled with a covering argument using our notion of disagreement covers to boost the result to one uniform in $\mathcal{G}$.

For the second claim, i.e., that $\mathcal{Q}_{ij}(\widehat{g})$ is controlled by $\mathcal{R}_{ij}(\widehat{g}, \widehat{\Theta})$, we provide a proof in Lemma D.14. For a fixed $g \in \mathcal{G}$, we decompose $\mathcal{R}_{ij}(g, \widehat{\Theta})$ into $\mathcal{Q}_{ij}(g)$ and an error term. We then use a generalization of the self-normalized martingale inequality from Abbasi-Yadkori et al. [2] to control the error in terms of $\mathcal{Q}_{ij}(g)$, providing a self-bounded inequality. Finally, we rearrange this inequality and apply a union bound on a disagreement cover to boost the result to one uniform in $\mathcal{G}$ and $\Theta$.

To prove the third claim, that $\Sigma_{ij}(\widehat{g})$ has singular values uniformly bounded from below, we split our argument into two parts. First, in Appendix D.2, we assume the following sliced-small ball condition on the data, i.e., for some $\zeta, \rho > 0$, it holds that $\mathbb{P}(\langle \bar{\mathbf{x}}_t, \mathbf{w} \rangle^2 \geq c \cdot \zeta^2 | \mathscr{F}_t, t \in I_{ij}(\widehat{g})) \geq c \cdot \rho$. Given the above condition, we establish a high probability lower bound on $\Sigma_{ij}(g)$ for fixed $g$ using a self-normalized martingale inequality, using analysis that may be of independent interest. We then again apply a union bound on a disagreement cover to lift this result to be uniform in $\mathcal{G}$. Finally, in Appendix D.3, we provide bounds on $\zeta, \rho$ using directional smoothness and Markov's inequality.

## 4.2 Mode Prediction

In this section, we address the second source of error, mode misclassification. The primary challenge is that directional smoothness, as opposed to independent data, is insufficiently strong to guarantee that $\widehat{g}_\tau$ generalizes well to unseen data in epoch $\tau + 1$. We take inspiration from online learning and stabilize our algorithm across epochs by modifying the classifier $\widehat{g}_\tau$. There are two challenges in ensuring good performance of our online classifier. First, the dynamics described in (2.1) are only identifiable up to a permutation of the labels. Thus, in order to enforce stability across epochs, we need to enforce consistency of labelling accross epochs. This task is made more difficult by the fact that different modes may be combined or split up across epochs due to the black box nature of ERMORACLE. We solve this by appealing to a subroutine, Reorder, described in Algorithm 3 in Appendix E, which combines modes that are sufficiently large and have sufficiently similar parameters and then relabels the modes so that similar nominal clusters persist accross epochs.

The second challenge is enforcing stability of our online classifier. We do this by introducing a surrogate loss $\widetilde{\ell}_{\gamma,t,\widehat{g}}$, the multi-class hinge loss with parameter $\gamma > 0$[2]. Formally, let $\mathbf{w}_{1:K} = (\mathbf{w}_1, \ldots, \mathbf{w}_K)$ for $\mathbf{w}_i \in \mathcal{B}^{d-1}$, and for $g \in \mathcal{G}$, define the $(1/\gamma)$-Lipschitz and convex surrogate loss:

$$\widetilde{\ell}_{\gamma,t,g}(\mathbf{w}_{1:K}) = \max\left(0, \max_{j \neq g(\bar{\mathbf{x}}_t)}\left(1 - \frac{\langle \mathbf{w}_{g(\bar{\mathbf{x}}_t)} - \mathbf{w}_j, \bar{\mathbf{x}}_t\rangle}{\gamma}\right)\right). \tag{4.1}$$

In OGD (see Algorithm 2), we modify the output of ERMORACLE to construct a classifier $\widetilde{g}_\tau$ by running lazy online gradient descent on this surrogate loss, using the reordered labels given by the output of ERMORACLE evaluated on the previous epoch. By invoking OGD, we show that our mode classifier is sufficiently stable to ensure low regret, as stated informally in the following result.

**Theorem 7** (Mode Prediction, Informal Statement of Theorem 13). *Suppose that Assumptions 2-5 hold and let $\mathcal{E}_t^{\mathrm{mode}}$ be the event that the learner, invoking Algorithm 1 with correctly tuned parameters, misclassifies the mode of $\bar{\mathbf{x}}_t$. Then, with probability at least $1 - \delta$, it holds that $\sum_{t=1}^T \mathbb{I}\{\mathcal{E}_t^{\mathrm{mode}}\} \leq \mathrm{poly}\left(\overline{\mathrm{par}}, \log\left(\frac{1}{\delta}\right)\right) \cdot T^{1-\Omega(1)}$.*

*Proof Sketch.* We begin by addressing the lack of identifiability of the modes. For the purposes of analysis, for each epoch $\tau$, we introduce the function $\pi_\tau : [K] \to [K]$ such that $\pi_\tau(i) = \mathrm{argmin}_{1 \leq j \leq K} \|\widehat{\boldsymbol{\Theta}}_{\tau,i} - \boldsymbol{\Theta}_j^\star\|_{\mathrm{F}}$. In words, $\pi_\tau(i)$ is the mode $j$ according to the ground truth whose parameters are closest to those estimated by ERMORACLE. We let $\mathcal{E}_t^{\mathrm{mode}}$ denote the event of misclassifying the mode, i.e. the event that $\pi_\tau(\widetilde{g}_\tau(\bar{\mathbf{x}}_t)) \neq g_\star(\bar{\mathbf{x}}_t)$. We can then decompose the misclassification as $\mathcal{E}_t^{\mathrm{mode}} \subset \{\pi_{\tau+1}(\widetilde{g}_\tau(\bar{\mathbf{x}}_t)) \neq g_\star(\bar{\mathbf{x}}_t)\} \cup \{\pi_\tau(\widetilde{g}_\tau(\bar{\mathbf{x}}_t)) \neq \pi_{\tau+1}(\widetilde{g}_\tau(\bar{\mathbf{x}}_t))\}$. We call the first event look-ahead classification error and the second permutation disagreement error. To bound the first of these, we further decompose the look-ahead classification error event: $\{\pi_{\tau+1}(\widetilde{g}_\tau(\bar{\mathbf{x}}_t)) \neq g_\star(\bar{\mathbf{x}}_t)\} \subset \{\pi_{\tau+1}(\widehat{g}_{\tau+1}(\bar{\mathbf{x}}_t)) \neq g_\star(\bar{\mathbf{x}}_t)\} \cup \{\widetilde{g}_\tau(\bar{\mathbf{x}}_t) \neq \widehat{g}_{\tau+1}(\bar{\mathbf{x}}_t)\}$. Controlling the first of these events is proved as a corollary of the bound on the permutation disagreement error and we defer its discussion for the sake of simplicity. To bound the probability of the second event, upper bound the indicator loss by $\widetilde{\ell}_{\gamma,t,\widehat{g}_{\tau+1}}$ and apply standard online learning techniques to show that regret of OGD on the $(1/\gamma)$-Lipschitz and convex surrogate loss is small. We then use smoothness to show that the optimal comparator with respect to the surrogate loss does not experience large regret with respect to the indicator loss, which in turn has a comparator experiencing zero loss due to the well-specified nature of $g_\star$. Thus, putting everything together, this provides bound on the look-ahead classification error, with full details presented in Appendix E.1.

Bounding the permutation disagreement error is what necessitates the gap assumption. We show that if there are sufficiently many data points that $\widehat{g}_\tau$ assigns to a given mode, with the threshold defined in terms of $\Delta_{\mathrm{sep}}$ and other parameters of the problem, then with high probability the cluster becomes stable in the sense that $\widehat{\boldsymbol{\Theta}}_{\tau,i} \approx \widehat{\boldsymbol{\Theta}}_{\tau+1,i}$. This result is proved using the fact that if $|I_{i,j}(\widehat{g}_\tau)|$ is large enough, then the parameter recovery results from the previous section tell us that $\|\widehat{\boldsymbol{\Theta}}_i - \widehat{\boldsymbol{\Theta}}_j\|_{\mathrm{F}}$ is small. Using the triangle inequality and the fact that $\|\boldsymbol{\Theta}_j^\star - \boldsymbol{\Theta}_{j'}^\star\|_{\mathrm{F}} > \Delta_{\mathrm{sep}}$ for $j' \neq j$ ensures that $j$ is unique in satisfying $\|\widehat{\boldsymbol{\Theta}}_{\tau,i} - \boldsymbol{\Theta}_j^\star\|_{\mathrm{F}} \ll 1$. A similar argument applies to epoch $\tau + 1$, and thus we can identify $\widehat{\boldsymbol{\Theta}}_{\tau,i}$ with some $\widehat{\boldsymbol{\Theta}}_{\tau+1,i'}$ by these matrices being sufficiently close in Frobenius norm. Algorithm 3 takes advantage of exactly this property and permutes the labels across epochs in order to maintain consistency and control the permutation disagreement error; full details can be found in Appendix E.2. Combining this argument with the bound on the look-ahead classification error suffices to control the online classification component of the regret. ∎

## 5 Discussion

We have given an efficient online learning algorithm for prediction in piecewise affine (PWA) systems. Our results are the first foray into the study of such systems, and a number of outstanding questions remain: Does directional smoothness facilitate low regret for planning and control, in addition to for simulation and prediction? If the systems are piecewise-affine but forced to be continuous on their boundaries, is there an oracle efficient algorithm which suffers low regret without assuming directional

---

[2]We use the hinge loss because it is convex, Lipschitz, and its relationship to the indicator loss is particularly convenient under directional smoothness. Other convex, Lipschitz surrogates provide similar guarantees.

smoothness? Can learning in piecewise affine systems be obtained under *partial observation* of the system state? Can the dependence of regret on horizon be improved? We hope that our work initiates investigation of the above as the field continues to expand its understanding beyond the linear regime.

## Acknowledgments and Disclosure of Funding

AB acknowledges support from the National Science Foundation Graduate Research Fellowship under Grant No. 1122374 as well as support from ONR under grant N00014-20-1-2336 and DOE under grant DE-SC0022199. MS acknowledges support from Amazon.com Services LLC grant; PO# 2D-06310236. We also acknowledge Zak Mhammedi, Terry H.J. Suh and Tao Pang for their helpful comments.

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
