# Contents

# Part I

# General Tools

## A   Notation

| Problem Parameter | Definition |
|---|---|
| $T$ | total number of time steps |
| $K$ | number of modes |
| $d$ | dimension of $\mathbf{x}$ (general PWA) |
| $m$ | dimension of $y$ |
| $\sigma_{\mathrm{dir}}$ | directional smoothness constant (Assumption 2) |
| $\nu$ | subGaussian constant of noise (Assumption 3) |
| $B$ | magnitude bound on $\|\bar{\mathbf{x}}_t\|$ (Assumption 4) |
| $R$ | magnitude bound on $\|\Theta^\star\|_{\mathrm{F}}$ (Assumption 4) |
| $\Delta_{\mathrm{sep}}$ | separation parameter (optional for sharper rates, see Assumption 5) |

| Algorithm Parameters | Definition |
|---|---|
| $E$ | epoch length |
| $\tau$ | current epoch |
| $\eta$ | step size (Line 4 in Algorithm 2) |
| $\gamma$ | hinge loss parameter |
| $A$ | cluster size threshold (Line 11 in Algorithm 3) |
| $\varepsilon_{\mathrm{crp}}$ | distance from realizability (see (2.1)) |
| **Algorithm Objects** | **Definition** |
| $\widetilde{\ell}_{\gamma,t,\widehat{g}}g$ | hinge loss on estimated labels (see (4.1)) |
| $\pi_\tau$ | stabilizing permutation (see (E.2)) |
| $\gamma$ | hinge loss parameter |
| **Analysis Parameters** | **Definition** |
| $\delta$ | probability of error |
| $\xi$ | scale of disagreement cover discretization (Theorem 11) |
| $\Xi$ | minimum cluster size to ensure continuity (Condition 1) |

# B  Lower Bounds

## B.1  Proof of Proposition 1

We suppose that $d = m = 1$ and consider the unit interval. Thus, $\mathcal{G}$ is just the set of thresholds on the unit interval, i.e.,

$$\mathcal{G} = \{\mathbf{x} \mapsto \mathbb{I}\{\mathbf{x} > \theta\} \,|\, \theta \in [0,1]\}.$$

We suppose that

$$\boldsymbol{\Theta}_0^\star = [0 \quad | \quad 1] \qquad\qquad \boldsymbol{\Theta}_1^\star = [0 \quad | \quad 0],$$

i.e., $\mathbf{y}_t = g_\star(\mathbf{x}_t)$. We are thus in the setting of adversarially learning thresholds with an oblivious adversary. It is well known that this is unlearnable, but we sketch a proof here. With $T$ fixed, let the adversary sample $\varepsilon_1, \ldots, \varepsilon_T$ as independent Rademacher random variables and let

$$\mathbf{x}_t = \frac{1}{2} + \sum_{s=1}^{t-1} \varepsilon_s 2^{-s-1}$$

and let $\theta^\star = \mathbf{x}_{T+1}$. We observe that $\mathbf{y}_t = -\varepsilon_t$ for all $t$. To see this, note that if $\varepsilon_t = 1$, then $x_s > x_t$ for all $s > t$ and similarly if $\varepsilon_t = -1$ then $x_s < x_t$; as $\theta^\star = \mathbf{x}_{T+1}$, the claim is clear. Note that due to the independence of the $\varepsilon_t$, any $\hat{y}_t$ chosen by the learner is independent of $\varepsilon_t$ and thus is independent of $\mathbf{y}_t$. Thus the expected number of mistakes the learner makes, independent of the learner's strategy, is $\frac{T}{2}$, concluding the proof.

## B.2  A Lower Bound for Identification

Consider a setting where there are $K = 3$ modes with state and input dimension $d = 1$. For a parameters $\alpha, \beta > 0$, define the linear functions

$$g_1(x; \alpha, \beta) = 0$$
$$g_3(x; \alpha, \beta) = x - (1 + \alpha)$$
$$g_2(x; \alpha, \beta) = 2(x - (1 + \alpha)) - \beta$$

For our lexicographic convention in the definition of the $\arg\max$ operator[3], we have

$$\arg\max_i g_i(x; \alpha, \beta) = \begin{cases} 1 & x \le 1 + \alpha \\ 2 & x \ge 1 + \alpha + \beta \\ 3 & x \in (1 + \alpha, 1 + \alpha + \beta) \end{cases}$$

That is, $x \mapsto \arg\max_i g_i(x; \alpha, \beta)$ defines three modes, with the third mode a segment of length $\beta$ between $1 + \alpha$ and $1 + \alpha + \beta$. We consider simple 3-piece PWA systems whose regions are defined by the above linear functions.

**Definition 8.** *Let $\mathscr{I}(\alpha, \beta, m, \mathcal{D})$ denote the problem instance with $K = 3$ pieces where where the dynamics abide by*

$$\mathbf{x}_{t+1} = \mathbf{u}_t + \mathbf{m}_{i_t} + \mathbf{w}_t \quad \mathbf{m}_i = \begin{cases} 0 & i = 1,2 \\ m & i = 3 \end{cases}, \quad i_t = \arg\max_i g_i(\mathbf{x}_t; \alpha, \beta), \quad \mathbf{w}_t \sim \mathcal{D}.$$

,

The following proposition shows that, regardless of the noise distribution $\mathcal{D}$, learning the parameter $_3$ can be make arbitrarily hard. This is because as the $\beta$ parameter is made small, making locating region 3 (which is necesary to learn $_3$) arbitrarily challenging.

**Proposition 9.** *Fix any positive integer $N \in \mathbb{N}$ and any arbitrary distribution $\mathcal{D}$ over $\mathbb{R}$. Then, any algorithm which adaptively selects inputs $\mathbf{u}_1, \ldots, \mathbf{u}_T$ returns an estimate $\hat{\mathbf{m}}_T$ of $\mathbf{m}_3$ must suffer*

$$\sup_{\iota \in \{-1,1\}} \sup_{j \in [2N]} \mathbb{P}_{\mathscr{I}(j/N, 1/N, \iota m, \mathcal{D})}[|\hat{\mathbf{m}}_T - \mathbf{m}_3| \ge m] \ge \frac{1}{2}\left(1 - \frac{T}{N}\right).$$

*As $N$ is arbitrary, this makes a constant-accuracy estimate of $\mathbf{m}_3$ arbitrarily difficult.*

---

[3]alternatively, we can make this unambiguous by ommiting the points $\{1 + \tau, 1 + 2\tau\}$

**Remark 3.** *The lower bound of Proposition 9 can be circumvented in two cases. First, if the dynamics are forced to be Lipschitz continuous, then deviations of the dynamics in small regions can only lead to small differences in parameter values. Second, if there is an assumption which stipulates that all linear regions have "large volume", then there is an upper bound on how large $N$ can be in the above construction, obviating our lower bound. It turns out that there are practical circumstances under which PWA systems (a) are not Lipschitz, and (b) certain regions have vanishingly small volume [33]. Still, whether stronger guarantees are possible under such conditions is an interesting direction for future work.*

*Proof of Proposition 9.* Introduce the shorthand $\mathbb{P}_{j,\iota}[\cdot] := \mathbb{P}_{\mathscr{I}(j/N, 1/N, \iota m)}[\cdot]$. Consider the regions $\mathcal{R}_j := [1 + j/4N, 1 + (j+1)/4N]$, and define the event $\mathcal{E}_{j,t} := \{\mathbf{u}_j \notin \mathcal{R}_j\}$ and $\mathcal{E}_j := \bigcap_{t=1}^{T} \mathcal{E}_{j,t}$. As the learner has information about $\mathbf{m}_3$ on $\mathcal{E}_j$, then if if $\iota \overset{unif}{\sim} \{-1, 1\}$, it holds that

$$\mathbb{E}_{\iota \sim \{-1,1\}} \mathbb{P}_{j,\iota}[|\hat{\mathbf{m}}_T - \mathbf{m}_3| \geq m \mid \mathcal{E}_j] \geq \min_{\mu \in \mathbb{R}} \mathbb{P}_\iota[|\iota m - \mu| \geq m] = \frac{1}{2}$$

Thus,

$$\sup_{j,\iota} \mathbb{P}_{j,\iota}[|\hat{\mathbf{m}}_T - \mathbf{m}_3| \geq m] \geq \frac{1}{2} \sup_{j,\iota} \mathbb{P}_{j,\iota}[\mathcal{E}_j]$$

Introduce a new measure $\mathbb{P}_0$ where the dynamics are given by

$$\mathbf{x}_{t+1} = \mathbf{u}_t + \mathbf{w}_t, \quad \mathbf{w}_t \overset{\text{i.i.d.}}{\sim} \mathcal{D}.$$

We observe that on $\mathbb{P}_{j,\iota}[\mathcal{E}_j] = \mathbb{P}_0[\mathcal{E}_j]$, because on $\mathcal{E}_j$ the learner has never visited region $i = 3$ where $i_t \neq 0$. Hence,

$$\sup_{j,\iota} \mathbb{P}_{j,\iota}[|\hat{\mathbf{m}}_T - \mathbf{m}_3| \geq m] \geq \frac{1}{2} \sup_j \mathbb{P}_0[\mathcal{E}_j] = \frac{1}{2}\left(1 - \min_j \mathbb{P}_0[\mathcal{E}_j^c]\right) \geq \frac{1}{2}\left(1 - \min_j \sum_{t=1}^{T} \mathbb{P}_0[\mathcal{E}_{j,t}^c]\right).$$

Observe that $\mathcal{E}_{j,t}^c := \{\mathbf{u}_t \in \mathcal{R}_j\}$. As $\mathcal{R}_{2\ell} \cap \mathcal{R}_{2(\ell+1)}$ are disjoint for $\ell \in \mathbb{N}$, it holds that $\mathcal{E}_{2\ell,t}^c$ are disjoint events for $\ell \in \mathbb{N}$. Upper bound bounding minimum by average on any subset, we have

$$\min_j \mathbb{P}_0[\mathcal{E}_{j,t}^c] \leq \frac{1}{N} \sum_{\ell=1}^{N} \mathbb{P}_0[\mathcal{E}_{2\ell+1,t}^c] \overset{(i)}{=} \frac{1}{N}\mathbb{P}_0[\bigcup_{\ell=1}^{N} \mathcal{E}_{2\ell+1,t}^c] \leq 1/N,$$

where $(i)$ uses disjointness of the events $\mathcal{E}_{2\ell,t}$ as argued above. Thus, continuing from the second-to-last display, we have

$$\sup_{j,\iota} \mathbb{P}_{j,\iota}[|\hat{\mathbf{m}}_T - \mathbf{m}_3| \geq m] \geq \frac{1}{2}\left(1 - \frac{T}{N}\right).$$

∎

# C  Properties of Smoothness

## C.1  Directional Smoothness of Gaussians and Uniform Distributions

**Lemma C.1.** *Let $\mathbf{w}$ be distributed as a centred Gaussian with covariance $\sigma^2 \mathbf{I}$ in $\mathbb{R}^d$. Then for any $\mathbf{z} \in \mathbb{R}^d$, if $\mathbf{w}$ is independent of $\mathbf{z}$, it holds that $\mathbf{x} = \mathbf{z} + \mathbf{w}$ is $\sigma_{\text{dir}}$-directionally smooth, with $\sigma_{\text{dir}} = \sqrt{2\pi}\sigma$.*

*Proof.* Fix some $\mathbf{u} \in \mathcal{S}^{d-1}$ and $c \in \mathbb{R}$. Then note that

$$\begin{aligned}
\mathbb{P}\left(|\langle \mathbf{x}, \mathbf{u}\rangle - c| < \delta\right) &= \mathbb{P}\left(|\langle \mathbf{w}, \mathbf{u}\rangle - (-\langle \mathbf{z}, \mathbf{u}\rangle + c)| < \delta\right) \\
&= \mathbb{P}\left(|\langle \mathbf{w}, \mathbf{u}\rangle - c'| < \delta\right) \\
&\leq \frac{\delta}{\sqrt{2\pi}\sigma},
\end{aligned}$$

where the last inequality follows by the fact that $\langle \mathbf{w}, \mathbf{u}\rangle$ is distributed as a centred univarate Gaussian with variance $\sigma^2$ and the fact that such a distribution has density upper bounded by $\frac{1}{\sqrt{2\pi}\sigma}$. The result follows. ∎

**Lemma C.2.** *Let $\mathbf{w}$ be uniform on a centred Euclidean ball of radius $\sigma$. Then for any $\mathbf{z} \in \mathbb{R}^d$, if $\mathbf{w}$ is independent of $\mathbf{z}$, it holds that $\mathbf{x} = \mathbf{z} + \mathbf{w}$ is $\sigma_{\mathrm{dir}}$-directionally smooth, with $\sigma_{\mathrm{dir}} \geq \frac{\sigma}{2}$.*

*Proof.* Let $\mathbf{v}$ denote a point sampled uniformly from the unit Euclidean ball and note that $\mathbf{w} \stackrel{d}{=} \sigma\mathbf{v}$. We then have for any $\mathbf{u} \in \mathbb{S}^{d-1}$,

$$\mathbb{P}\left(|\langle \mathbf{w}, \mathbf{u}\rangle - c'| < \delta\right) = \mathbb{P}\left(|\langle \mathbf{v}, \mathbf{u}\rangle - c| < \frac{\delta}{\sigma}\right).$$

Let $A = \left(c - \frac{\delta}{\sigma}, c + \frac{\delta}{\sigma}\right)$ and let $\phi : \mathbb{R}^d \to \mathbb{R}$ be defined so that $\phi(\mathbf{v}) = \langle \mathbf{v}, \mathbf{u}\rangle$. We note that $D\phi = \mathbf{u}^T$ and thus $\det\left(D\phi D\phi^T\right) = 1$ uniformly. Using the co-area formula [18], we see that

$$
\begin{aligned}
\mathbb{P}\left(|\langle \mathbf{v}, \mathbf{u}\rangle - c| < \frac{\delta}{\sigma}\right) &= \int_{\phi^{-1}(A)} d\operatorname{vol}_d(\mathbf{v}) \\
&= \int_{\phi^{-1}(A)} \sqrt{\det\left(D\phi(\mathbf{v})D\phi(\mathbf{v})^T\right)} d\operatorname{vol}_d(\mathbf{v}) \\
&= \int_A \operatorname{vol}_{d-1}\left(\phi^{-1}(y)\right) dy \\
&\leq \left(\sup_{y \in A} \operatorname{vol}_{d-1}\left(\phi^{-1}(y)\right)\right) \int_{c-\frac{\delta}{\sigma}}^{c+\frac{\delta}{\sigma}} dy \\
&\leq \frac{2\delta}{\sigma}.
\end{aligned}
$$

The result follows. ∎

## C.2 Directional Smoothness Equivalent to Lebesgue Density

**Lemma C.3.** *Let $\mathbf{x} \in \mathbb{R}^d$ be a (Borel-measurable) random vector. Then, $\mathbf{x}$ is $\sigma_{\mathrm{dir}}$-smooth if and only if, for any $\mathbf{w} \in \mathbb{S}^{d-1}$, $\langle \mathbf{w}, \mathbf{x}\rangle$ admits a density $p(\cdot)$ with respect to the Lebesgue measure on $\mathbb{R}$ with $\operatorname{ess\,sup}_{v \in \mathbb{R}} p(v) \leq 2/\sigma_{\mathrm{dir}}$.*

*Proof.* The "if" direction is immediate. For the converse, let $\mu(B)$ denote the Lebesgue measure of a Borel set $B \subset \mathbb{R}$. We observe that $\sigma_{\mathrm{dir}}$-smoothness proves that for any interval $I = (a, b]) \subset \mathbb{R}$, $\mathbb{P}[\langle \mathbf{w}, \mathbf{x}\rangle \in I] \leq \sigma_{\mathrm{dir}}\mu(I)/2$ (consider $c = (a + b)/2$ and $\delta = |b - a|/2$). Since the Borel sigma-algebra is the generated by open intervals, this implies that for any Borel subset $B$ of $\mathbb{R}$, $\mathbb{P}[\langle \mathbf{w}, \mathbf{x}\rangle \in B] \leq 2\mu(B)/\sigma_{\mathrm{dir}}$, where $\mu(\cdot)$ denotes the Lebesgue measure. Thus we see that the law of $\langle \mathbf{w}, \mathbf{x}\rangle$ is absolutely continuous with respect to the Lebesgue measure and, by definition of the Radon-Nikodym derivative, there exists some $p$ such that for all Borel $B$, it holds that

$$\mathbb{P}\left(\langle \mathbf{w}, \mathbf{x}\rangle \in B\right) = \int_B p(a)da.$$

Now, let

$$B = \left\{a \,\middle|\, p(a) > \frac{2}{\sigma_{\mathrm{dir}}}\right\}$$

and note that

$$\mathbb{P}\left(\langle \mathbf{w}, \mathbf{x}\rangle \in B\right) = \int_B p(a)da > \frac{2}{\sigma_{\mathrm{dir}}} \cdot \mu(B).$$

Combining this with the fact that $\mathbb{P}\left(\langle \mathbf{w}, \mathbf{x}\rangle \in B\right) \leq \frac{2\mu(B)}{\sigma_{\mathrm{dir}}}$, we see that $\mu(B) = 0$ and the result holds. ∎

## C.3 Concatenation Preserves Directional Smoothness

**Lemma C.4.** *Let $\mathbf{x}_1 \in \mathbb{R}^{d_1}$ and $\mathbf{x}_2 \in \mathbb{R}^{d_2}$ be two random vectors such that $\mathbf{x}_1 \mid \mathbf{x}_2$ and $\mathbf{x}_2 \mid \mathbf{x}_1$ are both $\sigma_{\mathrm{dir}}$-directionally smooth. Then, the concatenated vector $\tilde{\mathbf{x}} = (\mathbf{x}_1, \mathbf{x}_2)$ is $\frac{\sigma_{\mathrm{dir}}}{\sqrt{2}}$-directionally smooth.*

*Proof.* Fix $\tilde{\mathbf{w}} = (\mathbf{w}_1, \mathbf{w}_2) \in \mathbb{R}^{d_1+d_2}$ with $\|\tilde{\mathbf{w}}\| = 1$. Set $\alpha_1 := \|\mathbf{w}_1\|$ and $\alpha_2 = \|\mathbf{w}_2\|$. Assume without loss of generality that $\alpha_1 \geq \alpha_2$, so then necessarily $\alpha_1 \geq 1/\sqrt{2}$. Then,

$$
\begin{aligned}
\mathbb{P}[|\langle \tilde{\mathbf{w}}, \tilde{\mathbf{x}} \rangle - c| < \delta] &= \mathbb{E}_{\mathbf{w}_2} \, \mathbb{P}[|\langle \tilde{\mathbf{w}}, \tilde{\mathbf{x}} \rangle - c| < \delta \mid \mathbf{w}_2] \\
&= \mathbb{E}_{\mathbf{w}_2} \, \mathbb{P}\left[|\langle \mathbf{w}_1, \mathbf{x}_1 \rangle - (c - \langle \mathbf{w}_2, \mathbf{x}_2 \rangle)| < \delta \mid \mathbf{w}_2\right] \\
&= \mathbb{E}_{\mathbf{w}_2} \, \mathbb{P}\left[\left|\langle \frac{1}{\alpha_1}\mathbf{w}_1, \mathbf{x}_1 \rangle - \frac{1}{\alpha_1}(c - \langle \mathbf{w}_2, \mathbf{x}_2 \rangle)\right| < \frac{\delta}{\alpha_1} \mid \mathbf{w}_2\right] \leq \frac{\delta}{\sigma_{\mathrm{dir}}\alpha_1} \leq \frac{\sqrt{2}\delta}{\sigma_{\mathrm{dir}}}.
\end{aligned}
$$

The bound follows. ∎

**Lemma C.5.** *Let $\mathbf{z} \in \mathbb{R}^{d_1}$ and $\mathbf{u}, \mathbf{v} \in \mathbb{R}^{d_2}$ be random vectors such that $\mathbf{z} \mid \mathbf{v}$ is $\sigma_{\mathrm{dir}}$-directionally smooth, $\mathbf{v} \mid \mathbf{z}$ is $\sigma_{\mathrm{dir}}$-directionally smooth, and $\mathbf{u} = \mathbf{K}\mathbf{z} + \mathbf{v}$. Then, the concatenated vector $\mathbf{x} = (\mathbf{z}, \mathbf{u})$ is $\sigma_{\mathrm{dir}}/\sqrt{(1+\|\mathbf{K}\|_{\mathrm{op}})^2 + 1}$-smooth.*

*Proof.* Fix $\bar{\mathbf{w}} = (\mathbf{w}_1, \mathbf{w}_2) \in \mathbb{R}^{d_1+d_2}$ such that $\|\bar{\mathbf{w}}\| = 1$. Then,

$$
\begin{aligned}
\langle \bar{\mathbf{w}}, \mathbf{x} \rangle &= \langle \mathbf{w}_1, \mathbf{z} \rangle + \langle \mathbf{w}_2, \mathbf{K}\mathbf{z} \rangle + \langle \mathbf{w}_2, \mathbf{v} \rangle \\
&= \langle \mathbf{w}_1 + \mathbf{K}^\top \mathbf{w}_2, \mathbf{z} \rangle + \langle \mathbf{w}_2, \mathbf{v} \rangle.
\end{aligned}
$$

Define $\alpha_1 := \|\mathbf{w}_1 + \mathbf{K}^\top \mathbf{w}_2\|$, $\alpha_2 := \|\mathbf{w}_2\|$, and $X_1 := \langle \mathbf{w}_1 + \mathbf{K}^\top \mathbf{w}_2, \mathbf{z} \rangle$ and $X_2 := \langle \mathbf{w}_2, \mathbf{v} \rangle$. $X_1$ dependends only on $z$ and $X_2$ only on $v$. Hence, $X_1 \mid X_2$ is $\alpha_1 \sigma_{\mathrm{dir}}$ smooth and $X_2 \mid X_1$ is $\alpha_2 \sigma_{\mathrm{dir}}$ smooth. It follows that

$$
\mathbb{P}[|X_1 + X_2 - c| \leq \delta] = \mathbb{E}_{X_1} \, \mathbb{P}[|X_1 + X_2 - c| \leq \delta \mid X_1] \leq \frac{\delta}{\sigma_{\mathrm{dir}}\alpha_2},
$$

and so my symmetry under labels $i = 1, 2$,

$$
\mathbb{P}[|\langle \tilde{\mathbf{w}}, \mathbf{x} \rangle - c| \leq \delta] = \mathbb{P}[|X_1 + X_2 - c| \leq \delta] \leq \min\left\{\frac{1}{\alpha_1}, \frac{1}{\alpha_2}\right\} \frac{\delta}{\sigma_{\mathrm{dir}}} = \frac{1}{\max\{\alpha_1, \alpha_2\}} \cdot \frac{\delta}{\sigma_{\mathrm{dir}}}.
$$

We continue by bounding

$$
\begin{aligned}
\max\{\alpha_1, \alpha_2\} &= \max\{\|\mathbf{w}_1 + \mathbf{K}^\top \mathbf{w}_2\|, \|\mathbf{w}_2\|\} \\
&\geq \max\{\|\mathbf{w}\|_1 - \|\mathbf{K}\|_{\mathrm{op}}\|\mathbf{w}_2\|, \|\mathbf{w}_2\|\} \\
&= \min_{\alpha \in [0,1]} \max\{\sqrt{1-\alpha^2} - \|\mathbf{K}\|_{\mathrm{op}}\alpha, \, \alpha\}. \qquad (\|\mathbf{w}_1\|^2 + \|\mathbf{w}_2\|^2 = 1)
\end{aligned}
$$

The above is minimized when $\sqrt{1-\alpha^2} - \|\mathbf{K}\|_{\mathrm{op}}\alpha = \alpha$, so that $\alpha^2(1 + \|\mathbf{K}\|_{\mathrm{op}})^2 = 1 - \alpha^2$, yielding $\alpha = 1/\sqrt{(1+\|\mathbf{K}\|_{\mathrm{op}})^2 + 1}$. Hence, $\max\{\alpha_1, \alpha_2\} \geq 1/\sqrt{(1+\|\mathbf{K}\|_{\mathrm{op}})^2 + 1}$. The bound follows. ∎

## C.4 Smoothness of Parameters Induces Separation

In this section, we show that if the true parameters $\Theta_i^\star$ are taken from a smooth distribution, then with high probability Assumption 5 is satisfied with $\Delta_{\mathrm{sep}}$ not too small. In particular, we have the following result:

**Proposition 10.** *Suppose that $(\Theta_1^\star, \ldots, \Theta_K^\star)$ are sampled from a joint distribution on the $K$-fold product of the Frobenius-norm ball of radius $R$ in $\mathbb{R}^{m \times d}$. Suppose that the distribution of $(\Theta_1^\star, \ldots, \Theta_K^\star)$ is such that for all $1 \leq i < j \leq K$, the distribution of $\Theta_j^\star$ conditioned on the value of $\Theta_i^\star$ is $\sigma_{\mathrm{dir}}$-directionally smooth. Then, with probability at least $1 - \delta$, Assumption 5 is satisfied with*

$$
\Delta_{\mathrm{sep}} \geq \frac{md}{4\sqrt{\pi}} \cdot \left(\frac{\sigma_{\mathrm{dir}}\delta}{K^2}\right)^{\frac{1}{md}}.
$$

*Proof.* By a union bound, we have

$$
\begin{aligned}
\mathbb{P}\left(\min_{1 \leq i < j \leq K} \|\boldsymbol{\Theta}_i^\star - \boldsymbol{\Theta}_j^\star\|_{\mathrm{F}} < \Delta_{\mathrm{sep}}\right) &\leq K^2 \max_{1 \leq i < j \leq K} \mathbb{P}\left(\|\boldsymbol{\Theta}_i^\star - \boldsymbol{\Theta}_j^\star\|_{\mathrm{F}} < \Delta_{\mathrm{sep}}\right) \\
&= K^2 \max_{1 \leq i < j \leq K} \mathbb{E}_{\boldsymbol{\Theta}_i^\star}\left[\mathbb{P}\left(\|\widehat{\boldsymbol{\Theta}}_i - \widehat{\boldsymbol{\Theta}}_j\|_{\mathrm{F}} < \Delta_{\mathrm{sep}}|\boldsymbol{\Theta}_i^\star\right)\right] \\
&\leq K^2 \sup_{\boldsymbol{\Theta}_i^\star \in \mathbb{R}^{m \times d}} \mathbb{P}\left(\|\boldsymbol{\Theta}_j^\star - \boldsymbol{\Theta}_i^\star\|_{\mathrm{F}} < \Delta_{\mathrm{sep}}|\boldsymbol{\Theta}_i^\star\right) \\
&\leq K^2 \cdot \frac{\mathrm{vol}\left(\mathcal{B}_{\Delta_{\mathrm{sep}}}^{md}\right)}{\sigma_{\mathrm{dir}}},
\end{aligned}
$$

where $\mathcal{B}_{\Delta_{\mathrm{sep}}}^{md}$ denotes the Euclidean ball in $\mathbb{R}^{m \times d}$ of radius $\Delta_{\mathrm{sep}}$ and the last inequality follows by the smoothness assumption. Note that

$$
\mathrm{vol}\left(\mathcal{B}_{\Delta_{\mathrm{sep}}}^{md}\right) = \frac{\pi^{\frac{md}{2}}}{\Gamma\left(\frac{md}{2} + 1\right)} \cdot \Delta_{\mathrm{sep}}^{md}
$$

and thus

$$
\mathbb{P}\left(\min_{1 \leq i < j \leq K} \|\boldsymbol{\Theta}_i^\star - \boldsymbol{\Theta}_j^\star\|_{\mathrm{F}} < \Delta_{\mathrm{sep}}\right) \leq \frac{K^2 \pi^{\frac{md}{2}} \Delta_{\mathrm{sep}}^{md}}{\sigma_{\mathrm{dir}} \cdot \Gamma\left(\frac{md}{2} + 1\right)}.
$$

Noting that $\Gamma\left(\frac{md}{2} + 1\right)^{\frac{1}{md}} \geq \frac{md}{4}$ concludes the proof. ∎

**Remark 4.** *Note that by the previous result, Assumption 5 is in some sense generic. Indeed, in the original smoothed analysis of algorithms [57], it was assumed that the parameter matrices were smoothed by Gaussian noise; if, in addition to smoothness in contexts $\mathbf{x}_t$ we assume that the $\boldsymbol{\Theta}_i^\star$ are drawn from a directionally smooth distribution, then Proposition 10 below implies that with probability at least $1 - T^{-1}$, it holds that $\Delta_{\mathrm{sep}} \gtrsim md \left(\frac{\sigma_{\mathrm{dir}}}{K^2 T}\right)^{\frac{1}{md}}$. Furthermore, one reason why removing the gap assumption is difficult in our framework is that, computationally speaking, agnostically learning halfspaces is hard [25]. Without Assumption 5, ERMORACLE cannot reliably separate modes and thus the postprocessing steps our main algorithm (Algorithm 1) used to stabilize the predictions must also agnostically learn the modes; together with the previous observation on the difficulty of learning halfspaces, this suggests that if ERMORACLE is unable to separate modes, there is significant technical difficulty in achieving a oracle-efficient, no-regret algorithm.*

# Part II

# Supporting Proofs

## D   Parameter Recovery

In this section, we fix $\tau$ and let $\left\{\widehat{\boldsymbol{\Theta}}_i | i \in [K]\right\}, \widehat{g}$ denote the output of ERMORACLE$(\bar{\mathbf{x}}_{1:\tau E}, \mathbf{y}_{1:\tau E})$. For any $g \in \mathcal{G}$ and $i, j \in [K]$, we denote

$$
I_{ij}(g) = \left\{1 \leq t \leq \tau E | g(\bar{\mathbf{x}}_t) = i \text{ and } g_\star(\bar{\mathbf{x}}_t) = j\right\}. \tag{D.1}
$$

We will show the following result:

**Theorem 11** (Parameter Recovery). *Suppose that Assumptions 2-4 hold. Then there is a universal constant $C$ such that for any tunable parameter $\xi \in (0, 1)$ (which appears in the analysis but not in the algorithm), with probability at least $1 - \delta$, it holds for all $1 \leq i, j \leq K$ satisfying*

$$
|I_{ij}(\widehat{g})| \geq CK^2 T\xi + C\frac{B^8 Kd}{\sigma_{\mathrm{dir}}^8 \xi^8} \log\left(\frac{BKT}{\sigma_{\mathrm{dir}} \xi \delta}\right),
$$

*it holds that*

$$\|\widehat{\boldsymbol{\Theta}}_i - \boldsymbol{\Theta}_j^\star\|_{\mathrm{F}}^2 \le C \frac{B^2}{\sigma_{\mathrm{dir}}^2 \xi^2 |I_{ij}(\widehat{g})|} \left( \varepsilon_{\mathrm{orac}} + 1 + K^3 B^2 R^2 d^2 m \nu^2 \sqrt{T} \log \left( \frac{TRBmdK}{\delta} \right) \right) + 4BRK^2 \frac{1}{|I_{ij}(g)|} \sum_{t \in I_{ij}(g)} \|\boldsymbol{\delta}_t\|$$

We refer the reader to the notation table in Appendix A for a reminder about the parameters. We begin by defining for any $i, j \in [K]$ and $g \in \mathcal{G}$,

$$\Sigma_{ij}(g) = \sum_{\substack{1 \le t \le \tau E \\ t \in I_{ij}(g)}} \bar{\mathbf{x}}_t \bar{\mathbf{x}}_t^T,$$

the empirical covariance matrix on those $t \in I_{ij}(g)$. We will first show that for a fixed $g$, $\Sigma_{ij}(g) \succeq cI$ for a sufficiently small $c$ depending on problem parameters. We will then introduce a complexity notion we call a disagreement cover that will allow us to lift this statement to one uniform in $\mathcal{G}$, which will imply that $\Sigma_{ij}(\widehat{g}) \succeq cI$. We will then use the definition of ERMORACLE to show that $\sum_t \|(\widehat{\boldsymbol{\Theta}}_{\widehat{g}(\bar{\mathbf{x}}_t)} - \boldsymbol{\Theta}_{g_\star(\bar{\mathbf{x}}_t)}^\star)\bar{\mathbf{x}}_t\|_{\mathrm{F}}^2$ is small and the theorem will follow.

For the entirety of the proof and without loss of generality, we will assume that $T/E \in \mathbb{Z}$. Indeed, if $T$ is not a multiple of $E$ then we suffer regret at most $O(E)$ on the last episode, which we will see does not adversely affect our rates.

## D.1 Disagreement Covers

We begin by introducing a notion of complexity we call a disagreement cover; in contrast to standard $\varepsilon$-nets, we show below that the disagreement cover provides more uniform notion of coverage. Moreover, we then show that the size of a disagreement cover of $\mathcal{G}$ can be controlled under the assumption of directional smoothness.

**Definition 12.** *Let $\mathscr{D} = \left\{ (g_i, \mathscr{D}_i) \, | \, g_i \in \mathcal{G} \text{ and } \mathscr{D}_i \subset \mathbb{R}^d \right\}$. We say that $\mathscr{D}$ is an $\varepsilon$-disagreement cover if the following two properties hold:*

1. *For every $g \in \mathcal{G}$, there exists some $i$ such that $(g_i, \mathscr{D}_i) \in \mathscr{D}$ and $\left\{ \mathbf{x} \in \mathbb{R}^d \, | \, g_i(\mathbf{x}) \ne g(\mathbf{x}) \right\} \subset \mathscr{D}_i$.*

2. *For all $i$ and $t$, it holds that $\mathbb{P}\left( \mathbf{x}_t \in \mathscr{D}_i \mid \mathscr{F}_{t-1} \right) \le \varepsilon$*

*We will denote by $\mathrm{DN}(\mathcal{G}, \varepsilon)$ (or $\mathrm{DN}(\varepsilon)$ when $\mathcal{G}$ is clear from context) the minimal size of an $\varepsilon$-disagreement cover of $\mathcal{G}$.*

We remark that a disagreement cover is stronger than the more classical notion of an $\varepsilon$-net because the sets of points where multiple different functions $g$ disagree with a single element of the cover $g_i$ has to be contained in a single set. With an $\varepsilon$-net, there is nothing stopping the existence of an $i$ such that the set of points on which at least one $g$ satisfying $\mathbb{P}(g_i \ne g) \le \varepsilon$ is the entire space. The reason that this uniformity is necessary is to provide the following bound:

**Lemma D.1.** *Let $\mathscr{D}$ be an $\varepsilon$-disagreement cover for $\mathcal{G}$. Then, with probability at least $1 - \delta$, it holds that*

$$\sup_{g \in \mathcal{G}} \min_{(g_i, \mathscr{D}_i) \in \mathscr{D}} \sum_{t=1}^T \mathbb{I}\left[ g(\mathbf{x}_t) \ne g_i(\mathbf{x}_t) \right] \le 2T\varepsilon + 6 \log \left( \frac{|\mathscr{D}|}{\delta} \right).$$

*Proof.* Note that by the definition of a disagreement cover, it holds that

$$\sup_{g \in \mathcal{G}} \min_{(g_i, \mathscr{D}_i) \in \mathscr{D}} \sum_{t=1}^T \mathbb{I}\left[ g(\mathbf{x}_t) \ne g_i(\mathbf{x}_t) \right] \le \max_{(g_i, \mathscr{D}_i) \in \mathscr{D}} \sum_{t=1}^T \mathbb{I}\left[ \mathbf{x}_t \in \mathscr{D}_i \right].$$

Note that for any fixed $i$, it holds that $\mathbb{P}\left( \mathbf{x}_t \in \mathscr{D}_i | \mathscr{F}_{t-1} \right) \le \varepsilon$, also by construction. Applying a Chernoff bound (Lemma D.2), we see that

$$\mathbb{P}\left( \sum_{t=1}^T \mathbb{I}\left[ \mathbf{x}_t \in \mathscr{D}_i \right] \ge 2T\varepsilon + 6 \log \left( \frac{1}{\delta} \right) \right) \le \delta.$$

Taking a union bound over $\mathscr{D}_i \in \mathscr{D}$ concludes the proof. ∎

For the sake of completeness, we state and prove the standard Chernoff Bound with dependent data used in the previous argument:

**Lemma D.2** (Chernoff Bound). *Suppose that $X_1, \ldots, X_T$ are random variables such that $X_t \in \{0,1\}$ for all $1 \le t \le T$. Suppose that there exist $p_t$ such that $\mathbb{P}(X_t = 1 | \mathscr{F}_{t-1}) \le p_t$ almost surely, where $\mathscr{F}_{t-1}$ is the $\sigma$-algebra generated by $X_1, \ldots, X_{t-1}$. Then*

$$\mathbb{P}\left(\sum_{t=1}^{T} X_t > 2\sum_{t=1}^{T} p_t + \frac{1}{2}\log\left(\frac{1}{\delta}\right)\right) \le \delta$$

*Proof.* We use the standard Laplace transform trick:

$$\mathbb{P}\left(\sum_{t=1}^{T} X_t > 2\sum_{t=1}^{T} p_t + u\right) = \mathbb{P}\left(e^{\lambda \sum_{t=1}^{T} X_t} > e^{2\lambda u + 2\lambda \sum_{t=1}^{T} p_t}\right)$$

$$\le e^{-2\lambda u - 2\lambda \sum_{t=1}^{T} p_t} \cdot \mathbb{E}\left[\prod_{t=1}^{T} e^{\lambda X_t}\right]$$

$$= e^{-2\lambda u - 2\lambda \sum_{t=1}^{T} p_t} \cdot \mathbb{E}\left[\prod_{t=1}^{T} \mathbb{E}\left[e^{\lambda X_t} | \mathscr{F}_{t-1}\right]\right]$$

$$\le e^{-2\lambda u - 2\lambda \sum_{t=1}^{T} p_t} \cdot \mathbb{E}\left[\prod_{t=1}^{T} e^{\lambda} \mathbb{P}(X_t = 1 | \mathscr{F}_{t-1}) + (1 - \mathbb{P}(X_t = 1 | \mathscr{F}_{t-1}))\right]$$

$$\le e^{-2\lambda u - 2\lambda \sum_{t=1}^{T} p_t} \cdot \prod_{t=1}^{T} e^{(e^{\lambda} - 1)p_t}.$$

Thus we see that

$$\mathbb{P}\left(\sum_{t=1}^{T} X_t > 2\sum_{t=1}^{T} p_t + u\right) \le e^{-2\lambda u + \left(e^{\lambda} - 1 - 2\lambda\right)\sum_{t=1}^{T} p_t}.$$

Setting $\lambda = 1$ and noting that $\sum_{t=1}^{T} p_t > 0$ tells us that

$$\mathbb{P}\left(\sum_{t=1}^{T} X_t > 2\sum_{t=1}^{T} p_t + u\right) \le e^{-2u}$$

and the result follows. ∎

Returning to the main thread, we see that Lemma D.1 allows us to uniformly bound the approximation error of considering a disagreement cover. Before we can apply the result, however, we need to show that this complexity notion is small for the relevant class, $\mathcal{G}$. We have the following result:

**Lemma D.3.** *Let $\mathcal{G}$ be the set of classifiers considered above. Then it holds that*

$$\log\left(\mathrm{DN}(\mathcal{G}, \varepsilon)\right) \le K(d+1)\log\left(\frac{3BK}{\varepsilon}\right)$$

We prove the result in two parts. For the first part, we show that any function class that is constructed by aggregating $K$ simpler classes has a disagreement cover whose size is controlled by that of the $K$ classes:

**Lemma D.4.** *Let $\mathcal{G}_1, \ldots, \mathcal{G}_K$ be function classes mapping $\mathcal{X} \to \mathcal{Y}$. Let $h : \mathcal{Y}^{\times K} \to \mathbb{R}$ be some aggregating function and define*

$$\mathcal{G} = \{\mathbf{x} \mapsto h(g_1(\mathbf{x}), \ldots, g_K(\mathbf{x})) \mid g_1 \in \mathcal{G}_1, \ldots, g_K \in \mathcal{G}_K\}$$

*Let $\mathrm{DN}(\mathcal{G}_i, \varepsilon)$ be the minimal size of an $\varepsilon$-disagreement cover of $\mathcal{G}_i$. Then*

$$\mathrm{DN}(\mathcal{G}, \varepsilon) \le \prod_{i=1}^{K} \mathrm{DN}\left(\mathcal{G}_i, \frac{\varepsilon}{K}\right)$$

*Proof.* Suppose that $\mathscr{D}_i$ are $\left(\frac{\varepsilon}{K}\right)$-disagreement covers for $\mathcal{G}_i$ and for any $g_i \in \mathcal{G}_i$ denote by $(\pi(g_i), \mathscr{D}_i(g_i))$ the pair of functions and disagreement sets satisfying the definition of a disagreement cover. Then we claim that

$$\mathscr{D} = \left\{ \left( h(g_1, \ldots, g_K), \bigcup_i \mathscr{D}_i \right) \mid (g_i, \mathscr{D}_i) \in \mathscr{D}_i \right\}$$

is an $\varepsilon$-disagreement cover for $\mathcal{G}$. Note that $|\mathscr{D}|$ is clearly bounded by the desired quantity so this claim suffices to prove the result.

To prove the claim, we first note that a union bound ensures that $\mathbb{P}(\mathbf{x}_t \in \bigcup_i \mathscr{D}_i \mid \mathscr{F}_{t-1}) \leq \varepsilon$. We further note that if $g = h(g_1, \ldots, g_K)$ then

$$\{\mathbf{x} \mid g(\mathbf{x}) \neq h(\pi(g_1(\mathbf{x})), \ldots, \pi(g_K(\mathbf{x})))\} \subset \bigcup_{1 \leq i \leq K} \{\mathbf{x} \mid g_i(\mathbf{x}) \neq \pi(g_i(\mathbf{x}))\} \subset \bigcup_{1 \leq i \leq K} \mathscr{D}_i(g_i).$$

The result follows. ■

We now prove that the class of linear threshold functions has bounded disagreement cover:

**Lemma D.5.** *Let* $\mathcal{H} : \mathbb{R}^d \to \{\pm 1\}$ *be the class of affine thresholds given by* $\mathbf{x} \mapsto \text{sign}(\langle \mathbf{w}, \mathbf{x} \rangle + b)$ *for some* $\mathbf{w} \in B^d$ *and some* $b \in [-B, B]$. *Then*

$$\log \text{DN}(\mathcal{H}, \varepsilon) \leq (d+1) \log \left( \frac{3B}{\varepsilon} \right).$$

*Proof.* We first let $\mathcal{N} := \{(\mathbf{w}_i, b_i)\}$ denote an $\varepsilon$-net on $\mathcal{B}^d \times [-B, B]$. Associate each $(\mathbf{w}_i, b_i)$ to its corresponding classifier $h_i(\mathbf{x}) \text{sign}(\langle \mathbf{w}_i, \mathbf{x} \rangle + b_i)$. We claim that for each $h_i$, there is a region $\mathscr{D}_i \subset \mathbb{R}^d$ such that $\{(h_i, \mathscr{D}_i)\}$ is an $\varepsilon$-disagreement cover. To see this, consider some $h$ with parameters $(\mathbf{w}, b)$, and let $h_i$ with parameters $(\mathbf{w}_i, b_i)$ ensure $||\mathbf{w}_i - \mathbf{w}|| + |b_i - b| \leq \sigma_{\text{dir}}\varepsilon/B$. Consider any $\mathbf{x}$ such that

$$\text{sign}\left(\langle \mathbf{w}_i, \mathbf{x} \rangle + b_i\right) \neq \text{sign}\left(\langle \mathbf{w}, \mathbf{x} \rangle + b\right)$$

By the continuity of affine functions, there is some $\lambda \in (0, 1)$ such that if $\mathbf{w}_\lambda = (1 - \lambda)\mathbf{w}_i + \lambda\mathbf{w}$ and $b_\lambda = (1 - \lambda)b_i + \lambda b$ then

$$\langle \mathbf{w}_\lambda, \mathbf{x} \rangle + b_\lambda = 0$$

Note however that

$$|\langle \mathbf{w}_\lambda, \mathbf{x} \rangle + b_\lambda - \langle \mathbf{w}_i, \mathbf{x} \rangle - b_i| \leq \lambda |\langle \mathbf{w} - \mathbf{w}_i, \mathbf{x} \rangle| + \lambda |b_i - b| \leq B \left( \frac{\sigma_{\text{dir}}\varepsilon}{B} \right) \leq \sigma_{\text{dir}}\varepsilon$$

by the definition of our $\varepsilon$-net. Thus, let

$$\mathscr{D}_i = \left\{ \mathbf{x} \mid |\langle \mathbf{w}_i, \mathbf{x} \rangle + b| \leq \frac{\sigma_{\text{dir}}\varepsilon}{B} \right\}$$

and note that the above computation tells us that if $\mathbf{x} \notin \mathscr{D}_i$ then $h_i$ must agree with $h = (\mathbf{w}, b)$ for all $h$ that are mapped to $h_i$ by the projection onto the $\varepsilon$-net. Thus for all such $h$, it holds that

$$\mathbb{P}\left(h(\mathbf{x}_t) \neq h_i(\mathbf{x}_t) | \mathscr{F}_{t-1}\right) \leq \mathbb{P}\left( |\langle \mathbf{w}_i, \mathbf{x}_t \rangle + b| \leq \frac{\sigma_{\text{dir}}\varepsilon}{B} | \mathscr{F}_{t-1} \right) \leq \varepsilon.$$

Thus the claim holds and we have constructed an $\varepsilon$-disagreement cover. By noting that there are at most $\left(\frac{3}{\varepsilon}\right)^d \cdot \frac{B}{\varepsilon}$ members of this cover by a volume argument we conclude the proof. ■

By combining Lemma D.4 and Lemma D.5, we prove Lemma D.3. In the next section, we will apply Lemma D.3 and Lemma D.1 to lower bound $\Sigma_{ij}(\widehat{g})$.

## D.2 Lower Bounding the Covariance

We continue by lower bounding $\Sigma_{ij}(\widehat{g})$. Before we begin, we introduce some notation. For any $g \in \mathcal{G}$, we will let

$$Z_{t;ij}(g) = \mathbb{I}\left[g(\bar{\mathbf{x}}_t) = i \text{ and } g_\star(\bar{\mathbf{x}}_t) = j\right] \qquad \bar{Z}_{t;ij}(g) = \mathbb{E}\left[Z_{t;ij}(g)|\mathscr{F}_{t-1}\right]$$

or, in words, $Z_{t;ij}(g)$ is the indicator of the event that a classifier predicts label $i$ when $g_\star$ predicts $j$ and $\bar{Z}_{t;ij}(g)$ is probability of this event conditioned on the x-history. We will work under the small-ball assumption that there exist constants $c_0, c_1$ as well as $\zeta_t, \rho_t$ such that for any $\mathbf{w} \in \mathcal{S}^d$, it holds for all $g \in \mathcal{G}$ that

$$\mathbb{P}\left(\langle\bar{\mathbf{x}}_t, \mathbf{w}\rangle^2 \geq c_0\zeta_t^2 \mid \mathscr{F}_t, Z_{t;ij}(g) = 1\right) \geq c_1\rho_t \tag{D.2}$$

We will show that (D.2) holds and defer control of the values of $\zeta_t, \rho_t$ to Appendix D.3, but, for the sake of clarity, we take these constants as given for now. We proceed to show that for a single function $g$, that for all $i, j$ such that $I_{ij}(g)$ is big, it holds that $\Sigma_{ij}(g)$ is also large. We will then apply our results in the previous section to lift this statement to one uniform in $\mathcal{G}$. We have the following result:

**Lemma D.6.** *Let $\rho = \min_t \rho_t$ and let $\zeta = \min_t \zeta_t$, where $\rho_t, \zeta_t$ are from (D.2). Then with probability at least $1 - \delta$, for all $1 \leq i, j \leq K$ and all $g \in \mathcal{G}$ such that*

$$|I_{ij}(g)| \geq \max\left(\frac{2}{\rho^2}\left(\log\left(\frac{2T}{\delta}\right) + \frac{d}{2}\log\left(C\frac{B\zeta^2}{\rho}\right) + \log\left(K^2\mathrm{DN}(\varepsilon)\right)\right), C\frac{B^2}{\zeta^2}\left(2T\varepsilon + 6\log\left(\frac{\mathrm{DN}(\varepsilon)}{\delta}\right)\right)\right)$$

*it holds that*

$$\Sigma_{ij}(g) \succeq \frac{c_0\zeta^2}{8}|I_{ij}(g)|$$

We will prove Lemma D.6 by first fixing $g \in \mathcal{G}$ and showing the statement for the fixed $g$ and then using the results of Appendix D.3 to make the statements uniform in $\mathcal{G}$. To prove the statement for a fixed $g$, we require the following self-normalized martingale inequality:

**Lemma D.7.** *Let $\mathscr{F}_t$ be a filtration with $A_t \in \mathscr{F}_t$ and $B_t \in \mathscr{F}_{t-1}$ for all $t$. Let*

$$p_{A_t} = \mathbb{P}\left(A_t \mid \mathscr{F}_{t-1}, B_t\right) \qquad p_{B_t} = \mathbb{P}\left(B_t \mid \mathscr{F}_{t-1}\right)$$

*and suppose that $p_{A_t} \geq \rho$ for all t. Then, with probability at least $1 - \delta$,*

$$\sum_{t=1}^{T}\mathbb{I}[A_t]\mathbb{I}[B_t] \geq \rho\sum_{t=1}^{T}\mathbb{I}[B_t] - \frac{1}{2}\sqrt{2\left(\sum_{t=1}^{T}\mathbb{I}[B_t]\right)\log\left(\frac{2T}{\delta}\right)}$$

*In particular with probability at least $1 - \delta$, if*

$$\sum_{t=1}^{T}\mathbb{I}[B_t] \geq \frac{2}{\rho^2}\log\left(\frac{2T}{\delta}\right)$$

*then*

$$\sum_{t=1}^{T}\mathbb{I}[A_t]\mathbb{I}[B_t] \geq \frac{\rho}{2}\sum_{t=1}^{T}\mathbb{I}[B_t]$$

In order to prove Lemma D.7, we will require the following general result:

**Lemma D.8** (Theorem 1 from [2])**.** *Let $(u_t)$ be predictable with respect to a filtration $(\mathscr{G}_t)$, and let $(e_t)$ be such that $e_t \mid \mathscr{G}_t$ is $\nu^2$-subGaussian. Then, for any fixed parameter $\lambda > 0$, with probability $1 - \delta$,*

$$\left(\sum_{t=1}^{T}u_t e_t\right)^2 \leq 2\nu^2\left(\lambda + \sum_{t=1}^{T}u_t^2\right)\log\frac{(1 + \lambda^{-1}\sum_{t=1}^{T}u_t^2)^{1/2}}{\delta}$$

We now present the proof of Lemma D.7:

*Proof of Lemma D.7.* We apply Lemma D.8 with $u_t = \mathbb{I}[B_t]$, $e_t = \mathbb{I}[A_t] - p_{A_t}$, and $\mathcal{G}_t = \mathcal{F}_t$. Noting that the latter is $\frac{1}{8}$-subGaussian because it is conditionally mean zero and bounded in absolute value by 1, we have with probability at least $1 - \delta$,

$$\left| \sum_{t=1}^{T} (\mathbb{I}[A_t] - p_{A_t})\mathbb{I}[B_t] \right| \leq \frac{1}{2} \sqrt{\left( \lambda + \sum_{t=1}^{T} \mathbb{I}[B_t] \right) \log \left( \frac{\sqrt{1 + \frac{\sum_{t=1}^{T} \mathbb{I}[B_t]}{\lambda}}}{\delta} \right)}$$

Taking a union bound over $\lambda \in [T]$ and noting that $\sum_{t=1}^{T} \mathbb{I}[B_t] \in [T]$ almost surely, we recover the result. ∎

We now proceed to apply Lemma D.7 to prove a version of Lemma D.6 with the function $g \in \mathcal{G}$ fixed:

**Lemma D.9.** *Suppose that* (D.2) *holds and that* $\rho = \min_t \rho_t$ *as well as* $\zeta = \min_t \zeta_t$. *For fixed* $1 \leq i, j \leq K$ *and* $g \in \mathcal{G}$, *it holds with probability at least* $1 - \delta$ *that if*

$$|I_{ij}(g)| \geq \frac{2}{\rho^2} \log \left( \frac{2T}{\delta} \right) + \frac{d}{\rho^2} \log \left( C \frac{B\zeta^2}{\rho} \right)$$

*then*

$$\Sigma_{ij}(g) \succeq \frac{c_0 \zeta^2}{4} |I_{ij}(g)|$$

*Proof.* Note that for any fixed unit vector $\mathbf{u} \in \mathbb{S}^{d-1}$, the following holds:

$$\mathbb{P} \left( \mathbf{u}^T \Sigma_{ij}(g) \mathbf{u} \leq \frac{c_0 \rho \zeta^2 |I_{ij}(g)|}{2} \right) \leq \mathbb{P} \left( \sum_{t=1}^{T} \mathbb{I} \left[ \mathbf{u}^T \mathbf{x}_t \mathbf{x}_t^T \mathbf{u} \leq c_0 \zeta^2 \right] Z_{t;ij} \leq \frac{\rho |I_{ij}(g)|}{2} \right)$$

Now let $A_t$ denote the event that $(\langle \mathbf{u}, \mathbf{x}_t \rangle)^2 \geq c_0 \zeta^2$ and let $B_t$ denote the event that $Z_{t;ij} = 1$. Noting that $\mathbb{P}(A_t | B_t) \geq \rho$ by (D.2), we may apply Lemma D.7 and note that $|I_{ij}(g)|$ is just the sum of the $B_t$ to show that if $I_{ij}(g)$ satisfies the assumed lower bound, then

$$\mathbb{P} \left( \sum_{t=1}^{T} \mathbb{I} \left[ \mathbf{u}^T \mathbf{x}_t \mathbf{x}_t^T \mathbf{u} \leq c_0 \zeta^2 \right] Z_{t;ij} \leq \frac{\rho |I_{ij}(g)|}{2} \right) \leq 1 - \delta \exp \left( -\frac{d}{2} \log \left( C \frac{R\zeta^2}{\rho} \right) \right).$$

Taking a union bound over an appropriately sized $\varepsilon$-net on $\mathbb{S}^{d-1}$ to approximate $\mathbf{u}$, and applying Block and Simchowitz [7, Lemma 45] concludes the proof. ∎

We are now ready to prove the main result in this section, i.e., a lower bound on $\Sigma_{ij}(\widehat{g})$:

*Proof of Lemma D.6.* Fix $\mathscr{D} = \{(g_i, \mathscr{D}_i)\}$ an $\varepsilon$-disagreement cover of $\mathcal{G}$ of size $\text{DN}(\varepsilon)$. Taking a union bound over all $g_i$ in $\mathscr{D}$ and then applying Lemma D.9 tells us that with probability at least $1 - \frac{\delta}{2}$, it holds for all $1 \leq i, j \leq K$ and all $g_k \in \mathscr{D}$ such that

$$|I_{ij}(g)| \geq \frac{4}{\rho^2} \log \left( \frac{2TK\text{DN}(\varepsilon)}{\delta} \right) + \frac{d}{\rho^2} \log \left( C \frac{B\zeta^2}{\rho} \right)$$

we have

$$\Sigma_{ij}(g_k) \succeq \frac{c_0 \zeta^2}{4} |I_{ij}(g)|. \tag{D.3}$$

Applying Lemma D.1, we see that with probability at least $1 - \frac{\delta}{2}$,

$$\sup_{g \in \mathcal{G}} \min_{g_k \in \mathscr{D}} \sum_{t=1}^{T} \mathbb{I} [g(\mathbf{x}_t) \neq g_k(\mathbf{x}_t)] \leq 2T\varepsilon + 12 \log \left( \frac{\text{DN}(\varepsilon)}{\delta} \right).$$

Noting that by assumption, $\left\| \mathbf{x}_t \mathbf{x}_t^T \right\| \leq B^2$, we see that for any $g \in \mathcal{G}$, there is some $g_k \in \mathscr{D}$ such that

$$\Sigma_{ij}(g) \succeq \Sigma_{ij}(g_k) - B^2 \left( 4T\varepsilon + 6 \log \left( \frac{\text{DN}(\varepsilon)}{\delta} \right) \right).$$

Thus, applying the above lower bound on $\Sigma_{ij}(g_k)$ concludes the proof. ∎

Lemma D.6 has provided a lower bound on the empirical covariance matrices under the assumption that (D.2) holds; in the next section we show that this assumption is valid.

## D.3 Small Ball Inequality

Introduce the shorthand $\mathbb{P}_t(\cdot) := \mathbb{P}(\cdot \mid \mathscr{F}_{t-1})$ and $\mathbb{E}_t[\cdot] := \mathbb{E}[\cdot \mid \mathscr{F}_{t-1}]$. Recall from the previous section that we assumed that there are constants $c_0, c_1$ and $\zeta_t, \rho_t$ such that for any $\bar{\mathbf{w}} \in \mathcal{S}^d$, and any $g \in \mathcal{G}$, it holds that

$$\mathbb{P}_t\left(\langle \bar{\mathbf{x}}_t, \bar{\mathbf{w}}\rangle^2 \geq c_0 \zeta_t^2 \mid Z_{t;ij}(g) = 1\right) \geq c_1 \rho_t. \tag{D.4}$$

In this section, we will show that smoothness and an assumption on $Z_{t;ij}(g)$ suffice to guarantee that this holds. We will show the following result:

**Lemma D.10.** *Suppose that Assumptions 4 and 2 hold. Fix a $g \in \mathcal{G}$ and $1 \leq i, j \leq K$ and suppose that*

$$\mathbb{P}_t(Z_{t;ij}(g)) \geq \xi.$$

*For all $\bar{\mathbf{w}} \in \mathcal{S}^d$, it holds that, for universal constants $c_0$ and $c_1$,*

$$\mathbb{P}_t\left(\langle \bar{\mathbf{x}}_t, \bar{\mathbf{w}}\rangle^2 \geq c_0 \frac{\sigma_{\mathrm{dir}}^2 \xi^2}{B^2} \mid Z_{t;ij}(g) = 1\right) \geq c_1 \frac{\sigma_{\mathrm{dir}}^4 \xi^4}{B^4}.$$

We begin by defining some notation. First, we let $\mathbb{V}_{\mathbf{x}_t}$ denote variance with respect to $\mathbf{x}_t$, conditioned on the $\mathbf{x}$-history and $Z_{t;ij}(g)$. That is, for some function $f$ of $\mathbf{x}_t$, we let

$$\mathbb{V}_{\mathbf{x}_t}[f] = \mathbb{E}_t\left[f(\mathbf{x}_t)^2 \mid Z_{t;ij}(g)\right] - \mathbb{E}_t\left[f(\mathbf{x}_t) \mid Z_{t;ij}(g)\right]^2. \tag{D.5}$$

For a fixed $\bar{\mathbf{w}} \in \mathcal{S}^d$, denote by $\mathbf{w}$ the first $d$ coordinates. For fixed $g \in \mathcal{G}$, and $1 \leq i, j \leq K$, we will denote:

$$\rho_t(g, \bar{\mathbf{w}}) = \left(\frac{\mathbb{V}_{\mathbf{x}_t}[\langle \mathbf{x}_t, \mathbf{w}\rangle]}{||\mathbf{w}||^2 \vee \left(\mathbb{E}_t\left[\langle \mathbf{x}_t, \mathbf{w}\rangle^2 \mid Z_{t;ij}(g) = 1\right]\right)}\right)^2 \cdot \frac{1}{32\left(1 + \mathbb{E}_t\left[\langle \mathbf{x}_t, \mathbf{w}\rangle^4 \mid Z_{t;ij}(g) = 1\right]\right)}$$

$$\zeta_t^2(g, \bar{\mathbf{w}}) = \frac{\mathbb{V}_{\mathbf{x}_t}[\langle \mathbf{x}_t, \mathbf{w}\rangle]}{||\mathbf{w}||^2 \vee \left(\mathbb{E}_t\left[\langle \mathbf{x}_t, \mathbf{w}\rangle^2 \mid Z_{t;ij}(g) = 1\right]\right)}.$$

We will now show the following result:

**Lemma D.11.** *It holds for any fixed $1 \leq i, j \leq K$ and $g \in \mathcal{G}$ that*

$$\mathbb{P}_t\left(\langle \bar{\mathbf{x}}_t, \bar{\mathbf{w}}\rangle^2 \geq c_0 \zeta_t^2(g, \bar{\mathbf{w}}) Z_{t;ij}(g) = 1\right) \geq c_1 \rho_t(g, \bar{\mathbf{w}})$$

After proving Lemma D.11, we will provide lower bounds on $\zeta_t$ and $\rho_t$ under the assumptions of Lemma D.10, which will allow us to prove the main result. To begin the proof of Lemma D.11, we will begin by showing that $\langle \bar{\mathbf{x}}_t, \bar{\mathbf{w}}\rangle^2$ is lower bounded in expectation in the following result:

**Lemma D.12.** *Suppose that Assumptions refass:dirsmooth and 4 hold. Then for any fixed $1 \leq i, j \leq K$ and $g \in \mathcal{G}$, it holds for all $\mathbf{w} \in \mathcal{S}^d$ that*

$$\mathbb{E}_t\left[\langle \bar{\mathbf{x}}_t, \bar{\mathbf{w}}\rangle^2 \mid Z_{t;ij}(g)\right] \geq \zeta_t^2(g, \bar{\mathbf{w}})$$

*Proof.* We prove two distinct lower bounds and then combine them. Fix $i, j, g$, and $\bar{\mathbf{w}}$. Recall that $\mathbf{w}$ is $\bar{\mathbf{w}}$ without its last coordinate and let $w_0$ denote the last coordinate of $\bar{\mathbf{w}}$. We compute:

$$\mathbb{E}_t\left[\langle \bar{\mathbf{x}}_t, \bar{\mathbf{w}}\rangle^2 \mid Z_{t;ij}(g) = 1\right] = \mathbb{E}_t\left[(w_0 + \langle \mathbf{x}_t, \mathbf{w}_t\rangle)^2 \mid Z_{t;ij}(g) = 1\right]$$
$$\geq \mathbb{V}_{\mathbf{x}_t}[w_0 + \langle \mathbf{x}_t, \mathbf{w}\rangle]$$

where the equality comes from noting that the last coordinate of $\bar{\mathbf{x}}_t$ is 1 by construction, the inequality is trivial. This is our first lower bound. For our second lower bound, we compute:

$$\mathbb{E}_t\left[\langle \bar{\mathbf{x}}_t, \bar{\mathbf{w}}\rangle^2 \mid Z_{t;ij}(g) = 1\right] = \mathbb{E}_t\left[(w_0 + \langle \mathbf{x}_t, \mathbf{w}\rangle)^2 \mid Z_{t;ij}(g) = 1\right]$$

$$\geq \frac{w_0^2}{2} - \mathbb{E}\left[\langle \mathbf{x}_t, \mathbf{w}\rangle^2 \mid Z_{t;ij}(g) = 1\right]$$

where the inequality follows from applying the numerical inequality $2(x + y)^2 \geq x^2 - 2y^2$. This is our second lwoer bound. To ease notation, denote

$$\alpha = \frac{\mathbb{E}_t\left[\langle \mathbf{x}_t, \mathbf{w}\rangle^2 \mid Z_{t;ij}(g) = 1\right]}{\mathbb{V}_{\mathbf{x}_t}[\langle \mathbf{x}_t, \mathbf{w}\rangle]} \geq 1.$$

Combining the two previous lower bounds, we see that

$$\mathbb{E}_t\left[\langle \bar{\mathbf{x}}_t, \bar{\mathbf{w}}\rangle^2 \mid Z_{t;ij}(g) = 1\right] \geq \max\left(\mathbb{V}_{\mathbf{x}_t}[\langle \mathbf{x}_t, \mathbf{w}\rangle], \frac{w_0^2}{2} - 2\alpha\mathbb{V}_{\mathbf{x}_t}[\langle \mathbf{x}_t, \mathbf{w}\rangle]\right)$$

$$\geq \max_{0 \leq \lambda \leq 1}\left\{(1 - \lambda)\mathbb{V}_{\mathbf{x}_t}[\langle \mathbf{x}_t, \mathbf{w}\rangle] + \lambda\frac{w_0^2}{2} - 2\lambda\alpha\mathbb{V}_{\mathbf{x}_t}[\langle \mathbf{x}_t, \mathbf{w}\rangle]\right\}$$

$$= \max_{0 \leq \lambda \leq 1}\left\{(1 - \lambda - 2\alpha\lambda)\mathbb{V}_{\mathbf{u}_t}[\langle \mathbf{u}_t, \mathbf{w}\rangle] + \lambda\frac{w_0^2}{2}\right\}$$

$$\geq \max_{0 \leq \alpha \leq 1}\left\{(1 - 3\alpha\lambda)\mathbb{V}_{\mathbf{x}_t}[\langle \mathbf{x}_t, \mathbf{w}\rangle] + \lambda\frac{w_0^2}{2}\right\}$$

$$\geq \left(1 - \frac{3\alpha}{6\alpha}\right)\mathbb{V}_{\mathbf{x}_t}[\langle \mathbf{x}_t, \mathbf{w}\rangle] + \frac{1}{12\alpha}w_0^2$$

$$= \frac{1}{2}\|\mathbf{w}\|^2\mathbb{V}_{\mathbf{x}_t}\left[\left\langle \mathbf{x}_t, \frac{\mathbf{w}}{\|\mathbf{w}\|}\right\rangle\right] + \frac{w_0^2}{2}\cdot\frac{1}{6\alpha}$$

$$\geq \frac{1}{2}\left(\|\mathbf{w}\|^2 + w_0^2\right)\min\left(\mathbb{V}_{\mathbf{x}_t}\left[\left\langle \mathbf{x}_t, \frac{\mathbf{w}}{\|\mathbf{w}\|}\right\rangle\right], \frac{1}{6\alpha}\right)$$

$$= \frac{1}{2}\min\left(\mathbb{V}_{\mathbf{x}_t}\left[\left\langle \mathbf{x}_t, \frac{\mathbf{w}}{\|\mathbf{w}\|}\right\rangle\right], \frac{\mathbb{V}_{\mathbf{x}_t}[\langle \mathbf{x}_t, \mathbf{w}\rangle]}{6\mathbb{E}_t\left[\langle \mathbf{x}_t, \mathbf{w}\rangle^2 \mid Z_{t;ij}(g) = 1\right]}\right),$$

where the third inequality follows because $\alpha \geq 1$ and the last equality follows because $1 = \|\bar{\mathbf{w}}\|^2 = \|\mathbf{w}\|^2 + w_0^2$. The result follows. ∎

We now are prepared to use the Paley-Zygmund inequality to prove the lower bound depending on $\rho_t$ and $\zeta_t$:

*Proof of Lemma D.11.* We apply the Paley-Zygmund inequality and note that

$$\mathbb{P}_t\left(\langle \bar{\mathbf{x}}_t, \bar{\mathbf{w}}\rangle^2 \geq \frac{\mathbb{E}_t\left[\langle \bar{\mathbf{x}}_t, \mathbf{w}\rangle^2 \mid Z_{t;ij}(g) = 1\right]}{2} \mid Z_{t;ij}(g) = 1\right) \geq \frac{\mathbb{E}_t\left[\langle \bar{\mathbf{x}}_t, \mathbf{w}\rangle^2 \mid Z_{t;ij}(g) = 1\right]^2}{4\mathbb{E}_t\left[\langle \bar{\mathbf{x}}_t, \bar{\mathbf{w}}\rangle^4 \mid Z_{t;ij}(g) = 1\right]}$$

$$\geq \frac{\zeta_t(g, \bar{\mathbf{w}})^4}{4\mathbb{E}_t\left[\langle \bar{\mathbf{x}}_t, \bar{\mathbf{w}}\rangle^4 \mid Z_{t;ij}(g) = 1\right]}$$

where the second inequality follows from Lemma D.12. Applying the bound in Claim D.2, proved in a computation below for clarity, it holds that the last line above is lower bounded by:

$$\left(\frac{\mathbb{V}_{\mathbf{x}_t}[\langle \mathbf{x}_t, \mathbf{w}\rangle]}{\|\mathbf{w}\|^2 \vee \left(\mathbb{E}_t\left[\langle \mathbf{x}_t, \mathbf{w}\rangle^2 \mid Z_{t;ij}(g) = 1\right]\right)}\right)^2 \cdot \frac{1}{32\left(1 + \mathbb{E}_t\left[\langle \mathbf{x}_t, \mathbf{w}\rangle^4 \mid Z_{t;ij}(g) = 1\right]\right)}.$$

The result follows by the definition of $\rho_t(g, \bar{\mathbf{w}})$. ∎

We defer the more technical computation to the end of this section. We now have our desired small ball result, modulo the fact that we need to lower bound $\zeta_t$ and $\rho_t$. To do this, we have the following key result that lower bounds the conditional variance:

**Claim D.1.** *Suppose that Assumptions 2 and 4 holds and that*

$$\mathbb{P}\left(Z_{t;ij}(g) \mid \mathscr{F}_{t-1}\right) \geq \xi.$$

*Then it holds for any $\mathbf{w} \in \mathcal{B}^d$ that*

$$\mathbb{V}_{\mathbf{x}_t}\left[\langle \mathbf{x}_t, \mathbf{w}\rangle\right] \geq \frac{\sigma_{\mathrm{dir}}^2 \xi^2 \|\mathbf{w}\|^2}{4}.$$

*In particular, it holds that*

$$\rho_t(g, \mathbf{w}) \geq \frac{\sigma_{\mathrm{dir}}^4 \xi^4}{128 B^4}, \quad \zeta_t(g, \mathbf{w})^2 \geq \frac{\sigma_{\mathrm{dir}}^2 \xi^2}{4B^2}.$$

*Proof.* We begin by noting that by Markov's inequality, for any $\lambda > 0$, it holds that

$$\mathbb{P}_t\left(|\langle \mathbf{x}_t, \mathbf{w}\rangle - \mathbb{E}\left[\langle \mathbf{x}_t, \mathbf{w}\rangle \mid Z_{t;ij}(g) = 1\right]| > \lambda \mid Z_{t;ij}(g) = 1\right) \leq \frac{\mathbb{V}_{\mathbf{x}_t}\left[\langle \mathbf{x}_t, \mathbf{w}\rangle\right]}{\lambda^2}.$$

Rearranging, we see that

$$\mathbb{V}_{\mathbf{x}_t}\left[\langle \mathbf{x}_t, \mathbf{w}\rangle\right] \geq \sup_{\lambda > 0} \lambda^2 \cdot \mathbb{P}_t\left(|\langle \mathbf{x}_t, \mathbf{w}\rangle - \mathbb{E}_t\left[\langle \mathbf{x}_t, \mathbf{w}\rangle \mid Z_{t;ij}(g) = 1\right]| > \lambda \mid Z_{t;ij}(g) = 1\right).$$

Now we compute by Bayes' theorem,

$$\mathbb{P}_t\left(|\langle \mathbf{x}_t, \mathbf{w}\rangle - \mathbb{E}\left[\langle \mathbf{x}_t, \mathbf{w}\rangle \mid Z_{t;ij}(g) = 1\right]| \leq \lambda \mid Z_{t;ij}(g) = 1\right)$$
$$\leq \frac{\mathbb{P}_t\left(|\langle \mathbf{x}_t, \mathbf{w}\rangle - \mathbb{E}_t\left[\langle \mathbf{x}_t, \mathbf{w}\rangle \mid Z_{t;ij}(g) = 1\right]| \leq \lambda \mid\right)}{\xi}$$

and note that

$$\mathbb{P}_t\left(|\langle \mathbf{x}_t, \mathbf{w}\rangle - \mathbb{E}_t\left[\langle \mathbf{x}_t, \mathbf{w}\rangle \mid Z_{t;ij}(g) = 1\right]| \leq \lambda\right)$$
$$\leq \sup_{\mathbf{w}, c} \mathbb{P}_t\left(|\langle \mathbf{x}_t, \mathbf{w}\rangle - c| \leq \lambda\right)$$
$$\leq \sup_{\mathbf{w}, c} \mathbb{P}_t\left(\left|\left\langle \mathbf{x}_t, \frac{\mathbf{w}}{\|\mathbf{w}\|}\right\rangle - c\right| \leq \frac{\lambda}{\|\mathbf{w}\|}\right) \leq \frac{\lambda}{\|\mathbf{w}\| \sigma_{\mathrm{dir}}},$$

by Assumption 2. Thus we have

$$\mathbb{V}_{\mathbf{x}_t}\left[\langle \mathbf{x}_t, \mathbf{w}\rangle\right] \geq \sup_{\lambda > 0} \lambda^2 \left(1 - \frac{\lambda}{\xi \sigma_{\mathrm{dir}} \|\mathbf{w}\|}\right).$$

Setting $\lambda = \frac{1}{2} \cdot \xi \sigma_{\mathrm{dir}} \|\mathbf{w}\|$ concludes the proof of the first statement. The second statements follow by Assumption 4. ∎

Combining the lower bound in Claim D.1 with the small ball estimate of Lemma D.11 concludes the proof of Lemma D.10. We now prove the technical results required in previous proofs above.

**Claim D.2.** *Suppose we are in the situation of Lemma D.12. Then it holds that*

$$\mathbb{E}_t\left[\langle \bar{\mathbf{x}}_t, \mathbf{w}\rangle^4 \mid Z_{t;ij}(g) = 1\right] \leq 8 + 8\mathbb{E}_t\left[\langle \mathbf{x}_t, \mathbf{w}\rangle^4 \mid Z_{t;ij}(g) = 1\right]$$

*Proof.* Continuing to use notation from the proof of Lemma D.12, we compute:

$$\mathbb{E}\left[\langle \bar{\mathbf{x}}_t, \bar{\mathbf{w}}\rangle^4 \mid \mathscr{F}_{t-1}, Z_{t;ij}(g) = 1\right] = \mathbb{E}\left[(w_0 + \langle \mathbf{x}_t, \mathbf{w}\rangle)^4 \mid \mathscr{F}_{t-1}, Z_{t;ij}(g) = 1\right]$$
$$\leq 8w_0^4 + 8\mathbb{E}\left[\langle \mathbf{x}_t, \mathbf{w}\rangle^4 \mid \mathscr{F}_{t-1}, Z_{t;ij}(g) = 1\right].$$

The result follows by noting that $w_0^2 \leq w_0^2 + \|\mathbf{w}\|^2 = \|\bar{\mathbf{w}}\|^2 = 1$. ∎

With the covariance matrices lower bounded, we proceed to upper bound the regret.

## D.4 Upper Bounding the Empirical Error

In this section, we will show that for $1 \leq i, j \leq K$ such that $|I_{ij}(\widehat{g})|$ is sufficiently large, the fact that $\widehat{\Theta}$ is formed by minimizing the empirical risk will force

$$\mathcal{Q}_{ij}(\widehat{g}) = \sum_{t \in I_{ij}(\widehat{g})} \left|\left| \left( \widehat{\Theta}_i - \Theta_j^\star \right) \bar{\mathbf{x}}_t \right|\right|^2$$

to be small. In the end, we will combine this bound with our lower bound on $\Sigma_{ij}(\widehat{g})$ proved above to conclude the proof of the theorem. We will begin by introducing some notation. For fixed $1 \leq i, j \leq K$ and a fixed $g \in \mathcal{G}$, we define:

$$\mathcal{R}_{ij}(g) = \sum_{t \in I_{ij}(g)} \left|\left| \widehat{\Theta}_i \bar{\mathbf{x}}_t - \mathbf{y}_t \right|\right|^2 - \left|\left| \Theta_j^\star \bar{\mathbf{x}}_t - \mathbf{y}_t \right|\right|^2 .$$

The main result of this section is as follows:

**Lemma D.13.** *Under Assumptions 2-4, with probability at least $1 - \delta$, it holds that*

$$\mathcal{Q}_{ij}(\widehat{g}) \leq \varepsilon_{\mathrm{orac}} + CK^2 BRd \sqrt{TmK \log \left( \frac{TBRK}{\delta} \right)} + CK^3 d \left( 4B^2 R^2 + \nu^2 \log \left( \frac{T}{\delta} \right) \right) \log \left( \frac{BKT}{\delta} \right)$$

$$+ C\nu^2 d^2 Km \log \left( \frac{TRBmdK}{\delta} \right) + C\nu d \sqrt{Km \log \left( \frac{TRBmdK}{\delta} \right)} .$$

To prove this result, we begin by fixing $g \in \mathcal{G}$ and $1 \leq i, j \leq K$ and bounding $\mathcal{Q}_{ij}(g)$ by $\mathcal{R}_{ij}(g)$. We will then use the results of Appendix D.1 to make the statement uniform in $\mathcal{G}$. We proceed with the case of fixed $g$ and prove the following result:

**Lemma D.14.** *Suppose that Assumptions 2-4 all hold. Fix $1 \leq i, j \leq K$ and some $g \in \mathcal{G}$. For any $\lambda > 0$, with probability at least $1 - \delta$, it holds that*

$$\mathcal{Q}_{ij}(g) \leq \mathcal{R}_{ij}(g) + C\nu^2 dm \log \left( \frac{TRBmd}{\lambda \delta} \right) + C\nu \sqrt{\lambda dm \log \left( \frac{TRBmd}{\lambda \delta} \right)} + 16BR \sum_{t \in I_{ij}(g)} ||\boldsymbol{\delta}_t|| .$$

*Proof.* We begin by expanding

$$\mathcal{R}_{ij}(g) = \sum_{t \in I_{ij}(g)} \left|\left| \widehat{\Theta}_i \bar{\mathbf{x}}_t - \mathbf{y}_t \right|\right|^2 - \left|\left| \Theta_j^\star \bar{\mathbf{x}}_t - \mathbf{y}_t \right|\right|^2$$

$$= \sum_{t \in I_{ij}(g)} \left|\left| \left( \widehat{\Theta}_i - \Theta_j^\star \right) \bar{\mathbf{x}}_t - \mathbf{e}_t - \boldsymbol{\delta}_t \right|\right|^2 - ||\mathbf{e}_t + \boldsymbol{\delta}_t||^2$$

$$= \sum_{t \in I_{ij}(g)} \left|\left| \left( \widehat{\Theta}_i - \Theta_j^\star \right) \bar{\mathbf{x}}_t \right|\right|^2 - 2 \sum_{t \in I_{ij}(g)} \left\langle \mathbf{e}_t + \boldsymbol{\delta}_t, \left( \widehat{\Theta}_i - \Theta_j^\star \right) \bar{\mathbf{x}}_t \right\rangle$$

$$= \mathcal{Q}_{ij}(g) - 2 \sum_{t \in I_{ij}(g)} \left\langle \mathbf{e}_t + \boldsymbol{\delta}_t, \left( \widehat{\Theta}_i - \Theta_j^\star \right) \bar{\mathbf{x}}_t \right\rangle .$$

Thus, it suffices to bound the second term. By linearity, and the assumption on $\boldsymbol{\delta}_t$, we have

$$2 \sum_{t \in I_{ij}(g)} \left\langle \mathbf{e}_t + \boldsymbol{\delta}_t, \left( \widehat{\Theta}_i - \Theta_j^\star \right) \bar{\mathbf{x}}_t \right\rangle \leq 2 \sum_{t \in I_{ij}(g)} \left\langle \mathbf{e}_t, \left( \widehat{\Theta}_i - \Theta_j^\star \right) \bar{\mathbf{x}}_t \right\rangle + 4BR \sum_{t \in I_{ij}(g)} ||\boldsymbol{\delta}_t|| .$$

We use a generalization of the self-normalized martingale inequality, Lemma D.8 to bound the first term above. To do this, fix some $\boldsymbol{\Delta} \in \mathbb{R}^{m \times (d+1)}$ and define

$$E(\boldsymbol{\Delta}) = \sum_{t \in I_{ij}(g)} \langle \mathbf{e}_t, \boldsymbol{\Delta} \bar{\mathbf{x}}_t \rangle \qquad\qquad V(\boldsymbol{\Delta}) = \sum_{t \in I_{ij}(g)} ||\boldsymbol{\Delta} \bar{\mathbf{x}}_t||^2 .$$

Noting that $V(\widehat{\boldsymbol{\Theta}}_i - \boldsymbol{\Theta}_j^\star) = \mathcal{Q}_{ij}(g)$, we see from the above that

$$\mathcal{R}_{ij}(g) = V(\widehat{\boldsymbol{\Theta}}_i - \boldsymbol{\Theta}_j^\star) - 2E(\widehat{\boldsymbol{\Theta}}_i - \boldsymbol{\Theta}_j^\star) \tag{D.6}$$

We will now bound $E(\boldsymbol{\Delta})$ for some fixed $\boldsymbol{\Delta}$ and then apply a covering argument to lift the statement to apply to $\widehat{\boldsymbol{\Theta}}_i - \boldsymbol{\Theta}_j^\star$. By Assumption 4, we may take $\|\boldsymbol{\Delta}\| \le 2R$.

Recall that $\mathscr{F}_t^y$ denotes the filtration generated by $\mathbf{x}_1, \ldots, \mathbf{x}_t, \mathbf{y}_1, \ldots, \mathbf{y}_{t-1}$. Applying Simchowitz et al. [55, Lemma E.1], a generalization of Lemma D.8, with $e_t \leftarrow \langle \mathbf{e}_t, \boldsymbol{\Delta}\bar{\mathbf{x}}_t \rangle$, $u_t = \boldsymbol{\Delta}\bar{\mathbf{x}}_t$, and the filtration $\mathscr{F}_t^y$, tells us that for any $\lambda > 0$, it holds with probability at least $1 - \delta$ that

$$E(\boldsymbol{\Delta})^2 \le 2\nu^2(\lambda + V(\boldsymbol{\Delta}))\log\left(\frac{\sqrt{1 + \frac{V(\boldsymbol{\Delta})}{\lambda}}}{\delta}\right). \tag{D.7}$$

We note by Assumption 4, it holds that

$$V(\boldsymbol{\Delta}) \le 4TR^2B^2$$

and thus the subadditivity of the square root tells us that

$$E(\boldsymbol{\Delta})^2 \le 2\nu^2(\lambda + V(\boldsymbol{\Delta}))\log\left(\frac{1 + 2RB\sqrt{\frac{T}{\lambda}}}{\delta}\right)$$

with probability at least $1 - \delta$. Now let $\mathscr{N}$ be an $\varepsilon$-net of the Frobenius ball in $\mathbb{R}^{m \times (d+1)}$ of radius $2R$. For small $\varepsilon$, we may take $\mathscr{N}$ such that

$$\log(|\mathscr{N}|) \le m(d+1)\log\left(\frac{6R}{\varepsilon}\right). \tag{D.8}$$

For any $\boldsymbol{\Delta}$, denote by $\boldsymbol{\Delta}'$ its projection into $\mathscr{N}$. Then we compute
$E(\boldsymbol{\Delta})^2 \le E(\boldsymbol{\Delta}')^2 + E(\boldsymbol{\Delta})^2 - E(\boldsymbol{\Delta}')^2$

$$\le \left(\sum_{t \in I_{ij}(g)} \|\mathbf{e}_t\|_{\mathrm{F}}^2\right) 4RB^2T\sqrt{\|\boldsymbol{\Delta} - \boldsymbol{\Delta}'\|_{\mathrm{F}}} + 2\nu^2\left(\lambda + V(\boldsymbol{\Delta}')\right)\log\left(\frac{1 + 2RB\sqrt{\frac{T}{\lambda}}}{\delta}|\mathscr{N}|\right)$$

$$\le \left(\sum_{t \in I_{ij}(g)} \|\mathbf{e}_t\|^2\right) 4RB^2T\sqrt{\varepsilon} + 2\nu^2\left(\lambda + V(\boldsymbol{\Delta}')\right)\log\left(\frac{1 + 2RB\sqrt{\frac{T}{\lambda}}}{\delta}|\mathscr{N}|\right)$$

$$\le \left(\sum_{t \in I_{ij}(g)} \|\mathbf{e}_t\|^2\right) 4RB^2T\sqrt{\varepsilon} + 2\nu^2\left(\lambda + V(\boldsymbol{\Delta})\right)\log\left(\frac{1 + 2RB\sqrt{\frac{T}{\lambda}}}{\delta}|\mathscr{N}|\right)$$

$$+ (V(\boldsymbol{\Delta}') - V(\boldsymbol{\Delta}))\log\left(\frac{1 + 2RB\sqrt{\frac{T}{\lambda}}}{\delta}\right)$$

where the second inequality follows from Claim D.3 along with a union bound over $\mathscr{N}$ applied to (D.7), the third inequality follows from the definition of an $\varepsilon$-net, and the last inequality follows from simply adding and subtracting the same term. Applying Claim D.3 once again, we have with probability at least $1 - \delta$,

$$E(\boldsymbol{\Delta})^2 \le \left(\sum_{t \in I_{ij}(g)} \|\mathbf{e}_t\|^2\right) 4RB^2T\sqrt{\varepsilon} + 2\nu^2\left(\lambda + V(\boldsymbol{\Delta})\right)\log\left(\frac{1 + 2RB\sqrt{\frac{T}{\lambda}}}{\delta}|\mathscr{N}|\right)$$

$$+ 4RB^2T\sqrt{\varepsilon}\log\left(\frac{1 + 2RB\sqrt{\frac{T}{\lambda}}}{\delta}|\mathscr{N}|\right)$$

$$\le 2\nu^2\left(\lambda + V(\boldsymbol{\Delta})\right)\log\left(\frac{1 + 2RB\sqrt{\frac{T}{\lambda}}}{\delta}|\mathscr{N}|\right) + 4RB^2T^2m\nu^2\sqrt{\varepsilon}\log\left(\frac{1 + 2RB\sqrt{\frac{T}{\lambda}}}{\delta}|\mathscr{N}|\right)$$

uniformly for all $\boldsymbol{\Delta}$ in the Frobenius ball of radius $2R$. We now choose

$$\varepsilon = C \left( \frac{TRB^2 md}{\lambda} \right)^{-c}$$

for some universal constants $C, c$ to ensure that

$$4RB^2T^2m\nu^2\sqrt{\varepsilon}\log\left( \frac{1 + 2RB\sqrt{\frac{T}{\lambda}}}{\delta} |\mathcal{N}| \right) \le 2\nu^2\lambda$$

Thus with probability at least $1 - \delta$, it holds for all $\boldsymbol{\Delta}$ that

$$E(\boldsymbol{\Delta})^2 \le 2\nu^2(\lambda + V(\boldsymbol{\Delta})) \left( \log\left( \frac{1 + 2RB\sqrt{\frac{T}{\lambda}}}{\delta} \right) + m(d+1)\log\left( \frac{TR^2B^2md}{\lambda} \right) \right)$$

where we applied (D.8) to bound the size of $\mathcal{N}$. In particular, this holds for $\boldsymbol{\Delta} = \widehat{\boldsymbol{\Theta}}_i - \boldsymbol{\Theta}_j^\star$ and thus

$$E(\widehat{\boldsymbol{\Theta}}_i - \boldsymbol{\Theta}_j^\star)^2 \le 2\nu^2(\lambda + \mathcal{Q}_{ij}(g)) \left( \log\left( \frac{1 + 2RB\sqrt{\frac{T}{\lambda}}}{\delta} \right) + m(d+1)\log\left( \frac{TR^2B^2md}{\lambda} \right) \right)$$

Plugging this back into (D.6), we have that with probability at least $1 - \delta$, it holds that

$$4BR|I_{ij}(g)|\varepsilon_{\mathrm{crp}} + \mathcal{R}_{ij}(g) \ge \mathcal{Q}_{ij}(g) -$$
$$- C\nu\sqrt{(\lambda + \mathcal{Q}_{ij}(g)) \left( \log\left( \frac{1 + 2RB\sqrt{\frac{T}{\lambda}}}{\delta} \right) + m(d+1)\log\left( \frac{TR^2B^2md}{\lambda} \right) \right)}.$$

Rearranging concludes the proof. ∎

We defer proofs of the technical computations, Lemmas D.3 and D.4, used in the proof of Lemma D.14 until the end of the section. For now, we press on to lift our bound from a statement about a fixed $g \in \mathcal{G}$ to one uniform in $\mathcal{G}$. We require one last lemma before concluding this proof, however. We are aiming to bound $\mathcal{Q}_{ij}(\widehat{g})$ by $\mathcal{R}_{ij}(\widehat{g})$, but we need to upper bound $\mathcal{R}_{ij}(\widehat{g})$. This is the content of the following lemma:

**Lemma D.15.** *With probability at least $1 - \delta$, for all $1 \le i, j \le K$, it holds that*

$$\mathcal{R}_{ij}(\widehat{g}) \le \varepsilon_{\mathrm{orac}} + 4BRK^2 \sum_{t \in I_{ij}(g)} ||\boldsymbol{\delta}_t|| + CK^2BR\sqrt{Tmd\log\left( \frac{TBRK\mathrm{DN}(\varepsilon)}{\delta} \right)}$$
$$+ K^2 \left( 4B^2R^2 + \nu^2\log\left( \frac{T}{\delta} \right) \right) \left( 2T\varepsilon + 6\log\left( \frac{\mathrm{DN}(\varepsilon)}{\delta} \right) \right)$$

*Proof.* We begin by introducing some notation. For fixed $\boldsymbol{\Theta} \in \mathbb{R}^{m \times (d+1)}$ and $g \in \mathcal{G}$, let

$$\mathcal{R}_{ij}(g, \boldsymbol{\Theta}) = \sum_{t \in I_{ij}(g)} ||\boldsymbol{\Theta}_i\bar{\mathbf{x}}_t - \mathbf{y}_t||^2 - ||\mathbf{e}_t + \boldsymbol{\delta}_t||^2$$

and note that $\mathcal{R}_{ij}(g) = \mathcal{R}_{ij}(g, \widehat{\boldsymbol{\Theta}}_i)$. First, we note that by definition of ERMORACLE,

$$\varepsilon_{\mathrm{orac}} \ge \sum_{t=1}^{T} \left\|\widehat{\boldsymbol{\Theta}}_{\widehat{g}(\bar{\mathbf{x}}_t)}\bar{\mathbf{x}}_t - \mathbf{y}_t\right\|^2 - \inf_{g, \widehat{\boldsymbol{\Theta}}} \sum_{t=1}^{T} \left\|\widehat{\boldsymbol{\Theta}}_{\widehat{g}(\bar{\mathbf{x}}_t)}\bar{\mathbf{x}}_t - \mathbf{y}_t\right\|^2$$
$$\ge \sum_{t=1}^{T} \left\|\widehat{\boldsymbol{\Theta}}_{\widehat{g}(\bar{\mathbf{x}}_t)}\bar{\mathbf{x}}_t - \mathbf{y}_t\right\|^2 - \left\|\boldsymbol{\Theta}_{g_\star(\bar{\mathbf{x}}_t)}^\star\bar{\mathbf{x}}_t - \mathbf{y}_t\right\|^2$$
$$= \sum_{1 \le i, j \le K} \mathcal{R}_{ij}(\widehat{g}).$$

Thus, we have

$$\mathcal{R}_{ij}(\widehat{g}) \leq \varepsilon_{\text{orac}} - \sum_{\substack{1 \leq i',j' \leq K \\ (i,j) \neq (i',j')}} \mathcal{R}_{ij}(\widehat{g}). \tag{D.9}$$

Thus it will suffice to provide a lower bound on $\mathcal{R}_{ij}(\widehat{g})$ that holds with high probability. To do this, note that for fixed $g \in \mathcal{G}$ and $\boldsymbol{\Theta}_0$, we have

$$\mathcal{R}_{ij}(g, \boldsymbol{\Theta}_0) = \sum_{t \in I_{ij}(g)} \left|\left|\left(\boldsymbol{\Theta}_0 - \boldsymbol{\Theta}_j^\star\right)\bar{\mathbf{x}}_t\right|\right|^2 - 2\sum_{t \in I_{ij}(g)} \left\langle \mathbf{e}_t + \boldsymbol{\delta}_t, \left(\boldsymbol{\Theta}_0 - \boldsymbol{\Theta}_j^\star\right)\bar{\mathbf{x}}_t - \mathbf{e}_t - \boldsymbol{\delta}_t \right\rangle$$

$$\geq -2\sum_{t \in I_{ij}(g)} \left\langle \mathbf{e}_t + \boldsymbol{\delta}_t, \left(\boldsymbol{\Theta}_0 - \boldsymbol{\Theta}_j^\star\right)\bar{\mathbf{x}}_t \right\rangle$$

$$\geq -2\sum_{t \in I_{ij}(g)} \left\langle \mathbf{e}_t, (\boldsymbol{\Theta}_0 - \boldsymbol{\Theta}_j^\star)\bar{\mathbf{x}}_t \right\rangle - 4BR\sum_{t \in I_{ij}(g)} ||\boldsymbol{\delta}_t||.$$

Noting that $\mathbf{e}_t$ is subGaussian by Assumption 3 and by Assumption 4 we have control over $\left|\left|\left(\boldsymbol{\Theta}_0 - \boldsymbol{\Theta}_j^\star\right)\bar{\mathbf{x}}_t\right|\right|$, we see that with probability at least $1 - \delta$, it holds that

$$\mathcal{R}_{ij}(g, \boldsymbol{\Theta}_0) \geq -2\sum_{t \in I_{ij}(g)} \left\langle \mathbf{e}_t, \left(\boldsymbol{\Theta}_0 - \boldsymbol{\Theta}_j^\star\right)\bar{\mathbf{x}}_t \right\rangle - 4BR\sum_{t \in I_{ij}(g)} ||\boldsymbol{\delta}_t||$$

$$\geq -8BR\sqrt{|I_{ij}(g)| \cdot \log\left(\frac{1}{\delta}\right)} - 4BR\sum_{t \in I_{ij}(g)} ||\boldsymbol{\delta}_t||.$$

Now note that

$$\mathcal{R}_{ij}(g, \boldsymbol{\Theta}_0) - \mathcal{R}_{ij}(g, \boldsymbol{\Theta}_0') = \sum_{t \in I_{ij}(g)} \left|\left|\left(\boldsymbol{\Theta}_0 - \boldsymbol{\Theta}_j^\star\right)\bar{\mathbf{x}}_t - \mathbf{e}_t - \boldsymbol{\delta}_t\right|\right|^2 - \left|\left|\left(\boldsymbol{\Theta}_0' - \boldsymbol{\Theta}_j^\star\right)\bar{\mathbf{x}}_t - \mathbf{e}_t - \boldsymbol{\delta}_t\right|\right|^2$$

$$= \sum_{t \in I_{ij}(g)} \left\langle \left(\boldsymbol{\Theta}_0 + \boldsymbol{\Theta}_0' - 2\boldsymbol{\Theta}_j^\star\right)\bar{\mathbf{x}}_t - 2\mathbf{e}_t - 2\boldsymbol{\delta}_t, \left(\boldsymbol{\Theta}_0 - \boldsymbol{\Theta}_0'\right)\bar{\mathbf{x}}_t \right\rangle$$

Applying Cauchy Schwarz and noting that with probability at least $1 - \delta$ it holds that

$$\max_{1 \leq t \leq T} ||\mathbf{e}_t|| \leq \nu\sqrt{\log\left(\frac{T}{\delta}\right)},$$

we have that with probability at least $1 - \delta$,

$$\mathcal{R}_{ij}(g, \boldsymbol{\Theta}_0) - \mathcal{R}_{ij}(g, \boldsymbol{\Theta}_0') \leq 4RB^2 |I_{ij}(g)| \, ||\boldsymbol{\Theta}_0 - \boldsymbol{\Theta}_0'|| + 4BR\sum_{t \in I_{ij}(g)} ||\boldsymbol{\delta}_t||.$$

Thus if we take a union bound over a $\frac{1}{TRB^2}$-net of the Frobenius ball of radius $R$ in $\mathbb{R}^{m \times (d+1)}$, we see that with probability at least $1 - \delta$, it holds that

$$\mathcal{R}_{ij}(g) = \mathcal{R}_{ij}(g, \widehat{\boldsymbol{\Theta}}_i) \geq -CBR\sqrt{Tm(d+1)\log\left(\frac{TBR}{\delta}\right)} - 4BR\sum_{t \in I_{ij}(g)} ||\boldsymbol{\delta}_t||.$$

We now note that by Lemma D.1, if $\mathcal{D}$ is an $\varepsilon$-disagreement cover of $\mathcal{G}$, then

$$\sup_{g \in \mathcal{G}} \min_{g_k \in \mathcal{D}} \sum_{t=1}^{T} \mathbb{I}\left[g(\bar{\mathbf{x}}_t) \neq g_k(\bar{\mathbf{x}}_t)\right] \leq 2T\varepsilon + 6\log\left(\frac{\text{DN}(\varepsilon)}{\delta}\right).$$

We now apply Claim D.5 and note that this implies that if $g$ is the projection of $\widehat{g}$ onto the disagreement cover, then

$$\mathcal{R}_{ij}(\widehat{g}) \geq \mathcal{R}_{ij}(g) - \left(4B^2R^2 + \nu^2\log\left(\frac{T}{\delta}\right)\right)\left(2T\varepsilon + 6\log\left(\frac{\text{DN}(\varepsilon)}{\delta}\right)\right)$$

Taking a union bound over $g \in \mathcal{D}$ and all pairs $(i,j)$ tells us that with probability at least $1 - \delta$,

$$\mathcal{R}_{ij}(\widehat{g}) \geq -CBR\sqrt{Tmd\log\left(\frac{TBRK\text{DN}(\varepsilon)}{\delta}\right)} - 4BR\sum_{t \in I_{ij}(g)} ||\boldsymbol{\delta}_t|| - \left(4B^2R^2 + \nu^2\log\left(\frac{T}{\delta}\right)\right)\left(2T\varepsilon + 6\log\left(\frac{\text{DN}(\varepsilon)}{\delta}\right)\right)$$

Combining this with (D.9) concludes the proof. ∎

We are now finally ready to prove the main result of this section:

*Proof of Lemma D.13.* We begin by noting that Lemma D.1 tells us that with probability at least $1 - \delta$, it holds that

$$\sup_{g \in \mathcal{G}} \min_{g_k \in \mathcal{D}} \sum_{t=1}^{T} \mathbb{I}[g(\bar{\mathbf{x}}_t) \neq g_k(\bar{\mathbf{x}}_t)] \leq 2T\varepsilon + 6\log\left(\frac{\mathrm{DN}(\varepsilon)}{\delta}\right)$$

We observe that for $g, g' \in \mathcal{G}$, we have

$$\mathcal{Q}_{ij}(g) - \mathcal{Q}_{ij}(g') = \sum_{t \in I_{ij}(g)} \left\|(\widehat{\boldsymbol{\Theta}}_i - \boldsymbol{\Theta}_j^\star)\bar{\mathbf{x}}_t\right\|^2 - \sum_{t \in I_{ij}(g')} \left\|(\widehat{\boldsymbol{\Theta}}_i - \boldsymbol{\Theta}_j^\star)\bar{\mathbf{x}}_t\right\|^2$$

$$\leq 4R^2 B^2 \sum_t |Z_{t;ij}(g) - Z_{t;ij}(g')|$$

$$\leq 4R^2 B^2 \sum_{t=1}^{T} \mathbb{I}\left[g(\bar{\mathbf{x}}_t) \neq g'(\bar{\mathbf{x}}_t)\right].$$

By Claim D.5, it holds that with probability at least $1 - \delta$, for all $g, g' \in \mathcal{G}$, we have

$$\mathcal{R}_{ij}(g) - \mathcal{R}_{ij}(g') \leq \left(4B^2 R^2 + \nu^2 \log\left(\frac{T}{\delta}\right)\right) \sum_{t=1}^{T} \mathbb{I}\left[g(\bar{\mathbf{x}}_t) \neq g'(\bar{\mathbf{x}}_t)\right]$$

Now, taking a union bound over a minimal disagreement cover at scale $\varepsilon$ as well as all pairs $1 \leq i, j \leq K$, we see that Lemma D.14 implies that with probability at least $1 - \delta$, it holds that for all $g_k \in \mathcal{D}$ the disagreement cover,

$$\mathcal{Q}_{ij}(g_k) \leq \mathcal{R}_{ij}(g_k) + C\nu^2 dm \log\left(\frac{TRBmdK^2\mathrm{DN}(\varepsilon)}{\lambda\delta}\right) + C\nu\sqrt{\lambda dm \log\left(\frac{TRBmdK^2\mathrm{DN}(\varepsilon)}{\lambda\delta}\right)}.$$

Letting $g$ denote the projection of $\widehat{g}$ into the disagreement cover, we compute:

$$\mathcal{Q}_{ij}(\widehat{g}) = \mathcal{Q}_{ij}(\widehat{g}) - \mathcal{Q}_{ij}(g) + \mathcal{Q}_{ij}(g)$$

$$\leq 4R^2 B^2 \sum_{t=1}^{T} \mathbb{I}\left[\widehat{g}(\bar{\mathbf{x}}_t) \neq g(\bar{\mathbf{x}}_t)\right] + \mathcal{Q}_{ij}(g)$$

$$\leq 4R^2 B^2 \sum_{t=1}^{T} \mathbb{I}\left[\widehat{g}(\bar{\mathbf{x}}_t) \neq g(\bar{\mathbf{x}}_t)\right] + \mathcal{R}_{ij}(g) + C\nu^2 dm \log\left(\frac{TRBmdK^2\mathrm{DN}(\varepsilon)}{\lambda\delta}\right) + C\nu\sqrt{\lambda dm \log\left(\frac{TRBmdK^2\mathrm{DN}(\varepsilon)}{\lambda\delta}\right)}$$

$$\leq \left(8B^2 R^2 + \nu^2 \log\left(\frac{T}{\delta}\right)\right)\left(2T\varepsilon + 6\log\left(\frac{\mathrm{DN}(\varepsilon)}{\delta}\right)\right) + \mathcal{R}_{ij}(\widehat{g})$$

$$+ C\nu^2 dm \log\left(\frac{TRBmdK^2\mathrm{DN}(\varepsilon)}{\lambda\delta}\right) + C\nu\sqrt{\lambda dm \log\left(\frac{TRBmdK^2\mathrm{DN}(\varepsilon)}{\lambda\delta}\right)}$$

Setting $\lambda = 1$ and applying Lemma D.15 to bound $\mathcal{R}_{ij}(\widehat{g})$ then tells us that

$$\mathcal{Q}_{ij}(\widehat{g}) \leq \varepsilon_{\mathrm{orac}} + CK^2 BR\sqrt{Tmd \log\left(\frac{TBRK\mathrm{DN}(\varepsilon)}{\delta}\right)} + K^2\left(4B^2 R^2 + \nu^2 \log\left(\frac{T}{\delta}\right)\right)\left(2T\varepsilon + 6\log\left(\frac{\mathrm{DN}(\varepsilon)}{\delta}\right)\right)$$

$$+ C\nu^2 dm \log\left(\frac{TRBmdK^2\mathrm{DN}(\varepsilon)}{\delta}\right) + C\nu\sqrt{dm \log\left(\frac{TRBmdK^2\mathrm{DN}(\varepsilon)}{\delta}\right)} + 4BRK^2 T\varepsilon_{\mathrm{crp}}.$$

We can now plug in our bound on $\mathrm{DN}(\varepsilon)$ from Lemma D.3 which says that

$$\log\left(\mathrm{DN}(\mathcal{G}, \varepsilon)\right) \leq 2Kd \log\left(\frac{3BK}{\varepsilon}\right)$$

and set $\varepsilon = \frac{1}{T}$ to conclude the proof. ∎

With the proof of Lemma D.13 concluded, we now prove the technical lemmas:

**Claim D.3.** *Let $\boldsymbol{\Delta}, \boldsymbol{\Delta}' \in \mathbb{R}^{m \times (d+1)}$ have Frobenius norm at most $R$. Let $E, V$ be defined as in the proof of Lemma D.14. Then it holds that*

$$E(\boldsymbol{\Delta})^2 - E(\boldsymbol{\Delta}')^2 \leq \left( \sum_{t \in I_{ij}(g)} ||\mathbf{e}_t||^2 \right) 4RB^2 T \sqrt{||\boldsymbol{\Delta} - \boldsymbol{\Delta}'||_{\mathrm{F}}}$$

$$V(\boldsymbol{\Delta}') - V(\boldsymbol{\Delta}) \leq 4RB^2 T \sqrt{\boldsymbol{\Delta} - \boldsymbol{\Delta}'}$$

*Proof.* We compute:

$$
\begin{aligned}
E(\boldsymbol{\Delta})^2 - E(\boldsymbol{\Delta}')^2 &= E(\boldsymbol{\Delta} + \boldsymbol{\Delta}') \cdot E(\boldsymbol{\Delta} - \boldsymbol{\Delta}') \\
&= \left( \sum_{t \in I_{ij}(g)} \langle \mathbf{e}_t, (\boldsymbol{\Delta} + \boldsymbol{\Delta}')\mathbf{x}_t \rangle \right) \cdot \left( \sum_{t \in I_{ij}(g)} \langle \mathbf{e}_t, (\boldsymbol{\Delta} - \boldsymbol{\Delta}')\mathbf{x}_t \rangle \right) \\
&\leq \left( \sum_{t \in I_{ij}(g)} ||\mathbf{e}_t||^2 \right) \sqrt{V(\boldsymbol{\Delta} + \boldsymbol{\Delta}')} \cdot \sqrt{V(\boldsymbol{\Delta} - \boldsymbol{\Delta}')} \\
&\leq \left( \sum_{t \in I_{ij}(g)} ||\mathbf{e}_t||^2 \right) \sqrt{16TR^2B^2} \cdot \sqrt{V(\boldsymbol{\Delta} - \boldsymbol{\Delta}')},
\end{aligned}
$$

where the first equality follows from linearity, the first inequality follows by Cauchy-Schwartz, and the last follows by a Assumption 4. Similarly,

$$
\begin{aligned}
V(\boldsymbol{\Delta}) - V(\boldsymbol{\Delta}') &= \sum_{t \in I_{ij}(g)} \left( ||\boldsymbol{\Delta}\bar{\mathbf{x}}_t|| + ||\boldsymbol{\Delta}'\bar{\mathbf{x}}_t|| \right) \left( ||\boldsymbol{\Delta}\bar{\mathbf{x}}_t|| - ||\boldsymbol{\Delta}'\bar{\mathbf{x}}_t|| \right) \\
&\leq \sum_{t \in I_{ij}(g)} (4RB) \, ||(\boldsymbol{\Delta} - \boldsymbol{\Delta}')\bar{\mathbf{x}}_t|| \\
&\leq 4RB\sqrt{T} \cdot \sqrt{V(\boldsymbol{\Delta} - \boldsymbol{\Delta}')}.
\end{aligned}
$$

We now note that

$$
\begin{aligned}
V(\boldsymbol{\Delta} - \boldsymbol{\Delta}') &= \sum_{t \in I_{ij}(g)} ||(\boldsymbol{\Delta} - \boldsymbol{\Delta}')\bar{\mathbf{x}}_t||^2 \\
&\leq \sum_{t \in I_{ij}(g)} ||\boldsymbol{\Delta} - \boldsymbol{\Delta}'||_{\mathrm{F}}^2 \, ||\bar{\mathbf{x}}_t||^2 \\
&\leq TB^2 ||\boldsymbol{\Delta} - \boldsymbol{\Delta}'||_{\mathrm{F}}.
\end{aligned}
$$

The result follows. ∎

**Claim D.4.** *Suppose that Assumption 3 holds. Then with probability at least $1 - \delta$, it holds that*

$$\sum_{t=1}^{T} ||\mathbf{e}_t||^2 \leq 2\nu^2 \left( Tm \log(8) + \log\left(\frac{1}{\delta}\right) \right).$$

*Proof.* Denote by $\bar{\mathbf{e}} = (\mathbf{e}_1, \ldots, \mathbf{e}_T)$ the concatenation of all the $\mathbf{e}_t$ and note that $\bar{\mathbf{e}}$ is subGaussian by Assumption 3. Let $\bar{\mathbf{u}}$ denote a unit vector in $\mathbb{R}^{Tm}$ and note that for any fixed $\bar{\mathbf{u}} \in \mathbb{S}^{Tm-1}$, it holds with probability at least $1 - \delta$ that $\langle \bar{\mathbf{e}}, \bar{\mathbf{u}} \rangle \leq \sqrt{2\nu^2 \log\left(\frac{1}{\delta}\right)}$. Constructing a covering at scale $\frac{1}{4}$ of $\mathbb{S}^{Tm-1}$ and applying the standard discretization argument, we see that

$$\sum_{t=1}^{T} ||\mathbf{e}_t||^2 = ||\bar{\mathbf{e}}||^2 = \left( \sup_{\bar{\mathbf{u}} \in \mathbb{S}^{Tm-1}} \langle \bar{\mathbf{e}}, \bar{\mathbf{u}} \rangle \right)^2 \leq 2\nu^2 \left( Tm \log(8) + \log\left(\frac{1}{\delta}\right) \right)$$

with probability at least $1 - \delta$. ∎

**Claim D.5.** *Suppose that Assumptions 3 and 4 hold. Then, with probability at least $1 - \delta$, it holds for all $g, g'$,*

$$\mathcal{R}_{ij}(g) - \mathcal{R}_{ij}(g') \leq \left( 4B^2 R^2 + \nu^2 \log\left(\frac{T}{\delta}\right) \right) \sum_{t=1}^{T} \mathbb{I}\left[ g(\bar{\mathbf{x}}_t) \neq g'(\bar{\mathbf{x}}_t) \right].$$

*Proof.* We compute:

$$\mathcal{R}_{ij}(g) - \mathcal{R}_{ij}(g') = \sum_{t \in I_{ij}(g)} \left\| \widehat{\boldsymbol{\Theta}}_i \bar{\mathbf{x}}_t - \mathbf{y}_t \right\|^2 - \|\mathbf{e}_t\|^2 - \sum_{t \in I_{ij}(g')} \left\| \widehat{\boldsymbol{\Theta}}_i \bar{\mathbf{x}}_t - \mathbf{y}_t \right\|^2 - \|\mathbf{e}_t\|^2.$$

A union bound and Assumption 3 tell us that with probability at least $1 - \delta$,

$$\max_{1 \leq t \leq T} \|\mathbf{e}_t\| \leq \nu \sqrt{\log\left(\frac{T}{\delta}\right)}.$$

As $\|\widehat{\boldsymbol{\Theta}}_i - \boldsymbol{\Theta}_j^\star\|_{\mathrm{F}} \leq 2R$, the result follows. ∎

## D.5 Proof of Theorem 11

We are now ready to put everything together and conclude the proof of Theorem 11. We first need to prove the following bound by combining results from Appendices D.2 and D.3:

**Lemma D.16.** *For any fixed $\xi < 1$, Define the following notation:*

$$\Xi_1 = C \frac{B^8}{\sigma_{\mathrm{dir}}^8 \xi^8} \left( \log\left( \frac{2T}{\delta} + \frac{d}{2} \log\left( \frac{B}{\sigma_{\mathrm{dir}}\xi} \right) + \log\left( K^2 \mathrm{DN}(\varepsilon) \right) \right) \right)$$

$$\Xi_2 = C \frac{B^2}{\sigma_{\mathrm{dir}}^2 \xi^2} \left( 2T\varepsilon + 6 \log\left( \frac{\mathrm{DN}(\varepsilon)}{\delta} \right) \right)$$

$$\Xi_3 = 4K^2 T\xi + 12 \log\left( \frac{1}{\delta} \right) + \max\left( \Xi_1, \Xi_2 \right)$$

*With probability at least $1 - \delta$, it holds that for all $1 \leq i, j \leq K$ such that*

$$|I_{ij}(\widehat{g})| \geq \Xi_3. \tag{D.10}$$

*we have*

$$\|\widehat{\boldsymbol{\Theta}}_i - \boldsymbol{\Theta}_j^\star\|_{\mathrm{F}}^2 \leq C \frac{B^2}{\sigma_{\mathrm{dir}}^2 \xi^2 |I_{ij}(\widehat{g})|} \cdot \mathcal{Q}_{ij}(\widehat{g}). \tag{D.11}$$

*Proof.* We begin by applying a Chernoff bound (Lemma D.2) to bound the number of times $t$ such that $Z_{t;ij}(\widehat{g}) = 1$ and $\mathbb{P}_t(Z_{t;ij}(\widehat{g}) = 1) \leq \xi$. In particular, we have

$$\mathbb{P}\left( \sum_{t=1}^{T} \mathbb{I}\left[ \text{there exist } i, j \text{ such that } Z_{t;ij}(\widehat{g}) = 1 \text{ and } \mathbb{P}_t(Z_{t;ij}(\widehat{g})) \leq \xi \right] \geq 2TK^2\xi + 6\log\left( \frac{1}{\delta} \right) \right)$$

$$= \mathbb{P}\left( \sum_{t=1}^{T} \sum_{1 \leq i,j \leq K} \mathbb{I}\left[ Z_{t;ij}(\widehat{g}) = 1 \text{ and } \mathbb{P}_t(Z_{t;ij}(\widehat{g})) \leq \xi \right] \geq 2TK^2\xi + 6\log\left( \frac{1}{\delta} \right) \right)$$

$$\leq \delta$$

where the last step follows by Lemma D.2. Now, denote

$$\widetilde{I}_{ij}(\widehat{g}) = I_{ij}(\widehat{g}) \cap \{t | \mathbb{P}_t(Z_{t;ij}(\widehat{g})) \geq \xi\}$$

$$\widetilde{\Sigma}_{ij}(g) = \sum_{t \in \widetilde{I}_{ij}(\widehat{g})} \bar{\mathbf{x}}_t \bar{\mathbf{x}}_t^T$$

and note that $\Sigma_{ij}(\widehat{g}) \succeq \widetilde{\Sigma}_{ij}(\widehat{g})$ by construction. Furthermore, by the Chernoff bound above, with probability at least $1 - \delta$, if

$$|I_{ij}(\widehat{g})| \geq 4K^2 T\xi + 12 \log\left(\frac{1}{\delta}\right) + \max\left(\Xi_1, \Xi_2\right),$$

it holds that

$$\left|\widetilde{I}_{ij}(\widehat{g})\right| \geq \max\left(\frac{1}{2}\left|I_{ij}(\widehat{g})\right|, \max\left(\Xi_1, \Xi_2\right)\right).$$

Now, applying Lemmas D.6 and D.10, we see that with probability at least $1 - \delta$,

$$\Sigma_{ij}(\widehat{g}) \succeq \widetilde{\Sigma}_{ij}(\widehat{g}) \succeq c\frac{\sigma_{\mathrm{dir}}^2 \xi^2}{B^2}\left|\widetilde{I}_{ij}(\widehat{g})\right| I \succeq c\frac{\sigma_{\mathrm{dir}}^2 \xi^2}{B^2}\left|I_{ij}(\widehat{g})\right|.$$

We now compute:

$$
\begin{aligned}
\mathcal{Q}_{ij}(\widehat{g}) &= \sum_{t \in I_{ij}(\widehat{g})} \left\|\left(\widehat{\boldsymbol{\Theta}}_i - \boldsymbol{\Theta}_j^\star\right)\bar{\mathbf{x}}_t\right\|^2 \\
&= \sum_{t \in I_{ij}(\widehat{g})} \mathrm{Tr}\left(\bar{\mathbf{x}}_t^T \left(\widehat{\boldsymbol{\Theta}}_i - \boldsymbol{\Theta}_j^\star\right)^T \left(\widehat{\boldsymbol{\Theta}}_i - \boldsymbol{\Theta}_j^\star\right)\bar{\mathbf{x}}_t\right) \\
&= \sum_{t \in I_{ij}(\widehat{g})} \mathrm{Tr}\left(\left(\widehat{\boldsymbol{\Theta}}_i - \boldsymbol{\Theta}_j^\star\right)^T \left(\widehat{\boldsymbol{\Theta}}_i - \boldsymbol{\Theta}_j^\star\right)\bar{\mathbf{x}}_t \bar{\mathbf{x}}_t^T\right) \\
&\geq \|\widehat{\boldsymbol{\Theta}}_i - \boldsymbol{\Theta}_j^\star\|_{\mathrm{F}}^2 \cdot \lambda_{min}\left(\sum_{t \in I_{ij}(\widehat{g})} \bar{\mathbf{x}}_t \bar{\mathbf{x}}_t^T\right) \\
&= \|\widehat{\boldsymbol{\Theta}}_i - \boldsymbol{\Theta}_j^\star\|_{\mathrm{F}}^2 \cdot \lambda_{min}\left(\Sigma_{ij}(\widehat{g})\right)
\end{aligned}
$$

where we have denoted by $\lambda_{min}(\cdot)$ the minimal eigenvalue of a symmetric matrix. Thus, under our assumptions, with probability at least $1 - \delta$, it holds that

$$\|\widehat{\boldsymbol{\Theta}}_i - \boldsymbol{\Theta}_j^\star\|_{\mathrm{F}} \leq C\frac{B^2}{\sigma_{\mathrm{dir}}^2 \xi^2 |I_{ij}(\widehat{g})|} \cdot \mathcal{Q}_{ij}(\widehat{g})$$

for all $i, j$ satisfying (D.10). The result follows. ∎

We are now ready to combine Lemma D.16 with the results of Appendix D.4 to conclude the proof.

*Proof of Theorem 11.* By Lemma D.13, it holds with probability at least $1 - \delta$ that

$$
\begin{aligned}
\mathcal{Q}_{ij}(\widehat{g}) &\leq \varepsilon_{\mathrm{orac}} + CK^2 BRd\sqrt{TmK \log\left(\frac{TBRK}{\delta}\right)} + CK^3 d\left(4B^2 R^2 + \nu^2 \log\left(\frac{T}{\delta}\right)\right)\log\left(\frac{BKT}{\delta}\right) \\
&\quad + C\nu^2 d^2 Km \log\left(\frac{TRBmdK}{\delta}\right) + C\nu d\sqrt{Km \log\left(\frac{TRBmdK}{\delta}\right)} + 4BRK^2 |I_{ij}(g)| \varepsilon_{\mathrm{crp}} \\
&\leq \varepsilon_{\mathrm{orac}} + 1 + CK^3 B^2 R^2 d^2 m\nu^2 \sqrt{T} \log\left(\frac{TRBmdK}{\delta}\right) + 4BRK^2 |I_{ij}(g)| \varepsilon_{\mathrm{crp}}.
\end{aligned}
$$

By Lemma D.16, it holds with probability at least $1 - \delta$ that for all $i, j$ satisfying $|I_{ij}(\widehat{g})| \geq \Xi_3$, we have

$$\|\widehat{\boldsymbol{\Theta}}_i - \boldsymbol{\Theta}_j^\star\|_{\mathrm{F}}^2 \leq C\frac{B^2}{\sigma_{\mathrm{dir}}^2 \xi^2 |I_{ij}(\widehat{g})|}\mathcal{Q}_{ij}(\widehat{g}).$$

It now suffices to note that, by Lemma D.3, it holds that

$$\log\left(\mathbf{DN}(\mathcal{G}, \varepsilon)\right) \leq K(d+1)\log\left(\frac{3BK}{\varepsilon}\right)$$

and so, taking $\varepsilon = \frac{1}{T}$,

$$\Xi_3 \leq CK^2 T\xi + C\frac{B^8 Kd}{\sigma_{\mathrm{dir}}^8 \xi^8}\log\left(\frac{BKT}{\sigma_{\mathrm{dir}}\xi\delta}\right).$$

The result follows. ∎

---
**Algorithm 3** Combine and Permute Labels
---
1: **Initialize** Classifier $\widehat{g}$, new parameters $\left(\widehat{\mathbf{\Theta}}_{1,i}\right)_{1 \leq i \leq K}$, old parameters $\left(\widehat{\mathbf{\Theta}}_{0,i}\right)_{1 \leq i \leq K}$, threshold $A > 0$, gap $\Delta_{\mathrm{sep}} > 0$
2: **for** $i, j = 1, 2, \ldots, K$ **do**       (% Combine large clusters with similar $\widehat{\mathbf{\Theta}}_{1,i}$)
3:     **if** $i = j$ or $\min(I_i(\widehat{g}), I_j(\widehat{g})) < A$ **then**       (% Continue if cluster is too small)
4:         **Continue**
5:     **if** $\left\|\widehat{\mathbf{\Theta}}_{1,i} - \widehat{\mathbf{\Theta}}_{1,j}\right\| < \Delta_{\mathrm{sep}}$ **then**       (% Combine Cluster if parameters closer than gap)
6:         $\widehat{g} \leftarrow (j \mapsto i) \circ \widehat{g}$
7: $I \leftarrow [K]$
8: Empty permutation $\tilde{\pi} : [K] \to [K]$
9: Reorder $I$ so that if $i < i'$ then $|I_i(\widehat{g})| > |I_{i'}(\widehat{g})|$
10: **for** $i = 1, 2, \ldots, K$ **do**
11:     **if** $|I_i(\widehat{g})| > A$ **then**       (% Check if cluster is large enough)
12:         $j \leftarrow \operatorname{argmin}_{j' \in I} \left\|\widehat{\mathbf{\Theta}}_{1,i} - \widehat{\mathbf{\Theta}}_{0,j'}\right\|$
13:         $\tilde{\pi}(i) = j$
14:         $I \leftarrow I \setminus \{j\}$
15: **Return** $\tilde{\pi} \circ \widehat{g}$
---

# E   Modifying the Classifier and Mode Prediction

In this section, analyze OGD (Algorithm 2), the algorithm that modifies the classifer $\widehat{g}_\tau$ produced by *ermoracle* after epoch into a stabilized classifier $\widetilde{g}_\tau$ suitable for online prediction. The problem with the former classifer $\widehat{g}_\tau$ is that while it performs well on the past examples by construction, directional smoothness is not strong enough to imply generalization in the sense that $\widehat{g}_\tau$ will continue to perform well on epoch $\tau + 1$.

**Notation.**   We begin our analysis of OGD by defining some notation. For any $1 \leq t \leq T$, let $\tau(t)$ denote the epoch containing $t$, i.e., $\tau(t) = \max\left\{\tau' | \tau' E \leq t\right\}$. Further, recall the concatenated parameter notation $\mathbf{w}_{1:K} = (\mathbf{w}_1, \ldots, \mathbf{w}_K)$ for $\mathbf{w}_i \in \mathcal{B}^{d-1}$.

For a given epoch $\tau$, we let $\left\{\widehat{\mathbf{\Theta}}_{\tau,i} | i \in [K]\right\}, \widehat{g}_\tau$ denote the output of $\mathrm{ERMORACLE}(\bar{\mathbf{x}}_{1:\tau E}, \mathbf{y}_{1:\tau E})$. For any $g \in \mathcal{G}$ and $i, j \in [K]$, we denote

$$I_{ij;\tau}(g) = \{1 \leq t \leq \tau E | g(\bar{\mathbf{x}}_t) = i \text{ and } g_\star(\bar{\mathbf{x}}_t) = j\}. \tag{E.1}$$

Finally, for a fixed epoch $\tau$, if $\mathrm{ERMORACLE}$ has returned parameters $\widehat{\mathbf{\Theta}}_{\tau,i}$, define

$$\pi_\tau(i) = \operatorname*{argmin}_{1 \leq j \leq K} \|\widehat{\mathbf{\Theta}}_{\tau,i} - \mathbf{\Theta}_j^\star\|_{\mathrm{F}} \tag{E.2}$$

the function that takes a label $i$ according to $\widehat{g}_\tau$ and sends it to the closest label according to $g_\star$ as measured by difference in parameter matrices. The notation "$\pi$" signifies that $\pi_\tau$ represents a permutation when all the estimates $\widehat{\mathbf{\Theta}}_{\tau,i}$ are sufficiently accurate.

We conduct the analysis under the following condition, which informally states that for all sufficiently large clusters $I_{ij}$ considered in Algorithm 3 are sufficiently large, the associate parameters, Assumption 5

**Condition 1.** *We say that $\Xi \geq 0$ is a $\delta$-valid clusterability bound if it satisfies the following property. If Algorithm 3 is run with $A = 2K\Xi$ then with probability at least $1 - \delta$, for all $1 \leq i \leq K$ such that*

$$I_{i;\tau}(\widehat{g}_\tau) = \sum_{1 \leq j \leq K} |I_{ij;\tau}(\widehat{g}_\tau)| > 2K\Xi \tag{E.3}$$

*the following hold:*

   *1. There exists a unique $1 \leq j \leq K$ such that $|I_{ij;\tau}(\widehat{g}_\tau)| > \Xi$ and for that $j$, $|I_{ij;\tau}(\widehat{g}_\tau)| > K\Xi$.*

2. *For the $j$ given by the previous statement, $\left\|\widehat{\Theta}_i^\tau - \Theta_j^\star\right\| \leq \frac{\Delta_{\mathrm{sep}}}{4}$.*

3. *For the $j$ in the previous statements, it holds that in epoch $\tau + 1$, the classifer $\widehat{g}_{\tau+1}$ after the reordering step in Algorithm 2, the estimated parameter satisfies $|I_{ij;\tau}(\widehat{g}_{\tau+1})| > \Xi$ and in particular, $\left\|\widehat{\Theta}_i^{\tau+1} - \Theta_j^\star\right\| \leq \frac{\Delta_{\mathrm{sep}}}{4}$.*

Recall that the the pseudocode for OGD is in Algorithm 2. In words, the algorithm runs projected online gradient descent on $\widetilde{\ell}_{\gamma,t,\widehat{g}}$. We have the following result.

**Theorem 13.** *Suppose that we run Algorithm 1 in the setting described in Section 2. If we set $A = 2K\Xi$, then with probability at least $1 - \delta$, it holds that*

$$\sum_{t=1}^T \mathbb{I}\left[\pi_\tau(\widetilde{g}_{\tau(t)}(\bar{\mathbf{x}}_t)) \neq g_\star(\bar{\mathbf{x}})\right] \leq \frac{2BET\eta}{\gamma} + 3\left(1 + \frac{1}{\gamma}\right)(KE + 2K^2\Xi) + \frac{4K}{\eta} + \frac{\eta T}{\gamma^2} + \frac{T\gamma}{\sigma_{\mathrm{dir}}} + \sqrt{T\log\left(\frac{1}{\delta}\right)}$$

(E.4)

*where $\Xi$ is a parameter depending on the gap and the problem, defined in Lemma E.2.*

**Proof Strategy.** One challenge is that $\widetilde{g}_{\tau(t)}$ is that it is updated at the start of epoch $\tau + 1$, and is trained using labels corresponding to the permutation $\pi_\tau$. Therefore, we decompose to the error indicator into the case where $\pi_{\tau+1}(\widetilde{g}_{\tau(t)}(\bar{\mathbf{x}}_t)) = \pi_\tau(\widetilde{g}_{\tau(t)}(\bar{\mathbf{x}}_t))$, so this difference is immaterial, and into the cases where $\pi_{\tau+1}$ and $\pi_\tau$ differ.

$$\mathbb{I}\left[\pi_\tau(\widetilde{g}_{\tau(t)}(\bar{\mathbf{x}}_t)) \neq g_\star(\bar{\mathbf{x}})\right] \leq \underbrace{\mathbb{I}\left[\pi_{\tau+1}(\widetilde{g}_{\tau(t)}(\bar{\mathbf{x}}_t)) \neq g_\star(\bar{\mathbf{x}}_t)\right]}_{\text{(look-ahead classification error)}} + \underbrace{\mathbb{I}\left[\pi_\tau(\widetilde{g}_{\tau(t)}(\bar{\mathbf{x}}_t)) \neq \pi_{\tau+1}(\widetilde{g}_{\tau(t)}(\bar{\mathbf{x}}_t))\right]}_{\text{(permutation disagreement error)}}$$

(E.5)

We call the former term the "look-ahead classification error", because it applies the permutation from the subsequent epoch $\tau + 1$; the name for the second term is self-explanatory. Our online update OGD(Algorithm 2) controls the cumulative sum of the look-ahead classification error ( see Theorem 14 in the following section), while our labeling protocol (Algorithm 3) bounds the permutation disagreement error (see Lemma E.3 in the Appendix E.2).

### E.1 Look-ahead Classification Error

In this section we prove that the first term of (E.5) is small with high probability, which is the content of the following result:

**Theorem 14.** *With probability at least $1 - \delta$, the look-ahead classification error is at most*

$$\sum_{t=1}^T \mathbb{I}\left[\pi_{\tau(t)+1}(\widetilde{g}_\tau(\bar{\mathbf{x}}_t)) \neq g^\star(\bar{\mathbf{x}}_t)\right] \leq \frac{2BET\eta}{\gamma} + 2\left(1 + \gamma^{-1}\right)(KE + 2K^2\Xi) + \frac{4K}{\eta} + \frac{\eta T}{\gamma^2} + \frac{K^2 T\gamma}{\sigma_{\mathrm{dir}}} + \sqrt{T\log\left(\frac{1}{\delta}\right)}.$$

(E.6)

*Proof.* In addition to the notation $\widetilde{\ell}_{\gamma,t,\widehat{g}}$ defined in Equation (4.1), we introduce the following notation:

$$i_t^\star = g^\star(\bar{\mathbf{x}}_t) \qquad \widehat{i}_t = \arg\max_{1 \leq i \leq K}\left\langle \mathbf{w}_i^{\tau(t)}, \bar{\mathbf{x}}_t\right\rangle = \widetilde{g}_{\tau(t)}(\bar{\mathbf{x}}_t) \qquad \bar{i}_t = \widehat{g}_{\tau(t)+1}(\bar{\mathbf{x}}_t) \qquad \text{(E.7)}$$

or, in words, $i_t^\star$ is the groundtruth correct label, $\widehat{i}_t$ is the class predicted by the current epoch's linear predictors, and $\bar{i}_t$ is the class predicted by the ERM trained with the current epoch's data included. Additionally, let

$$\widetilde{\mathbf{w}}_{1:K}^t = \Pi_{(\mathcal{B}^{d-1})\times K}\left(\widetilde{\mathbf{w}}_{1:K}^{t-1} - \eta\nabla\widetilde{\ell}_{\gamma,t,\widehat{g}_{k(t)+1}}(\widetilde{\mathbf{w}}_{1:K}^{t-1})\right) \qquad \text{(E.8)}$$

i.e., the predicted weight if we were able to apply gradient descent with the labels predicted by the ERM trained on the current epoch's data and

$$\widetilde{i}_t = \arg\max_{1 \leq i \leq k}\left\langle \widetilde{\mathbf{w}}_i^t, \bar{\mathbf{x}}_t\right\rangle, \qquad \text{(E.9)}$$

the class predicted by $\widetilde{\mathbf{w}_{1:K}}$. We also consider the following "losses" that we will use in the analysis:

$$\widehat{\ell}_t(\mathbf{w}_{1:K}) = \mathbb{I}\left[\widehat{i}_t \neq \bar{i}_t\right] \qquad\qquad \ell_t^\star(\mathbf{w}_{1:K}) = \mathbb{I}\left[\pi_{\tau(t)+1}(\widehat{i}_t) \neq i_t^\star\right] \tag{E.10}$$

or, in words, $\widehat{\ell}_t$ is the event that our prediction is not equal to that of $\widehat{g}_{k(t)+1}$ and $\ell_t^\star$ is the event that our prediction is not equal to the groundtruth.

**Mistake Decomposition.** We now compute:

$$\sum_{t=1}^T \ell_t^\star(\mathbf{w}_{1:K}^{\tau(t)}) \leq \underbrace{\left(\sum_{t=1}^T \mathbb{I}\left[\pi_{\tau+1}(\widehat{g}_{\tau+1}(\bar{\mathbf{x}}_t)) \neq g_\star(\bar{\mathbf{x}}_t)\right]\right)}_{\text{Term}_1} + \left(\sum_{t=1}^T \mathbb{I}\left[\widetilde{g}_\tau(\bar{\mathbf{x}}_t) \neq \widehat{g}_{\tau+1}(\bar{\mathbf{x}}_t)\right]\right) \tag{E.11}$$

Lemma E.4 in Appendix E.2 controls $\text{Term}_1$. To control the other term, we note that for any $t, \gamma, g, \mathbf{w}_{1:K}$,

$$\widetilde{\ell}_{\gamma,t,g}(\mathbf{w}_{1:K}) = \max\left(0, \max_{j \neq g(\bar{\mathbf{x}}_t)}\left(1 - \frac{\langle \mathbf{w}_{g(\bar{\mathbf{x}}_t)} - \mathbf{w}_j, \bar{\mathbf{x}}_t\rangle}{\gamma}\right)\right) \geq \mathbb{I}[g_{\mathbf{w}_{1:K}}(\bar{\mathbf{x}}_t) \neq g(\bar{\mathbf{x}}_t)],$$

where $g_{\mathbf{w}_{1:K}}(\bar{\mathbf{x}}_t)$ is the classifier induced by the paramters $\mathbf{w}_{1:K}$. As $\widetilde{g}_\tau(\bar{\mathbf{x}}_t)$ is the classifier induced by $\mathbf{w}_{1:K}^\tau$, we have

$$\sum_{t=1}^T \mathbb{I}\left[\widetilde{g}_\tau(\bar{\mathbf{x}}_t) \neq \widehat{g}_{\tau+1}(\bar{\mathbf{x}}_t)\right] \leq \sum_{t=1}^T \widetilde{\ell}_{\gamma,t,\widehat{g}_{\tau+1}}(\mathbf{w}_{1:K}^\tau)$$

$$\leq \underbrace{\left(\sum_{t=1}^T \widetilde{\ell}_{\gamma,t,\widehat{g}_{\tau+1}}(\mathbf{w}_{1:K}^\tau) - \widetilde{\ell}_{\gamma,t,\widehat{g}_{\tau+1}}(\widetilde{\mathbf{w}}_{1:K}^t)\right)}_{\text{Term}_2} + \left(\sum_{t=1}^T \widetilde{\ell}_{\gamma,t,\widehat{g}_{\tau+1}}(\widetilde{\mathbf{w}}_{1:K}^t)\right)$$

Contining, we have

$$\sum_{t=1}^T \ell_t^\star(\mathbf{w}_{1:K}^{\tau(t)}) \leq \text{Term}_1 + \text{Term}_2 + \sum_{t=1}^T \widetilde{\ell}_{\gamma,t,\widehat{g}_{\tau+1}}(\widetilde{\mathbf{w}}_{1:K}^t)$$

$$\leq \text{Term}_1 + \text{Term}_2 + \underbrace{\sup_{\mathbf{w}_{1:K}}\left(\sum_{t=1}^T \widetilde{\ell}_{\gamma,t,\widehat{g}_{\tau+1}}(\widetilde{\mathbf{w}}_{1:K}^t) - \widetilde{\ell}_{\gamma,t,\widehat{g}_{\tau+1}}(\mathbf{w}_{1:K})\right)}_{\text{Term}_3}$$

$$+ \underbrace{\inf_{\mathbf{w}_{1:K}} \sum_{t=1}^T \widetilde{\ell}_{\gamma,t,\widehat{g}_{\tau+1}}(\mathbf{w}_{1:K})}_{\text{Term}_4}$$

**Bounding the "delay" penalty: $\text{Term}_2$.** $\text{Term}_2$ corresponds to the error we may suffer from using delayed gradient updates. We now observe that

$$\left\|\nabla\widetilde{\ell}_{\gamma,t,\widehat{g}_{\tau(t)+1}}\right\| \leq \frac{\|\bar{\mathbf{x}}_t\|}{\gamma} \leq \frac{B}{\gamma} \tag{E.12}$$

and that projection onto a convex body is a contraction. Furthermore, the gradient update in Algorithm 2, Line 3 only affects at most two distinct $i, j$ in the coordinates of $\mathbf{w}_{1:K}$ per update. Thus it holds that, for all $t$,

$$\left\|\widetilde{\mathbf{w}}_{1:K}^t - \mathbf{w}_{1:K}^{\tau(t)}\right\| \leq 2E\eta.$$

Applying (E.12) tells us that

$$\text{Term}_2 := \sum_{t=1}^T \widetilde{\ell}_{\gamma,t,\widehat{g}_{\tau+1}}(\mathbf{w}_{1:K}^\tau) - \widetilde{\ell}_{\gamma,t,\widehat{g}_{\tau+1}}(\widetilde{\mathbf{w}}_{1:K}^t) \leq \frac{2BET\eta}{\gamma}. \tag{E.13}$$

**Bounding the regret term:** $\text{Term}_3$. We see that $\text{Term}_3$ is just the regret for Online Gradient Descent for losses with gradients bounded in norm by $\frac{1}{\gamma}$ on the $K$-fold product of unit balls, having diameter $2\sqrt{K}$. Thus it holds by classical results (c.f. Hazan et al. [32, Theorem 3.1]) that

$$\text{Term}_3 = \sum_{t=1}^{T} \widetilde{\ell}_{\gamma,t,\widehat{g}_{k(t)}}(\widetilde{\mathbf{w}}_t) - \inf_{\mathbf{w}_{1:K}} \widetilde{\ell}_{\gamma,t,\widehat{g}_{k(t)}}(\mathbf{w}_{1:K}) \leq \frac{4K}{\eta} + \frac{\eta T}{\gamma^2}. \qquad (\text{E}.14)$$

**Bounding the comparator:** $\text{Term}_4$. To bound $\text{Term}_4$, we aim to move from the losses with margin $\widetilde{\ell}_{\gamma,t,\widehat{g}_{\tau+1}}$ back to sum measure of comulative classification error. The central object in this analysis is the following "ambiguous set", defined for each parameter $\mathbf{w}_{1:K}$:

$$D_\gamma(\mathbf{w}_{1:K}) = \left\{ \bar{\mathbf{x}} : |\langle \mathbf{w}_{i,1:d} - \mathbf{w}_{j,1:d}, \mathbf{x} \rangle - w_{i,d+1} + w_{j,d+1}| \leq \gamma \text{ for some } 1 \leq i \leq K \right\}, \quad (\text{E}.15)$$

where we denote the first $d$ coordinates of $\mathbf{w}_i$ by $\mathbf{w}_{i,1:d}$ and its last coordinate by the scalar $w_{i,d+1}$. Then, by directional smoothness and a union bound, the following is true for any fixed $\mathbf{w}_{1:K}$:

$$\mathbb{P}(\bar{\mathbf{x}}_t \in D_\gamma(\mathbf{w}_{1:K}) \mid \mathcal{F}_t) \leq K^2 \sup_{i,j} \mathbb{P}(|\langle \mathbf{w}_{i,1:d} - \mathbf{w}_{j,1:d}, \mathbf{x}_t \rangle - w_{i,d+1} + w_{j,d+1}| \leq \gamma \mid \mathcal{F}_t) \leq \frac{K^2\gamma}{\sigma_{\text{dir}}}.$$

Let $\mathbf{w}_{1:K}^\star$ be the parameter associated with $g^\star$. Then a consequence of the above display and Azuma's inequality is that most $\bar{\mathbf{x}}_t$'s fall outside of $D_\gamma(\mathbf{w}_{1:K}^\star)$:

$$\sum_{t=1}^{T} \mathbb{I}\left[\bar{\mathbf{x}}_t \in D_\gamma(\mathbf{w}_{1:K}^\star)\right] \leq \sum_{t=1}^{T} \mathbb{P}_t\left(\bar{\mathbf{x}}_t \in D_\gamma(\mathbf{w}_{1:K}^\star)\right) + \sqrt{T \log\left(\frac{1}{\delta}\right)}$$

$$\leq \frac{K^2 T \gamma}{\sigma_{\text{dir}}} + \sqrt{T \log\left(\frac{1}{\delta}\right)}. \qquad (\text{E}.16)$$

We compute:

$$\inf_{\mathbf{w}_{1:K}} \sum_{t=1}^{T} \widetilde{\ell}_{\gamma,t,\widehat{g}_{\tau+1}}(\mathbf{w}_{1:K}) \leq \inf_{\mathbf{w}_{1:K}} \sum_{t=1}^{T} \mathbb{I}\left[\bar{\mathbf{x}}_t \in D_\gamma(\mathbf{w}_{1:K})\right] + \left(1 + \frac{2}{\gamma}\right) \sum_{t=1}^{T} \widehat{\ell}_t(\mathbf{w}_{1:K})$$

$$\leq \inf_{\mathbf{w}_{1:K}} \sum_{t=1}^{T} \mathbb{I}\left[\bar{\mathbf{x}}_t \in D_\gamma(\mathbf{w}_{1:K})\right] + \left(1 + \frac{2}{\gamma}\right) \sum_{t=1}^{T} \ell_t^\star(\mathbf{w}_{1:K}) + \mathbb{I}\left[\pi_{\tau(t)+1}(\bar{i}_t) \neq i_t^\star\right]$$

$$\leq \left(1 + \frac{2}{\gamma}\right) \sum_{t=1}^{T} \mathbb{I}\left[\pi_{\tau(t)+1}(\bar{i}_t) \neq i_t^\star\right] + \inf_{\mathbf{w}_{1:K}} \sum_{t=1}^{T} \mathbb{I}\left[\bar{\mathbf{x}}_t \in D_\gamma(\mathbf{w}_{1:K})\right] + \left(1 + \frac{2}{\gamma}\right) \ell_t^\star(\mathbf{w}_{1:K})$$

$$\leq \left(1 + \frac{2}{\gamma}\right) \sum_{t=1}^{T} \mathbb{I}\left[\pi_{\tau(t)+1}(\bar{i}_t) \neq i_t^\star\right] + \sum_{t=1}^{T} \mathbb{I}\left[\bar{\mathbf{x}}_t \in D_\gamma(\mathbf{w}_{1:K}^\star)\right]$$

$$\leq \left(1 + \frac{2}{\gamma}\right) \underbrace{\sum_{t=1}^{T} \mathbb{I}\left[\pi_{\tau(t)+1}(\bar{i}_t) \neq i_t^\star\right]}_{=\text{Term}_1} + \frac{K^2 T \gamma}{\sigma_{\text{dir}}} + \sqrt{T \log\left(\frac{1}{\delta}\right)}.$$

where the first inequality follows by Lemma E.1, the second inequality is trivial, the third inequality follows because the final term does not depend on $\mathbf{w}_{1:K}$, the fourth inequality follows from the realizability assumption, i.e., that $\ell_t^\star(\mathbf{w}_{1:K}^\star) = 0$ for all $t$, and the last from Equation (E.16) (recalling the definition of $\text{Term}_1$).

Thus it holds with probability at least $1 - \delta$ that

$$\text{Term}_4 \leq \left(1 + \frac{2}{\gamma}\right) \text{Term}_1 + \frac{K^2 T \gamma}{\sigma_{\text{dir}}} + \sqrt{T \log\left(\frac{1}{\delta}\right)}.$$

**Concluding the proof.** To conclude, we see that

$$\sum_{t=1}^{T} \ell_t^{\star}(\mathbf{w}_{1:K}^{\tau(t)}) \leq \mathrm{Term}_1 + \mathrm{Term}_2 + \mathrm{Term}_3 + \mathrm{Term}_4 \tag{E.17}$$

$$\leq \left(1 + \frac{2}{\gamma}\right)\mathrm{Term}_1 + \frac{2BET\eta}{\gamma} + \frac{4K}{\eta} + \frac{\eta T}{\gamma^2} + \frac{K^2 T\gamma}{\sigma_{\mathrm{dir}}} + \sqrt{T\log\left(\frac{1}{\delta}\right)}. \tag{E.18}$$

By Lemma E.4 it holds that

$$\mathrm{Term}_1 = \sum_{t=1}^{T} \mathbb{I}\left[\pi_{\tau(t)+1}(\bar{i}_t) \neq i_t^{\star}\right] := \sum_{t=1}^{T} \mathbb{I}\left[\pi_{\tau(t)+1}(\widehat{g}_{\tau(t)+1}(\bar{\mathbf{x}}_t)) \neq g_{\star}(\bar{\mathbf{x}}_t)\right] \leq KE + 2K^2\Xi,$$
$$\tag{E.19}$$

where $\Xi$ is defined in Lemma E.2. This concludes the proof. ∎

**Lemma E.1** (Bound indicator by soft margin). *For any $\gamma, t, \widehat{g}$, it holds that if each component of $\mathbf{w}$ has unit norm, then*

$$\widetilde{\ell}_{\gamma,t,\widehat{g}}(\mathbf{w}) \leq \mathbb{I}\left[\bar{\mathbf{x}}_t \in D_{\gamma}(\mathbf{w}_{1:K})\right] + \left(1 + \frac{2}{\gamma}\right)\widehat{\ell}_t(\mathbf{w}_{1:K}). \tag{E.20}$$

*Proof.* We prove this by casework. Suppose that $\widehat{\ell}_t(\mathbf{w}_{1:K}) = 0$. Then it holds that

$$\widehat{g}(\bar{\mathbf{x}}_t) = \arg\max_{1 \leq j \leq K} \langle \mathbf{w}_j, \bar{\mathbf{x}}_t \rangle \tag{E.21}$$

and in particular for all $j \neq \widehat{g}(\bar{\mathbf{x}}_t)$, it holds that $\langle \mathbf{w}_{\widehat{g}(\bar{\mathbf{x}}_t)} - \mathbf{w}_j, \bar{\mathbf{x}}_t \rangle \geq 0$. If $\bar{\mathbf{x}}_t \notin D_{\gamma}(\mathbf{w}_{1:K})$ then it holds by construction that

$$\max_{j \neq \widehat{g}(\bar{\mathbf{x}}_t)} 1 - \frac{\langle \mathbf{w}_{\widehat{g}(\bar{\mathbf{x}}_t)} - \mathbf{w}_j, \bar{\mathbf{x}}_t \rangle}{\gamma} \leq 0 \tag{E.22}$$

and the conclusion clearly holds. If it holds that

$$\max_{j \neq \widehat{g}(\bar{\mathbf{x}}_t)} 1 - \frac{\langle \mathbf{w}_{\widehat{g}(\mathbf{x}_t)} - \mathbf{w}_j, \bar{\mathbf{x}}_t \rangle}{\gamma} > 0 \tag{E.23}$$

then either there is some $j$ such that $\langle \mathbf{w}_{\widehat{g}(\mathbf{x}_t)-\mathbf{w}_j}, \bar{\mathbf{x}}_t \rangle < 0$ or there is some $j$ such that $\langle \mathbf{w}_{\widehat{g}(\bar{\mathbf{x}}_t)} - \mathbf{w}_j, \bar{\mathbf{x}}_t \rangle < \gamma$ and so $\bar{\mathbf{x}}_t \in D_{\gamma}(\mathbf{w}_{1:K})$. We cannot have the former by the assumption that $\widehat{\ell}_t(\mathbf{w}_{1:K}) = 0$ so the latter holds and the inequality follows in this case.

Now suppose that $\widehat{\ell}_t(\mathbf{w}_{1:K}) = 1$. Then we see that as $||\mathbf{w}_i|| \leq 1$ for all $i$, $\widetilde{\ell}_{\gamma,t,\widehat{g}} \leq \left(1 + \frac{2}{\gamma}\right)$ uniformly and the result holds. ∎

### E.2 Bounding Permutation Disagreement and $\mathrm{Term}_1$

In this section, we provide a bound on the permutation disagreement error - the second term of (E.5) - as well as on $\mathrm{Term}_1$ from the section above . We begin by notion that Algorithm 3 ensures that large clusters remain large accross epochs, a statement formalized by the following result:

**Lemma E.2.** *Define the following terms:*

$$\Xi_1 = CK^2 T\xi + C\frac{B^8 Kd}{\sigma_{\mathrm{dir}}^8 \xi^8}\log\left(\frac{BKT}{\sigma_{\mathrm{dir}}\xi\delta}\right) \tag{E.24}$$

$$\Xi_2 = C\frac{B^2}{\sigma_{\mathrm{dir}}^2 \xi^2 \Delta_{\mathrm{sep}}^2}\left(\varepsilon_{\mathrm{orac}} + 1 + K^3 B^2 R^2 d^2 m\nu^2 \sqrt{T}\log\left(\frac{TRBmdK}{\delta}\right)\right) + \sum_{t=1}^{T} ||\boldsymbol{\delta}_t||. \tag{E.25}$$

*Then $\Xi := \max(\Xi_1, \Xi_2)$ is a $\delta$-valid clusterability parameter in the sense of Condition 1.*

*Proof.* By the pigeonhole principle, if $I_i(\widehat{g}_\tau) > 2K\Xi$ there must be at least one $j$ such that $|I_{ij;\tau}(\widehat{g}_\tau)| > \Xi$ and thus by Theorem 11, it holds that $\left\|\widehat{\Theta}_i^\tau - \Theta_j^\star\right\| \leq \frac{\Delta_{\text{sep}}}{4}$ and thus the second statement holds. If there is another $j'$ such that $|I_{ij';\tau}(\widehat{g}_\tau)| > \Xi$ then it would also hold that $\left\|\widehat{\Theta}_i^\tau - \Theta_{j'}^\star\right\| \leq \frac{\Delta_{\text{sep}}}{4}$ and the triangle inequality would then imply that $\|\Theta_j^\star - \Theta_{j'}^\star\| \leq \frac{\Delta_{\text{sep}}}{2} < \Delta_{\text{sep}}$, which ensures that $j = j'$ by Assumption 5. Applying pigeonhole again then shows that

$$|I_{ij;\tau}(\widehat{g}_\tau)| = I_{i;\tau}(\widehat{g}_\tau) - \sum_{j' \neq j} |I_{ij';\tau}(\widehat{g}_\tau)| > 2K\Xi - (K-1)\Xi = (K+1)\Xi. \tag{E.26}$$

Thus the first statement holds. For the last statement, note that as there are at least $K\Xi$ times $t < \tau E$ such that $g_\star(\bar{\mathbf{x}}_t) = j$, there are at least $\Xi$ times $t < (\tau+1)E$ such that $t \in I_{i'j;\tau}(\widehat{g}_{\tau+1})$. This implies, again by Theorem 11, that $\left\|\widehat{\Theta}_{i'}^{\tau+1} - \Theta_j^\star\right\| \leq \frac{\Delta_{\text{sep}}}{4}$ and thus $\left\|\widehat{\Theta}_{i'}^{\tau+1} - \widehat{\Theta}_i^\tau\right\| \leq \frac{\Delta_{\text{sep}}}{2}$. If $i' \neq i$ after running Algorithm 3 then there must be an $i''$ such that $I_{i'';\tau}(\widehat{g}_{\tau+1}) > K\Xi$ and $\left\|\widehat{\Theta}_{i''}^{\tau+1} - \widehat{\Theta}_i^\tau\right\| \leq \frac{\Delta_{\text{sep}}}{2}$. But if this is the case then the triangle inequality implies that $\left\|\widehat{\Theta}_{i''}^{\tau+1} - \widehat{\Theta}_{i'}^{\tau+1}\right\| < \Delta_{\text{sep}}$ and as they both are sufficiently large, there were merged by the first half of Algorithm 3, implying that $i'' = i'$ and, in turn, that $i' = i$, by the the second half of Algorithm 3. Thus the first half of the third statement holds. To conclude, simply apply Theorem 11. ∎

We are now ready to prove the main bound:

**Lemma E.3.** *With probability at least $1 - \delta$, it holds that*

$$\sum_{t=1}^T \mathbb{I}\left[\pi_\tau(\widetilde{g}_{\tau(t)}(\bar{\mathbf{x}}_t)) \neq \pi_{\tau+1}(\widetilde{g}_{\tau(t)}(\bar{\mathbf{x}}))\right] \leq KE + 2K^2\Xi \tag{E.27}$$

*Proof.* We begin by fixing some $1 \leq i \leq K$ and bounding the number of epochs $\tau$ such that $\pi_\tau(i) \neq \pi_{\tau+1}(i)$. We will restrict our focus to the probability $1 - \delta$ event such that the conclusion of Lemma E.2 holds. Suppose that there is an $i$ such that $I_{i;\tau}(\widehat{g}_\tau) > 2K\Xi$ and $\pi_\tau(i) = j \neq j' = \pi_{\tau+1}(i)$. Then by Assumption 5 and the triangle inequality, we have that

$$\left\|\widehat{\Theta}_i^\tau - \Theta_j^\star\right\| + \frac{\Delta_{\text{sep}}}{4} \leq \left\|\widehat{\Theta}_i^{\tau+1} - \Theta_{j'}^\star\right\| \tag{E.28}$$

$$\left\|\widehat{\Theta}_i^{\tau+1} - \Theta_{j'}^\star\right\| + \frac{\Delta_{\text{sep}}}{4} \leq \left\|\widehat{\Theta}_i^\tau - \Theta_j^\star\right\| \tag{E.29}$$

Rearranging and again applying Assumption 5 tells us that

$$\frac{\Delta_{\text{sep}}}{2} < \left\|\widehat{\Theta}_i^\tau - \widehat{\Theta}_i^{\tau+1}\right\|. \tag{E.30}$$

Applying the second and third statements of Lemma E.2, however, brings a contradiction. Thus we have, on the high probability event from Lemma E.2,

$$\mathbb{I}\left[\pi_\tau(\widetilde{g}_{\tau(t)}(\bar{\mathbf{x}}_t)) \neq \pi_{\tau+1}(\widetilde{g}_{\tau(t)}(\bar{\mathbf{x}}))\right] \leq \mathbb{I}\left[I_{i;\tau}(\widehat{g}_{\tau(t)}) \leq 2K\Xi\right]. \tag{E.31}$$

Thus we have

$$\sum_{t=1}^T \mathbb{I}\left[\pi_\tau(\widetilde{g}_{\tau(t)}(\bar{\mathbf{x}}_t)) \neq \pi_{\tau+1}(\widetilde{g}_{\tau(t)}(\bar{\mathbf{x}}))\right] \leq \sum_{i=1}^K \sum_{t=1}^T \mathbb{I}\left[\widetilde{g}_{\tau(t)}(\bar{\mathbf{x}}_t) = i\right] \mathbb{I}\left[I_{i;\tau}(\widehat{g}_{\tau(t)}) \leq 2K\Xi\right] \tag{E.32}$$

$$\leq K\left(E + 2K\Xi\right). \tag{E.33}$$

The result follows. ∎

We may also apply Lemma E.2 to show that $\pi_{\tau+1}(\widehat{g}_{\tau+1}(\bar{\mathbf{x}}_t)) = g_\star(\bar{\mathbf{x}}_t)$ most of the time. We formalize this statement in the following result:

**Lemma E.4** (Wrong Labels Bound). *With probability at least $1 - \delta$, it holds that*

$$\sum_{t=1}^{T} \mathbb{I}\left[\pi_{\tau+1}(\widehat{g}_{\tau(t)+1}(\bar{\mathbf{x}}_t)) \neq g_\star(\bar{\mathbf{x}}_t)\right] \leq KE + 2K^2\Xi \tag{E.34}$$

*Proof.* By Lemma E.2, it holds that

$$\mathbb{I}\left[\pi_{\tau+1}(\widehat{g}_{\tau(t)+1}(\bar{\mathbf{x}}_t)) \neq g_\star(\bar{\mathbf{x}}_t)\right] \leq \sum_{i=1}^{K} \mathbb{I}\left[\widehat{g}_{\tau(t)+1}(\bar{\mathbf{x}}_t) = i\right] \mathbb{I}\left[I_{i;\tau}(\widehat{g}_{\tau(t)+1}) \leq 2K\Xi\right]. \tag{E.35}$$

Thus we may argue exactly as in Lemma E.3 to conclude the proof. ∎

# F Proving Theorem 3

In this section we formally state and prove our main result, i.e., that Algorithm 1 is an oracle-efficient algorithm for achieving expected regret polynomial in all the parameters and scaling as $T^{1-\Omega(1)}$ as the horizon tends to infinity. We have the following formal statement, which we first estalbish under the assumption of a strict separation between parameters (Assumption 5).

**Theorem 15.** *Suppose that Assumptions 2-5 hold. Then, for the correct choices of $\gamma, \eta$, and $E$, given in the proof below (found above (F.7)), Algorithm 1 experiences expected regret:*

$$\mathbb{E}\left[\text{Reg}_T\right] \leq CB^{10}R^2 md\nu^2 K^4 \sigma_{\text{dir}}^{-4} \Delta_{\text{sep}}^{-1} T^{\frac{35}{36}}\left(\varepsilon_{\text{orac}} + \log(TRBmdK)\right) + CBRK^2\sum_{t=1}^{T}\|\boldsymbol{\delta}_t\|.$$

*Note that the last term is $O\left(BRK^2 T\varepsilon_{\text{crp}}\right)$ in the worst case.*

*Proof.* We recall that Algorithm 1 proceeds in epochs of length $E$, a parameter to be specified. At the beginning of epoch $\tau$, at time $(\tau - 1)E + 1$, ERMORACLE is called, producing $\widehat{\boldsymbol{\Theta}}_{\tau,i}$ for $1 \leq i \leq K$ and $\widehat{g}_\tau$, for a total of $\frac{T}{E} \leq T$ calls to ERMORACLE. We then use Algorithm 2 to modify $\widehat{g}_\tau$ to $\widetilde{g}_\tau$ and predict $\hat{\mathbf{y}}_t = \widehat{\boldsymbol{\Theta}}_{\tau,\widetilde{g}_\tau(\bar{\mathbf{x}}_t)}\bar{\mathbf{x}}_t$. Thus we have

$$\begin{aligned}
\text{Reg}_T &= \sum_{t=1}^{T}\left\|\hat{\mathbf{y}}_t - \boldsymbol{\Theta}_{g_\star(\bar{\mathbf{x}}_t)}^\star\bar{\mathbf{x}}_t\right\|^2 \\
&= \sum_{\tau=1}^{T/E}\sum_{t=(\tau-1)E+1}^{\tau E}\left\|\left(\widehat{\boldsymbol{\Theta}}_{\tau,\widetilde{g}_\tau(\bar{\mathbf{x}}_t)} - \boldsymbol{\Theta}_{g_\star(\bar{\mathbf{x}}_t)}^\star\right)\bar{\mathbf{x}}_t\right\|^2 \\
&= \left(\sum_{\tau=1}^{T/E}\sum_{t=(\tau-1)E+1}^{\tau E}\left\|\left(\widehat{\boldsymbol{\Theta}}_{\tau,\widetilde{g}_\tau(\bar{\mathbf{x}}_t)} - \boldsymbol{\Theta}_{g_\star(\bar{\mathbf{x}}_t)}^\star\right)\bar{\mathbf{x}}_t\right\|^2 \mathbb{I}\left[\pi_\tau\left(\widetilde{g}_\tau(\bar{\mathbf{x}}_t)\right) \neq g_\star(\bar{\mathbf{x}}_t)\right]\right) \\
&\quad + \left(\sum_{\tau=1}^{T/E}\sum_{t=(\tau-1)E+1}^{\tau E}\left\|\left(\widehat{\boldsymbol{\Theta}}_{\tau,\widetilde{g}_\tau(\bar{\mathbf{x}}_t)} - \boldsymbol{\Theta}_{\pi_\tau(\widetilde{g}_\tau(\bar{\mathbf{x}}_t))}^\star\right)\bar{\mathbf{x}}_t\right\|^2\right).
\end{aligned}$$

For the first term, we have

$$\sum_{\tau=1}^{T/E}\sum_{t=(\tau-1)E+1}^{\tau E}\left\|\left(\widehat{\boldsymbol{\Theta}}_{\tau,\widetilde{g}_\tau(\bar{\mathbf{x}}_t)} - \boldsymbol{\Theta}_{g_\star(\bar{\mathbf{x}}_t)}^\star\right)\bar{\mathbf{x}}_t\right\|^2 \mathbb{I}\left[\pi_\tau\left(\widetilde{g}_\tau(\bar{\mathbf{x}}_t)\right) \neq g_\star(\bar{\mathbf{x}}_t)\right] \leq \sum_{\tau=1}^{T/E}\sum_{t=(\tau-1)E+1}^{\tau E}4B^2R^2\mathbb{I}\left[\pi_\tau\left(\widetilde{g}_\tau(\bar{\mathbf{x}}_t)\right) \neq g_\star(\bar{\mathbf{x}}_t)\right]$$

$$\leq 4B^2R^2\sum_{t=1}^{T}\mathbb{I}\left[\pi_\tau\left(\widetilde{g}_\tau(\bar{\mathbf{x}}_t)\right) \neq g_\star(\bar{\mathbf{x}}_t)\right]$$

and thus, applying Theorem 13, it holds that the above expression is bounded by

$$4B^2R^2\left(\frac{2BET\eta}{\gamma} + 3\left(1 + \frac{1}{\gamma}\right)(KE + 2K^2\Xi) + \frac{4K}{\eta} + \frac{\eta T}{\gamma^2} + \frac{T\gamma}{\sigma_{\text{dir}}} + \sqrt{T\log\left(\frac{1}{\delta}\right)}\right). \tag{F.1}$$

For the second term, we have

$$\sum_{\tau=1}^{T/E} \sum_{t=(\tau-1)E+1}^{\tau E} \left\| \left(\widehat{\boldsymbol{\Theta}}_{\tau,\widetilde{g}_\tau(\bar{\mathbf{x}}_t)} - \boldsymbol{\Theta}^\star_{\pi_\tau(\widetilde{g}_\tau(\bar{\mathbf{x}}_t))}\right)\bar{\mathbf{x}}_t \right\|^2 \leq B^2 \sum_{\tau=1}^{T/E} \sum_{t=(\tau-1)E+1}^{\tau E} \|\widehat{\boldsymbol{\Theta}}_{\tau,\widetilde{g}_\tau(\bar{\mathbf{x}}_t)} - \boldsymbol{\Theta}^\star_{\pi_\tau(\widetilde{g}_\tau(\bar{\mathbf{x}}_t))}\|_{\mathrm{F}}^2$$

$$= B^2 \sum_{i=1}^{K} \sum_{\tau=1}^{T/E} \sum_{t=(\tau-1)E+1}^{\tau E} \|\widehat{\boldsymbol{\Theta}}_{\tau,i} - \boldsymbol{\Theta}^\star_{\pi_\tau(i)}\|_{\mathrm{F}}^2 \mathbb{I}\left[\widetilde{g}_\tau(\bar{\mathbf{x}}_t) = i\right].$$

Now, let

$$\Xi_1 = CK^2 T\xi + C\frac{B^8 Kd}{\sigma_{\mathrm{dir}}^8 \xi^8} \log\left(\frac{BKT}{\sigma_{\mathrm{dir}}\xi\delta}\right) \tag{F.2}$$

$$\Xi_2' = C\frac{B^2}{\sigma_{\mathrm{dir}}^2 \xi^2 \alpha}\left(\varepsilon_{\mathrm{orac}} + 1 + K^3 B^2 R^2 d^2 m\nu^2 \sqrt{T} \log\left(\frac{TRBmdK}{\delta}\right)\right) + \sum_{t=1}^{T} \|\boldsymbol{\delta}_t\| \tag{F.3}$$

$$\Xi' = \max(\Xi_1, \Xi_2). \tag{F.4}$$

Note that as long as $\alpha < \Delta_{\mathrm{sep}}^2$ then Lemma E.2 tells us that if we run Algorithm 1 with $A \geq 2K\Xi'$, then with probability at least $1 - \delta$ it holds for all $1 \leq i \leq K$ and all $\tau \leq \tau'$ that if

$$I_i(\widehat{g}_\tau) > 2K\Xi'$$

then

$$\left|I_{i\pi_\tau'(i)}(\widehat{g}_{\tau'})\right| > \Xi'$$

and

$$\|\widehat{\boldsymbol{\Theta}}_i^{\tau'} - \boldsymbol{\Theta}^\star_{\pi_{\tau'}(i)}\|_{\mathrm{F}}^2 \leq \alpha.$$

Furthermore, under this condition, Lemma E.2 tells us that

$$I_i(\widehat{g}_{\tau'}) > 2K\Xi.$$

Fixing an $i$, then, we see that

$$\sum_{\tau=1}^{T/E} \sum_{t=(\tau-1)E+1}^{\tau E} \|\widehat{\boldsymbol{\Theta}}_{\tau,i} - \boldsymbol{\Theta}^\star_{\pi_\tau(i)}\|_{\mathrm{F}}^2 \mathbb{I}\left[\widetilde{g}_\tau(\bar{\mathbf{x}}_t) = i\right] = \sum_{\tau=1}^{T/E} \sum_{t=(\tau-1)E+1}^{\tau E} \|\widehat{\boldsymbol{\Theta}}_{\tau,i} - \boldsymbol{\Theta}^\star_{\pi_\tau(i)}\|_{\mathrm{F}}^2 \mathbb{I}\left[\widetilde{g}_\tau(\bar{\mathbf{x}}_t) = i \wedge I_i(\widehat{g}_\tau) \leq 2K\Xi'\right]$$

$$+ \sum_{\tau=1}^{T/E} \sum_{t=(\tau-1)E+1}^{\tau E} \|\widehat{\boldsymbol{\Theta}}_{\tau,i} - \boldsymbol{\Theta}^\star_{\pi_\tau(i)}\|_{\mathrm{F}}^2 \mathbb{I}\left[\widetilde{g}_\tau(\bar{\mathbf{x}}_t) = i \wedge I_{i(\widehat{g}_\tau)} > 2K\Xi'\right].$$

For the first term, we see that

$$\sum_{\tau=1}^{T/E} \sum_{t=(\tau-1)E+1}^{\tau E} \|\widehat{\boldsymbol{\Theta}}_{\tau,i} - \boldsymbol{\Theta}^\star_{\pi_\tau(i)}\|_{\mathrm{F}}^2 \mathbb{I}\left[\widetilde{g}_\tau(\bar{\mathbf{x}}_t) = i \wedge I_i(\widehat{g}_\tau) \leq 2K\Xi'\right] \leq 4R^2 \sum_{\tau=1}^{T/E} \sum_{t=(\tau-1)E+1}^{\tau E} \mathbb{I}\left[\widetilde{g}_\tau(\bar{\mathbf{x}}_t) = i \wedge I_i(\widehat{g}_\tau) \leq 2K\Xi'\right]$$

$$\leq 4R^2 \sum_{\tau=1}^{T/E} \sum_{t=(\tau-1)E+1}^{\tau E} 2K\Xi'$$

$$\leq \frac{4R^2 TK\Xi'}{E}.$$

For the second term, we have with high probability that

$$\sum_{\tau=1}^{T/E} \sum_{t=(\tau-1)E+1}^{\tau E} \|\widehat{\boldsymbol{\Theta}}_{\tau,i} - \boldsymbol{\Theta}^\star_{\pi_\tau(i)}\|_{\mathrm{F}}^2 \mathbb{I}\left[\widetilde{g}_\tau(\bar{\mathbf{x}}_t) = i \wedge I_{i(\widehat{g}_\tau)} > 2K\Xi'\right] \leq \sum_{\tau=1}^{T/E} \sum_{t=(\tau-1)E+1}^{\tau E} \alpha$$

$$\leq \alpha T.$$

Summing over all $K$, we see that

$$\sum_{i=1}^{K} \sum_{\tau=1}^{T/E} \sum_{t=(\tau-1)E+1}^{\tau E} \|\widehat{\Theta}_{\tau,i} - \Theta_{\pi_\tau(i)}^\star\|_F^2 \mathbb{I}\left[\tilde{g}_\tau(\bar{\mathbf{x}}_t) = i\right] \leq B^2 K \left(4R^2 \frac{T}{E} K\Xi' + \alpha T\right). \qquad \text{(F.5)}$$

Thus, combining (F.1) and (F.5) tells us that with probability at least $1 - \delta$ it holds that $\text{Reg}_T$ is upper bounded by

$$4B^2 R^2 \left(\frac{2BET\eta}{\gamma} + 3\left(1 + \frac{1}{\gamma}\right)(KE + 2K^2\Xi) + \frac{4K}{\eta} + \frac{\eta T}{\gamma^2} + \frac{T\gamma}{\sigma_{\text{dir}}} + \sqrt{T\log\left(\frac{1}{\delta}\right)}\right)$$
$$+ B^2 K \left(4R^2 \frac{T}{E} K\Xi' + \alpha T\right)$$

where $\Xi'$ is given in (F.4) and $\Xi$ is given in Lemma E.2. Now, setting

$$E = T^{\frac{17}{18}}, \qquad \gamma = T^{-\frac{1}{36}}, \qquad \xi = T^{-\frac{1}{9}}, \qquad \alpha = T^{-\frac{1}{6}}, \qquad \eta = T^{-\frac{19}{36}}, \qquad \text{(F.6)}$$

gives, with probability at least $1 - \delta$,

$$\text{Reg}_T \leq CB^{10}R^2 md\nu^2 K^4 \sigma_{\text{dir}}^{-4} \Delta_{\text{sep}}^{-1} T^{\frac{35}{36}} \left(\varepsilon_{\text{orac}} + \log(TRBmdK)\right) + CBRK^2 \sum_{t=1}^{T} \|\boldsymbol{\delta}_t\|. \qquad \text{(F.7)}$$

This concludes the proof. ∎

# G   Application to PWA Dynamical Systems

In this section, we apply our main results to prediction in PWA systems with user-provided controls.

## G.1   Regret for One-Step Prediction in PWA Systems

A direct application of our main result is online prediction in piecewise affine systems. Consider the following dynamical system with state $\mathbf{z}_t \in \mathbb{R}^{d_z}$ and input $\mathbf{u}_t \in \mathbb{R}^{d_u}$:

$$\mathbf{z}_{t+1} = \mathbf{A}_{i_t}^\star \mathbf{z}_t + \mathbf{B}_{i_t}^\star \mathbf{u}_t + \mathbf{m}_{i_t}^\star + \mathbf{e}_t, \quad i_t = g^\star(\mathbf{z}_t, \mathbf{u}_t). \qquad \text{(G.1)}$$

Substitute $\Theta_i^\star := [\mathbf{A}_i \mid \mathbf{B}_i \mid \mathbf{m}_i]$ and define the concatenations $\mathbf{x}_t = [\mathbf{z}_t \mid \mathbf{u}_t]$ and $\bar{\mathbf{x}}_t = [\mathbf{x}_t \mid 1]$. The following lemma, proven in Appendix G.2, gives sufficient conditions on the system noise $\mathbf{e}_t$ and structure of the control inputs $\mathbf{u}_t$ under which $\mathbf{x}_t$ is directionally smooth.

**Lemma G.1.** *Let $\mathcal{F}_t$ denote the filtration generated by $\mathbf{x}_t = [\mathbf{z}_t \lfloor \mathbf{u}_t]$. Suppose that $\mathbf{e}_{t-1} \mid \mathcal{F}_{t-1}$ is $\sigma_{\text{dir}}$ smooth, and that in addition, $\mathbf{u}_t = \bar{\mathbf{K}}_t \mathbf{x}_t + \bar{\mathbf{u}}_t + \bar{\mathbf{e}}_t$, where $\bar{\mathbf{K}}_t$ and $\bar{\mathbf{u}}_t$ are $\mathcal{F}_{t-1}$-measurable[4], and $\bar{\mathbf{e}}_t \mid \mathcal{F}_{t-1}, \mathbf{e}_t$ is $\sigma_{\text{dir}}$-directionally smooth. Then, $\mathbf{x}_t \mid \mathcal{F}_{t-1}$ is $\sigma_{\text{dir}}/\sqrt{(1 + \|\mathbf{K}_t\|_{\text{op}})^2 + 1}$-directionally smooth.*

Directionally smooth noise distributions, such as Gaussians, are common in the study of online control [15, 56], and the smoothing condition on the input can be achieved by adding fractionally small noise, as is common in many reinforcement learning domains, such as to compute gradients in policy learning [60] or for Model Predictive Path Integral (MPPI) Control [65].

Throughout, we keep the notation for compactly representing our parameters by letting $\Theta_i^\star = [\mathbf{A}_i^\star \mid \mathbf{B}_i^\star \mid \mathbf{m}_i^\star]$, the estimate at time $t$ be $\widehat{\Theta}_{t,i} = [\hat{\mathbf{A}}_{t,i} \mid \hat{\mathbf{B}}_{t,i} \mid \hat{\mathbf{m}}_{t,i}]$, and covariates $\mathbf{x}_t = (\mathbf{z}_t, \mathbf{u}_t)$. We let $\mathscr{F}_t$ denote the filtration generated by $\mathbf{x}_{1:t}$, and note that $\mathbf{e}_t$ is $\mathscr{F}_{t+1}$-measurable.

**Assumption 7** (Boundedness). *The covariates and parameters, as defined above, satisfy Assumption 4.*

**Assumption 8** (SubGaussianity and Smoothness so as to satisfy Lemma G.1). *We assume that $\mathbf{e}_t \mid \mathscr{F}_t$ is $\sigma_{\text{dir}}$-directionally smooth and $\mathbf{u}_t = \bar{\mathbf{K}}_t \mathbf{x}_t + \bar{\mathbf{u}}_t + \bar{\mathbf{e}}_t$, where $\bar{\mathbf{K}}_t$ and $\bar{\mathbf{u}}_t$ are $\mathcal{F}_{t-1}$-measurable and $\bar{\mathbf{e}}_t \mid \mathcal{F}_{t-1}, \mathbf{e}_t$ is $\sigma_{\text{dir}}$-directionally smooth. Further, we assume that $\mathbf{e}_t \mid \mathscr{F}_t$ is $\nu^2$-subGaussian.*

---

[4]This permits, for example, that $\bar{\mathbf{K}}_t$ is chosen based on the previous mode $i_{t-1}$, or any estimate thereof that does not use $\mathbf{x}_t$.

Under these assumptions, we can apply our main result, Theorem 3, to bound the one-step prediction error in PWA systems. In particular, we have the following:

**Theorem 16** (One-Step Regret in PWA Systems)**.** *Suppose that* $\mathbf{z}_t, \mathbf{u}_t$ *evolve as in* (G.1) *with the attendant notation defined therein. Suppose further that Assumptions 5, 7, and 8 hold and that at each time $t$, the learner predicts $\widehat{\mathbf{z}}_{t+1}$ with the aim of minimizing the cumulative square loss with respect to the correct $\mathbf{z}_{t+1}$. If the learner applies Algorithm 1 to this setting, then with probability at least $1 - \delta$, the learner experiences regret at most $\sum_{t=0}^{T-1} \|\mathbf{z}_{t+1} - \widehat{\mathbf{z}}_{t+1}\|^2 \leq T\nu^2 +$ $\mathsf{poly}\left(\overline{\mathsf{par}}, \max_{1 \leq t \leq T} \|\mathbf{K}_t\|_{\mathrm{op}}, \log(1/\delta)\right) \cdot T^{1-\Omega(1)}$ where the exact polynomial dependence is given in Theorem 15.*

*Proof.* The result follows from applying Lemma G.1 to demonstrate directional smoothness and using Assumptions 8 and 7 and smoothness to apply Theorem 3. ∎

## G.2 Proof of Lemma G.1

We observe that this result follows directly from Lemma C.5. Indeed, note that by assumption, $\mathbf{z}_t | \mathscr{F}_{t-1}, \mathbf{u}_t$ is $\sigma_{\mathrm{dir}}$-directionally smooth by the assumption that $\mathbf{e}_{t-1} | \mathscr{F}_{t-1}$ is thus. Similarly, $\mathbf{u}_t | \mathscr{F}_{t-1}, \mathbf{z}_t$ is $\sigma_{\mathrm{dir}}$-directionally smooth by the assumption that $\bar{\mathbf{e}}_t | \mathscr{F}_{t-1}, \mathbf{e}_t$ is smooth. Therefore, the conclusion follows from Lemma C.5.

## G.3 Formal Simulation Regret Setup

In this setting, let $\bar{\mathscr{F}}_t$ denote the filtration generated by $(\mathbf{x}_{s,h})_{1 \leq s \leq t, 1 \leq h \leq H}$. Let $\mathscr{F}_{t,h}$ denote the filtration generated by $\bar{\mathscr{F}}_{t-1}$ and $(\mathbf{x}_{t,h'})_{1 \leq h' \leq h}$, with the convention $\mathscr{F}_{t,0} = \bar{\mathscr{F}}_{t-1}$. We require that our estimates $\widehat{\mathbf{\Theta}}_{t,i}$ be $\bar{\mathscr{F}}_{t-1}$-measurable. Further, we assume that our planner returns open-loop stochastic policies.

**Assumption 9** (Smoothed, Open Loop Stochastic Policies)**.** *We assume that our policy $\pi_t$ is stochastic and does not depend on state. That is, we assume that for all all $h \in [H]$ and $t \in [T]$. $\mathbf{u}_{t,h}, \ldots, \mathbf{u}_{t,H} \perp \mathbf{x}_{t,1:h} \mid \mathscr{F}_{t-1}$. We further assume that $\mathbf{u}_{t,h} \mid \mathbf{u}_{t,1:h-1}, \mathscr{F}_{t-1}$ is $\sigma_{\mathrm{dir}}$-smooth.*

**Assumption 10** (Noise Distribution)**.** *We assume that $\|\mathbf{e}_{t,h}\| \leq B_{\mathrm{noise}}$ for all $t, h$. Further, we assume that there exists a zero-mean, $\sigma_{\mathrm{dir}}$-directionally smooth distribution $\mathcal{D}$ over $\mathbb{R}^{d_z}$ such that $\mathbf{e}_{t,h} \mid \mathscr{F}_{t,h-1} \sim \mathcal{D}$.[5] Furthermore, we also assume that $\mathbf{z}_{t,1} \mid \bar{\mathscr{F}}_{t-1}$ is $\sigma_{\mathrm{dir}}$-smooth.*

**Definition 17** (Wasserstein Distance)**.** *Let $(\mathcal{X}, \mathsf{d})$ be a Polish space with metric $\mathsf{d}(\cdot, \cdot)$, and let $\mu, \nu$ be two Borel measures on $\mathcal{X}$. Define the Wasserstein distance as*

$$\mathcal{W}_2^2(\mu, \nu) = \inf_{\Gamma} \mathbb{E}_{(X,Y) \sim \Gamma} \left[\mathsf{d}(X,Y)^2\right]$$

*where the infimum is over all couplings $\Gamma$ such that the marginal of $X$ is distributed as $\mu$ and the marginal over $Y$ is distributed as $\nu$.*

We will show that with minor modifications, outlined below, Algorithm 1 can be leveraged to produce $H$-step simulated predictions with laws that are close in Wasserstein distance. In particular, this allows us to control the error of the associated predictions in expected squared norm.

## G.4 Algorithm Modifications

In order to produce good simulation regret, we run a variant of Algorithm 1 with two changes. First, we need to use Assumption 6 in some way to ensure that our estimated dynamics are contractive. Second, we need to iterate our predicitions in order to generate a trajectory of length $H$ for each time $t$. To address the first problem, suppose that we are in Line 3 of Algorithm 1, which applies ERMORACLE to the past data, resulting in $\left\{(\widehat{\mathbf{\Theta}}_{\tau,i})_{1 \leq i \leq K}, \widehat{g}_\tau\right\}$. For each $i$, we modify $\widehat{\mathbf{\Theta}}_{\tau,i}$ to become $\widetilde{\mathbf{\Theta}}_{\tau,i}$ by projecting $\widehat{\mathbf{\Theta}}_{\tau,i}$ onto the convex set

$$\mathcal{C}_{\mathbf{P}} = \left\{\mathbf{\Theta} | \mathbf{\Theta}^\top \mathbf{P} \mathbf{\Theta} \preceq \mathbf{P}\right\}.$$

---

[5]Results can be extended to $\mathcal{D}_{t,h}$ changing with the time step as long as these distributions are known to the learner.

---
**Algorithm 4** Algorithm for achieving low Simulation Regret
---
1: **Initialize** Epoch length $E$, Classifiers $\mathbf{w}^0 = (\mathbf{0}, \ldots, \mathbf{0})$, margin parameter $\gamma > 0$, learning rate $\eta > 0$, noise distributions $\mathcal{D}$
2: **for** $\tau = 1, 2, \ldots, T/E$ **do**
3: $\quad \left\{ (\widehat{\boldsymbol{\Theta}}_{\tau,i})_{1 \le i \le K}, \widehat{g}_\tau \right\} \quad \leftarrow \quad \text{ERMORACLE}((\bar{\mathbf{x}}_{1:\tau E}, \mathbf{y}_{1:\tau E}))$        (% self.modelUpdate)
4: $\quad\quad \widetilde{\boldsymbol{\Theta}}_{\tau_i} \leftarrow \text{Project}_{\mathcal{C}_{\mathbf{P}}}(\widehat{\boldsymbol{\Theta}}_{\tau_i})$            (% See (G.2))
5: $\quad\quad \widehat{g}_\tau \leftarrow \text{Reorder}\left( \widehat{g}_\tau, \left(\widetilde{\boldsymbol{\Theta}}_{\tau,i}\right)_{1 \le i \le K}, \left(\widetilde{\boldsymbol{\Theta}}_{\tau-1,i}\right)_{1 \le i \le K} \right)$     (% see Algorithm 3)
6: $\quad\quad \widetilde{g}_\tau \leftarrow \text{OGD}(\bar{\mathbf{x}}_{(\tau-1)E:\tau E}, \widehat{g}_\tau, \gamma, \eta)$       (% see Algorithm 2)
7: $\quad\quad$ **for** $t = \tau E, \ldots, (\tau + 1)E - 1$ **do**
8: $\quad\quad\quad$ **Receive** $\bar{\mathbf{x}}_t$ and set $\widehat{\mathbf{x}}_{t,0} = \bar{\mathbf{x}}_t$
9: $\quad\quad\quad$ **for** $h = 1, 2, \ldots, H$ **do**
10: $\quad\quad\quad\quad$ **Sample** $\mathbf{e}'_{t,h} \sim \mathcal{D}$
11: $\quad\quad\quad\quad$ **Predict** $\widehat{\mathbf{x}}_{t,h} = \widetilde{\boldsymbol{\Theta}}_{\tau, \widetilde{g}_\tau(\widehat{\mathbf{x}}_{t,h-1})} \cdot \widehat{\mathbf{x}}_{t,h-1} + \mathbf{e}'_{t,h}$
12: $\quad\quad\quad$ **Receive** $\mathbf{y}_t$
---

Formally, we define the projection to be

$$\text{Project}_{\mathcal{C}_{\mathbf{P}}}(\boldsymbol{\Theta}) = \underset{\boldsymbol{\Theta}' \in \mathcal{C}_{\mathbf{P}}}{\arg\min} \|\boldsymbol{\Theta} - \boldsymbol{\Theta}'\|_{\mathrm{F}}. \tag{G.2}$$

Because $\mathcal{C}_{\mathbf{P}}$ is convex, this step can be accomplished efficiently. As we shall observe in the sequel, this projection step never hurts our error guarantees due, again, to the convexity of $\mathcal{C}_{\mathbf{P}}$.

Moving on to the second modification, at each epoch $\tau$, we have a fixed set of parameters $\left\{ (\widehat{\boldsymbol{\Theta}}_{\tau,i})_{1 \le i \le K}, \widehat{g}_\tau \right\}$ where the $\widetilde{g}_\tau$ are modified from $\widehat{g}_\tau$ exactly as in Algorithm 1. Thus, at each time, we independently sample $\mathbf{e}'_{t,h} \sim \mathcal{D}$, the noise distribution, for $1 \le h \le H$ and predict $\widehat{\mathbf{x}}_{t,h} = \widetilde{\boldsymbol{\Theta}}_{\tau, \widetilde{g}_\tau(\widehat{\mathbf{x}}_{t,h-1})} \cdot \widehat{\mathbf{x}}_{t,h-1} + \mathbf{e}'_{t,h}$. The entire modified algorithm is given in Algorithm 4 and we prove in the sequel that we experience low simulation regret.

### G.5 Formal Guarantees for Simulation Regret

In this section, we prove that Algorithm 4 attains low simulation regret under our stated assumptions. Noting that Algorithm 4 is oracle-efficient, this result provides the first efficient algorithm that provably attains low simulation regret in the PWA setting. The main result is as follows:

**Theorem 18** (Simulation Regret Bound). *Suppose that we are in the setting of (3.1) and that Assumptions 5, 7, 8, 6, 9, and 10 hold. Suppose that we run Algorithm 4. Then, with probability at least $1 - \delta$, it holds that*

$$\text{SimReg}_{T,H} \le \frac{9B^2 H^8}{\sigma_{\mathrm{dir}}^2} \cdot \left( CB^{10}R^2 md\nu^2 K^4 \sigma_{\mathrm{dir}}^{-4} \Delta_{\mathrm{sep}}^{-1} T^{\frac{35}{36}} \left( \varepsilon_{\mathrm{orac}} + \log(TRBmdK) \right) + CBRK^2 T \varepsilon_{\mathrm{crp}} \right).$$

Before we prove Theorem 18, we must state the following key lemma, which says that if both the true and estimated dynamics are stable, and we are able to predict $\mathbf{x}_{t+1}$ with high accuracy, then our simulation regret $H$ steps into the future is also small.

**Lemma G.2.** *Let $\mathcal{G} : \mathbb{R}^d \to [K]$ denote the class of multi-class affine classifiers and let $\widehat{f}, f^\star$ denote functions satisfying the following properties:*

1. *The function $f^\star$ can be decomposed as $f^\star(\mathbf{x}) = f^\star_{g_\star(\mathbf{x})}(\mathbf{x})$ and similarly for $\widehat{f}$ it holds that $\widehat{f}(\mathbf{x}) = \widehat{f}_{\widehat{g}(\mathbf{x})}(\mathbf{x})$ for some $g_\star, \widehat{g} \in \mathcal{G}$.*

2. *For all $i \in [K]$, it holds that both $\widehat{f}_i$ and $f^\star_i$ are contractions, i.e., $\|f^\star_i(\mathbf{x}) - f^\star_i(\mathbf{x}')\| \le \|\mathbf{x} - \mathbf{x}'\|$.*

*Suppose that for* $0 \le h \le H - 1$, *we let*

$$\mathbf{x}_{h+1} = f^\star(\mathbf{x}_h) + \mathbf{e}_h \qquad\qquad \widehat{\mathbf{x}}_{h+1} = \widehat{f}(\widehat{\mathbf{x}}_h) + \mathbf{e}'_h$$

*for* $\mathbf{e}_h, \mathbf{e}'_h$ *identically distributed satisfying Assumption 10. Suppose that* $\mathbf{x}_0$ *is* $\sigma_{\mathrm{dir}}$*-smooth and that, almost surely,* $\mathbf{x}_h, \widehat{\mathbf{x}}_h$ *have norms bounded uniformly by* $B$. *If* $\max_{1 \le h < H} \left\| \widehat{f}(\mathbf{x}_h) - f^\star(\mathbf{x}_h) \right\| \le \varepsilon$ *then it holds that*

$$\mathcal{W}_2^2\left(\mathbf{x}_H, \widehat{\mathbf{x}}_H\right) \le \frac{9 B^2 H^6}{\sigma_{\mathrm{dir}}^2} \varepsilon^2 \varepsilon.$$

*Proof.* We begin by introducing intermediate random variables $\mathbf{x}_H^{(\ell)}$ for $1 \le \ell \le H$. We let $\mathbf{x}_h^{(\ell)} = \mathbf{x}_h$ for $h \le \ell$ and let

$$\mathbf{x}_{h+1}^{(\ell)} = \widehat{f}\left(\mathbf{x}_h^{(\ell)}\right) + \mathbf{e}'_t$$

otherwise. Note that $\mathbf{x}_H^{(1)} = \widehat{\mathbf{x}}_H$ and that $\mathbf{x}_H^{(H)} = \mathbf{x}_H$. By the triangle inequality applied to Wasserstein distance, we observe that

$$\mathcal{W}_2(\mathbf{x}_H, \widehat{\mathbf{x}}_H) \le \sum_{\ell=1}^{H} \mathcal{W}_2\left(\mathbf{x}_H^{(\ell)}, \mathbf{x}_H^{(\ell+1)}\right).$$

Consider a coupling where $\mathbf{e}_h = \mathbf{e}'_h$ for all $h$. Note that under this coupling, for any $h > \ell$, it holds that

$$\left\| \mathbf{x}_h^{(\ell)} - \mathbf{x}_h^{(\ell+1)} \right\| = \left\| \widehat{f}(\mathbf{x}_{h-1}^{(\ell)}) + \mathbf{e}_t - \widehat{f}(\mathbf{x}_{h-1}^{(\ell+1)}) - \mathbf{e}'_t \right\|$$

$$= \left\| \widehat{f}(\mathbf{x}_{h-1}^{(\ell)}) - \widehat{f}(\mathbf{x}_{h-1}^{(\ell+1)}) \right\|$$

$$\le \left\| \mathbf{x}_{h-1}^{(\ell)} - \mathbf{x}_{h-1}^{(\ell+1)} \right\| + B \cdot \mathbb{I}\left[ \widehat{g}(\mathbf{x}_{h-1}^{(\ell)}) \ne \widehat{g}(\mathbf{x}_{h-1}^{(\ell+1)}) \right].$$

Thus, in particular,

$$\left\| \mathbf{x}_H^{(\ell)} - \mathbf{x}_H^{(\ell+1)} \right\| \le \left\| \mathbf{x}_{\ell+1}^{(\ell)} - \mathbf{x}_{\ell+1}^{(\ell+1)} \right\| + B \sum_{h=\ell}^{H-1} \mathbb{I}\left[ \widehat{g}(\mathbf{x}_{h-1}^{(\ell)}) \ne \widehat{g}(\mathbf{x}_{h-1}^{(\ell+1)}) \right]$$

$$\le \left\| \widehat{f}(\mathbf{x}_\ell) - f^\star(\mathbf{x}_\ell) \right\| + B \sum_{h=\ell}^{H-1} \mathbb{I}\left[ \widehat{g}(\mathbf{x}_{h-1}^{(\ell)}) \ne \widehat{g}(\mathbf{x}_{h-1}^{(\ell+1)}) \right]$$

$$\le \varepsilon + B \sum_{h=\ell}^{H-1} \mathbb{I}\left[ \widehat{g}(\mathbf{x}_{h-1}^{(\ell)}) \ne \widehat{g}(\mathbf{x}_{h-1}^{(\ell+1)}) \right]$$

by the assumption that $\left\| \widehat{f}(\mathbf{x}_h) - f^\star(\mathbf{x}_h) \right\| \le \varepsilon$. We now note that a union bound implies:

$$\sum_{h=\ell}^{H-1} \mathbb{I}\left[ \widehat{g}(\mathbf{x}_{h-1}^{(\ell)}) \ne \widehat{g}(\mathbf{x}_{h-1}^{(\ell+1)}) \right] = \sum_{h=\ell}^{H-1} \mathbb{I}\left[ \widehat{g}(\mathbf{x}_{h-1}^{(\ell)}) \ne \widehat{g}(\mathbf{x}_{h-1}^{(\ell+1)}) \text{ and } \widehat{g}(\mathbf{x}_{h'}^{(\ell)}) = g_\star(\mathbf{x}_{h'}^{(\ell+1)}) \text{ for } h' \le h \right]$$

$$+ \sum_{h=\ell}^{H-1} \mathbb{I}\left[ \widehat{g}(\mathbf{x}_{h-1}^{(\ell)}) \ne \widehat{g}(\mathbf{x}_{h-1}^{(\ell+1)}) \text{ and there exists } h' \le h \text{ such that } \widehat{g}(\mathbf{x}_{h'}^{(\ell)}) \ne g_\star(\mathbf{x}_{h'}^{(\ell+1)}) \right]$$

$$\le \sum_{h=\ell}^{H-1} \mathbb{I}\left[ \widehat{g}(\mathbf{x}_{h-1}^{(\ell)}) \ne \widehat{g}(\mathbf{x}_{h-1}^{(\ell+1)}) \text{ and } \widehat{g}(\mathbf{x}_{h'}^{(\ell)}) = g_\star(\mathbf{x}_{h'}^{(\ell+1)}) \text{ for } h' \le h \right]$$

$$+ \sum_{h=\ell}^{H-1} (H-1-\ell) \mathbb{I}\left[ \widehat{g}(\mathbf{x}_{h-1}^{(\ell)}) \ne \widehat{g}(\mathbf{x}_{h-1}^{(\ell+1)}) \text{ and } \widehat{g}(\mathbf{x}_{h'}^{(\ell)}) = g_\star(\mathbf{x}_{h'}^{(\ell+1)}) \text{ for } h' \le h \right]$$

$$\le H \sum_{h=\ell}^{H-1} \mathbb{I}\left[ \widehat{g}(\mathbf{x}_{h-1}^{(\ell)}) \ne \widehat{g}(\mathbf{x}_{h-1}^{(\ell+1)}) \text{ and } \widehat{g}(\mathbf{x}_{h'}^{(\ell)}) = g_\star(\mathbf{x}_{h'}^{(\ell+1)}) \text{ for } h' \le h \right].$$

We now observe that

$$\mathbb{I}\left[\widehat{g}(\mathbf{x}_{h-1}^{(\ell)}) \neq \widehat{g}(\mathbf{x}_{h-1}^{(\ell+1)}) \text{ and } \widehat{g}(\mathbf{x}_{h'}^{(\ell)}) = g_\star(\mathbf{x}_{h'}^{(\ell+1)}) \text{ for } h' \leq h\right] \leq \mathbb{I}\left[\widehat{g}(\mathbf{x}_{h-1}^{(\ell)}) \neq \widehat{g}(\mathbf{x}_{h-1}^{(\ell+1)}) \text{ and } \left\|\mathbf{x}_{h-1}^{(\ell)} - \mathbf{x}_{h-1}^{(\ell+1)}\right\| \leq \varepsilon\right]$$

By the fact that $\mathcal{G}$ is given by multi-class affine thresholds, we see that

$$\mathbb{I}\left[\widehat{g}(\mathbf{x}_{h-1}^{(\ell)}) \neq \widehat{g}(\mathbf{x}_{h-1}^{(\ell+1)}) \text{ and } \left\|\mathbf{x}_{h-1}^{(\ell)} - \mathbf{x}_{h-1}^{(\ell+1)}\right\| \leq \varepsilon\right] \leq \sum_{1 \leq i,j \leq K} \mathbb{I}\left[|\langle \mathbf{w}_{ij}, \mathbf{x}_{h-1}\rangle| \leq 2\varepsilon\right]$$

where the $\mathbf{w}_{ij}$ are the unit vectors determining the decision boundaries induced by $\widehat{g}$. By smoothness, then, we see that

$$\mathcal{W}(\mathbf{x}_H^{(\ell)}, \mathbf{x}_H^{(\ell+1)}) \leq \varepsilon + BH \cdot \mathbb{P}\left(\sum_{h=\ell}^{H-1} \mathbb{I}\left[\widehat{g}(\mathbf{x}_{h-1}^{(\ell)}) \neq \widehat{g}(\mathbf{x}_{h-1}^{(\ell+1)}) \text{ and } \widehat{g}(\mathbf{x}_{h'}^{(\ell)}) = g_\star(\mathbf{x}_{h'}^{(\ell+1)}) \text{ for } h' \leq h\right] > 1\right)$$

$$\leq \varepsilon + BH \sum_{h=1}^{H} \mathbb{P}\left(\widehat{g}(\mathbf{x}_{h-1}^{(\ell)}) \neq \widehat{g}(\mathbf{x}_{h-1}^{(\ell+1)}) \text{ and } \widehat{g}(\mathbf{x}_{h'}^{(\ell)}) = g_\star(\mathbf{x}_{h'}^{(\ell+1)}) \text{ for } h' \leq h\right)$$

$$\leq \varepsilon + BH \sum_{h=1}^{H} \sum_{1 \leq i,j \leq K} \mathbb{P}\left(|\langle \mathbf{w}_{ij}, \mathbf{x}_h\rangle| \leq 2\varepsilon\right)$$

$$\leq \left(1 + \frac{2BH^2}{\sigma_{\text{dir}}}\right)\varepsilon.$$

Plugging this into our bound at the beginning and noting that $1 \leq \frac{BH^2}{\sigma_{\text{dir}}}$ concludes the proof. ∎

We are now ready to prove Theorem 18. In particular, we use Lemma G.2 to reduce a bound on simulation regret to one of the distance between $\widehat{f}$ and $f^\star$ evaluated on the data sequence. We then apply Theorems 11 and 13 to control this distance. The details are below:

*Proof of Theorem 18.* We begin by applying Lemma G.2. Observe that in our setting, $\mathcal{G}$ is the class of multi-class affine classifiers and $f^\star$, $\widehat{f}_t$ can be decomposed as required by the theorem, where we let $\tau$ denote the epoch in which $t$ is placed and let

$$\widehat{f}_t(\mathbf{x}) = \widetilde{\boldsymbol{\Theta}}_{\tau, \widetilde{g}_\tau(\mathbf{x})} \cdot \mathbf{x}.$$

By the fact that $\widetilde{\boldsymbol{\Theta}}_{\tau,i}, \boldsymbol{\Theta}_i^\star \in \mathcal{C}_{\mathbf{P}}$, which follows from the construction of $\widetilde{\boldsymbol{\Theta}}_{\tau,i}$ and Assumption 6, the contractivity assumption required by Lemma G.2 holds. Furthermore, note that because $\mathcal{C}_{\mathbf{P}}$ is convex and $\boldsymbol{\Theta}_i^\star \in \mathcal{C}_{\mathbf{P}}$, it holds that

$$\|\widetilde{\boldsymbol{\Theta}}_{\tau,i} - \boldsymbol{\Theta}_{\pi_\tau(i)}^\star\|_{\text{F}} \leq \|\widehat{\boldsymbol{\Theta}}_{\tau,i} - \boldsymbol{\Theta}_{\pi_\tau(i)}^\star\|_{\text{F}}. \tag{G.3}$$

Now, note that Algorithm 1 does not update the predicting functions within the epoch and thus for all $t$ such that $\{t, t+1, \ldots, t+H\}$ does not contain an integral multiple of $E$, it holds that our prediction function $\widehat{f}_t$ is constant on the interval. Let $\mathcal{I}$ denote the set of times $t$ such that the previous condition fails, i.e.,

$$\mathcal{I} = \{t \leq T \mid \tau E - t > H \text{ for all } \tau \in \mathbb{N}\}.$$

Then, we may apply Lemma G.2 to get

$$
\begin{aligned}
\mathrm{SimReg}_{T,H} &= \sum_{t=1}^{T}\sum_{h=1}^{H}\mathcal{W}_2^2(\mathbf{x}_{t+h},\widehat{\mathbf{x}}_{t+h}|\mathscr{F}_{t-1}) \\
&= \sum_{t=1}^{T}\mathbb{I}\left[t\in\mathcal{I}\right]\sum_{h=1}^{H}\mathcal{W}_2^2(\mathbf{x}_{t+h},\widehat{\mathbf{x}}_{t+h}|\mathscr{F}_{t-1}) + \sum_{t=1}^{T}\mathbb{I}\left[t\notin\mathcal{I}\right]\sum_{h=1}^{H}\mathcal{W}_2^2(\mathbf{x}_{t+h},\widehat{\mathbf{x}}_{t+h}|\mathscr{F}_{t-1}) \\
&\leq \sum_{t=1}^{T}\mathbb{I}\left[t\in\mathcal{I}\right]\sum_{h=1}^{H}\frac{9B^2H^6}{\sigma_{\mathrm{dir}}^2}\max_{1\leq h\leq H}\left\|\widehat{f}(\mathbf{x}_{t+h})-f^{\star}(\mathbf{x}_{t+h})\right\|^2 + 4HB^2R^2\sum_{t=1}^{T}\mathbb{I}\left[t\notin\mathcal{I}\right] \\
&\leq \frac{9B^2H^7}{\sigma_{\mathrm{dir}}^2}\sum_{t=1}^{T}\mathbb{I}\left[t\in\mathcal{I}\right]\sum_{h=1}^{H}\left\|\widehat{f}(\mathbf{x}_{t+h})-f^{\star}(\mathbf{x}_{t+h})\right\|^2 + 4B^2R^2H\cdot\frac{HT}{E} \\
&\leq \frac{9B^2H^8}{\sigma_{\mathrm{dir}}^2}\sum_{t=1}^{T}\mathbb{I}\left[t\in\mathcal{I}\right]\left\|\widehat{f}(\mathbf{x}_t)-f^{\star}(\mathbf{x}_t)\right\|^2 + 4B^2R^2H^2\cdot\frac{T}{E} \\
&\leq \frac{9B^2H^8}{\sigma_{\mathrm{dir}}^2}\sum_{t=1}^{T}\|\widehat{\mathbf{y}}_t-f^{\star}(\mathbf{x}_t)\|^2 + 4B^2R^2H^2\cdot\frac{T}{E},
\end{aligned}
$$

where $\widehat{\mathbf{y}}_t$ is the prediction of Algorithm 1. We may now apply Theorem 15 to bound this last quantity by applying (G.3). The result follows. ∎

## G.6 Extensions

For simplicity, we considered a setting where all linear dynamics were stable with a common Lyapunov function. Our results can be extended to the case where there are mode-dependent gains $(\mathbf{K}_i)_{i\in K}$, where $\mathbf{K}_i\in\mathbb{R}^{d_z\times d_u}$, as well, because our proof demonstrates that as we achieve low-regret, our simulation accurately recovers the correct mode sequence. Thus, if we know gains which ensure mutual stability, we can apply these gains as well. More general closed-loop policies can also be accomodated, provided we maintain the requisite smoothness (as, for example, ensured by Lemma G.1). Lastly, as our regret-guarantees ensure parameter recovery, we can envision settings where the gains are constructed, for example, by certainty-equivalent control synthesis, and analyze their suboptimality via perturbation bounds such as those in [53].