# OpenReview forum: "Smoothed Online Learning for Prediction in Piecewise Affine Systems"
_NeurIPS.cc/2023/Conference — NeurIPS 2023 spotlight_

### Official Review · Reviewer_Zpmv · 2023-06-09

**Soundness:** 3 good
**Presentation:** 3 good
**Contribution:** 3 good
**Rating:** 7
**Confidence:** 4

**Summary:**

The authors consider a smoothed nonlinear regression problem with time-series data in which the hypothesis class is decomposed into the product of (i)  affine classes across polyhedral components and (ii) an indicator/classifier "choosing" the component. They first show that parameter recovery for the affine component of the class is possible at a slow rate and then proceed to control a notion of simulation regret for the prediction error at a (marginally) sublinear rate. They establish a few more results on similar lines along the way.

**Strengths:**

Overall, this is an interesting theory paper contributing to the growing field of learning in dynamical/control systems and I recommend it to be accepted contingent on a few of my minor concerns being addressed (after which I will gladly raise my score to 7 or higher). It is relatively original in terms of formulation and the quality of writing is good. The technical details also appear sound to me, although I did not check them all beyond quick "sanity checks". Details:

* The directional smoothness concept taken from [7] and its use here are interesting. Overcoming the curse of dimensionality in learning nonlinear systems is a nontrivial matter and this is an interesting contribution applying this concept.

* The problem is well-chosen and relatively well-motivated in terms of control applications (MPC). The claim to being foundational is perhaps a bit over the top but otherwise I quite liked the formulation.

* The paper is generally quite well-written and it is easy to follow the main developments.



**Weaknesses:**

Overall, as I state above, I quite liked the paper and found the techniques/results to be interesting. However, it does feel a little unfinished for the reasons I outline below:

* The regret bound is quite weak (which is basically just $o(T)$). While I appreciate that this perhaps is more to be seen as a "proof of concept" I cannot help but wonder whether some more effort should have been put into proving a more palatable rate before submission.

Perhaps my main concern is the following:

* I also wonder whether the parameter recovery bound could have been improved, which as it stands, appears to be at a slow rate. I did not really appreciate this being passed of as a "parametric rate" (line 270). The authors prove in Theorem 4 that
$$error^2_i \lesssim \sqrt{T} / (\textnormal{number of visits to i}) \leq 1/\sqrt{\textnormal{number of visits to i}}$$
where I have omitted dimensional and complexity factors. Normally, the error in square norm for parameter recovery occurs at a fast rate (i.e. removing the square-root in the RHS above). If I squint I can also see how this could be obtained by controlling the $L^2$ prediction error of the hypothesis class as in [A] below. Heuristically I would expect by lower isomorphism in $L^2$:
$$ error^2_i (L^2) \times \textnormal{occupancy rate}_i \lesssim error^2(L^2) \lesssim  error^2(\textnormal{empirical }L^2) \lesssim \textnormal{self-normalized term}.$$
Re-arranging this should in principle give a fast rate for parameter recovery. I am sure the predictor $\hat g$ complicates matters, but the additional complexity should be controllable in terms of the number of regions. While it would be appreciated, let me also say that I do not expect the authors to  entirely "fix" this for the final submission but I think it at least deserves comment.

* While on the topic of [A], this reference should be added to the related work section on nonlinear system identification for (stochastically) stable systems. The ideas therein may also be useful for answering parts of the discussion in Section 5.

[A] Ziemann, Ingvar, and Stephen Tu. "Learning with little mixing." Advances in Neural Information Processing Systems, 2022.


**minor:**

* overflow on line 723-724.

*

**Questions:**

* Is assumption 4 really consistent with the use of the self-normalized inequality of de la Pena et al/Abbasi-Yadkori et al? It looks like this restricts the ERM to search over bounded regions in Euclidean space.

* relating to the above weakness about parametric rates: is Lemma D.13 really anywhere near sharp? Why is there a factor $\sqrt{T}$ on the RHS? Assuming realizability ($\delta=0$) I would expect this to be  $\tilde O(\textnormal{complexity hyp class})$. Could this perhaps be removed by using an offset style of analysis as in [B] (which is for iid data, but see also [A] above for the dynamic (martingale) case,
and see also your own reference [48])?

* Is your claim that you have provided an "efficient algorithm" (line 399) really motivated without further nuance? Certainly, no claim to statistical efficiency can be made based on your analysis. See also my comments about your rates above.

[B] Liang, Tengyuan, Alexander Rakhlin, and Karthik Sridharan. "Learning with square loss: Localization through offset rademacher complexity." Conference on Learning Theory. PMLR, 2015.

* How reasonable is your assumption 6? It is not obvious to me how to construct/verify this condition.


**Limitations:**

See above caveats about rates and assumptions.

---

> ### Author Rebuttal · Authors · 2023-08-04
>
> # Regret Bound
>
> Thank you for your review.  Regarding the regret bound, we agree that the resulting oracle complexity is likely suboptimal.  Note that if one is happy with computationally inefficient algorithms, then running exponential weights on an exponentially large cover will attain optimal regret scaling like the square root of the horizon, so the more challenging question is what the oracle complexity of achieving average regret bounded by some epsilon is.  The primary cause for the poor regret is in our parameter recovery proof, specifically coming from our lower bound on the empirical covariance matrix of data on each mode, described in lines 339-345.  While the analysis is already fairly sophisticated in the proof of this result, one may hope that tighter bounds are possible, which would then translate into a significantly better regret bound.  We emphasize that it is already somewhat surprising that an oracle complexity that is polynomial in all parameters is possible in light of the general lower bound of Hazan & Koren (2016) cited in the paper.  While we conjecture that an optimal algorithm should not argue directly through parameter recovery, an advantage of our analysis is that it exposes the various components and challenges in addressing our setting. Moreover, by ensuring parameter recovery, we obtain guarantees for simulation regret which may be of independent interest.
>
> # Fast vs Slow Rate
> Regarding the fast vs slow rates of the parameter recovery in Theorem 4, we will clarify this point better in the revision and remove the reference to a parametric rate.  Note that minor changes to our analysis do *not* yield fast rates in the manner that is being suggested precisely because of the dependence on the classifier.  The “slow” square root only comes in through Lemma D.15 with the core issue being that while we have control over the *total* excess risk summed over all pairs of modes (i,j), in order for our parameter recovery to hold, we need the excess risk on each *individual* pair to also be small.  Due to the stochasticity of the perturbations, some of these excess risks can be negative and the size of the fluctuations is on the order of the square root of the number of points for Gaussian perturbations.  Thus, fast rates *cannot* be obtained without modifying the definition of excess risk.  One might hope for some improvement by changing the definition of the excess risk and considering perturbations around the empirically optimal predictor on the data, as opposed to the ground truth, but this would add substantial technical difficulty to an already highly intricate proof; furthermore, it is not clear that this method would help because, again, we need to control the lower tail of excess risk on individual pairs of modes, whereas most fast rate analyses follow by careful control of the upper tail.  To summarize, because the fundamental difficulty is in translating a bound on the sum of many different terms (some of which can be negative) into a bound that holds uniformly over each term, we pay for the largest excess risk on a given term, which may be substantially larger than one might initially expect; thus, in order to improve this rate, we suspect that a substantially different analysis would be needed.
>
> # Questions
>
> 1. Note that we use the self-normalized bounds not because we are concerned that our numerators are too large, but rather because the denominators might be too small; indeed, their main utility is in easily allowing us to specialize bounds to apply only to time steps that fall into a given mode.  We suspect that a similar analysis without self-normalization but with more involved conditioning would also hold.  Meanwhile, the compactness assumption is necessary for us to bound the size of the disagreement cover.
> 2. See the above discussion on these bounds.  In particular, the slow rate comes only from Lemma D.15, with all other parts of the proof having the desired dependence on horizon.
> 3. What we mean by “efficient” is that all dependences are polynomial in all problem parameters, in contrast to something like exponential weights on a cover of the function class, whose computational complexity suffers an unavoidable exponential-in-dimension dependence.  This is also in contrast to naively applying the algorithms of Block et al (2022) and Haghtalab et al (2022) from the smoothed online learning community, which also suffer this exponential dependence.  We agree that the polynomials in our bounds are not small; we consider the mere fact that they are polynomial as opposed to exponential surprising in light of computational lower bounds in Hazan & Koren (2016), Block et al (2022), and Haghtalab (2022), which is where the word “efficient” is coming from.
> 4. Assumption 6 was the simplest (albeit strong) assumption that we needed to ensure simulation regret. Its purpose is to show that possible instability of the predicted system, rather than horizon per se, is the main obstacle for PWA simulation. In some sense, stability is necessary for multi-step prediction problems (as has been studied in the setting of imitation learning (see e.g. https://arxiv.org/abs/2205.14812). We could replace Assumption 6 with the weaker assumption that the Lypaunov matrix P* is unknown, and instead propose to synthesize this common Lyapunov function from data (certifying a Lyapunov equation is an SDP and hence can be certified for many matrices simultaneously). Because simulation regret is intended as an application of our result, we did not go down this route.

---

> > ### Comment · Reviewer_Zpmv · 2023-08-14
> >
> > Many thanks to the authors for their response. I believe that my main concern---fast vs slow rates---has been clarified to a satisfactory extent. I don't have any significant concerns remaining and am happy to raise my score to a 7 as promised.

---

### Official Review · Reviewer_MA9v · 2023-07-04

**Soundness:** 4 excellent
**Presentation:** 3 good
**Contribution:** 4 excellent
**Rating:** 8
**Confidence:** 3

**Summary:**

This paper develops the theory to show that the online smoothed learning of piecewise affine model (PWA), under mild assumption, can achieve sublinear regret in dynamics prediction, and also the regret bound is polynomial in system parameters.



**Strengths:**


I think the results of this paper are of great importance both to robotics community, where the online learning of PWA models can be used to represent non-smooth physical contact systems, and to online learning community, where some new results, e.g., the sublinear regret is polynomial instead of exponential in system dimension, is developed.

Most importantly, the main algorithm is easy to implement and seems really promising to apply to the real problem.


**Weaknesses:**


To further improve the paper, I have the following comments:
- How does Definition 1 of directionally smooth imply that  "directional smoothness ensures the system spends little time close to a boundary" ?
- It is not sure if Assumption 2 is mild, given there are jumps/discontinuities in PWA.
- line 216, what's the meaning of "$\Theta(\sigma_{dir}^{d})$-smoothness"?
- In Algorithm 1, the re-labeling step (Algorithm 3)  needs to check each pair of labels between the new and old partitions. How is the Algorithm scalable with large number of partitions?
- Line 312, "a sketch of the proof of Theorem 15", I didn't see any Theorem 15 in the main text.
- Typo in Equ. (3.1), $z_t$ should be $z_{t,h}$
- Typo in the text right below Euq. (3.1), $\hat{m}_{t,\hat{i}_{t,h}}\hat{e}_{t,h}$ should be $\hat{m}_{t,\hat{i}_{t,h}}+\hat{e}_{t,h}$
- Since the paper is partially motivated by the recent progress in random smoothing in physical contact systems, the authors may need to discuss the promise or limitation of the proposed algorithm when it is applied to that domain.

**Questions:**

Please see my questions in the above section.

**Limitations:**

Please see my questions in the above section.

---

> ### Author Rebuttal · Authors · 2023-08-04
>
> Thank you for your review.  We will address each question below:
> 1. Directional smoothness means that for any direction, the random variable projected onto this direction is anti-concentrated in any interval.  Because the boundaries are determined by affine sets, for the system to be near to the boundary, its projection onto the direction defining the boundary must lie in a small interval, which is unlikely to happen by directional smoothness.  We will clarify this in the revision.
> 2. Note that smoothness in Assumption 2 refers to the distribution in a directional sense, not a function and thus whether or not it holds is not determined by the discontinuities in the system.  Indeed, randomized smoothing added for the purposes of optimization already ensures that Assumption 2 holds.
> 3. By standard smoothness we mean the original notion that there exists a uniform bound on the density of some random variable, which is studied in the earlier works on smoothed analysis we cite in lines 145-165.  The distinction is included in order to emphasize why directional smoothness is so important as opposed to this previous notion.  We will add this formal definition in the revision.
> 4. This is a good question and we will add some more discussion of practical considerations in the revision.  Note that a natural way to scale the algorithm to a large number of modes would be to only keep track of modes already visited and update this set accordingly between epochs.  Unfortunately, scaling polynomially with the number of modes is unavoidable in general.
> 5. Thank you for catching this typo; we meant theorem 3.
> 6. Thank you for catching these typos.
> 7. We think that one of the most significant advantages of our algorithm is demonstrating that there are no fundamental learning-theoretic barriers to obtaining low-regret in smoothed piecewise affine prediction in an oracle efficient manner.

---

### Official Review · Reviewer_QtL6 · 2023-07-07

**Soundness:** 4 excellent
**Presentation:** 3 good
**Contribution:** 4 excellent
**Rating:** 8
**Confidence:** 3

**Summary:**

This paper presents an algorithm for learning policies for piecewise affine systems, with both parameter-accuracy and regret bound guarantees, by using a kind of randomized smoothing of the boundaries of the affine regions. It identifies an anticoncentration property ("directional smoothness") that was previously proposed, along with some more standard assumptions on the noise distribution and separation of parameters in the affine regions that enable efficient identification; the problem was known to be inherently hard in the absence of such assumptions.

**Strengths:**

The problem tackled in this work is natural and was known to be quite challenging. Although the basic idea of randomized smoothing to avoid the accumulation of error around the boundaries of affine regions is quite clear, actually executing this plan required quite a bit of technical sophistication. The paper itself gives a decent sketch of the approach and where these difficulties lay.

**Weaknesses:**

The only weakness is that the bounds, while meeting the requirements of being polynomial, sublinear, etc. are a bit extravagant. I don't know if this is a practical algorithm, and I hope this work isn't the last word on this problem.

**Questions:**

What avenues do you see for potentially improving the algorithm or its analysis? What would you conjecture the correct order of dependence on the time horizon to be?

**Limitations:**

I think this is fine. Although it isn't clear whether or not the algorithm is practical, the work is positioned as being a first work to meet basic theoretical desiderata, and the assumptions used seem to be well motivated.

---

> ### Author Rebuttal · Authors · 2023-08-04
>
> Thank you for your review.  Regarding the regret bound, we agree that the resulting oracle complexity is likely suboptimal.  Note that if one is happy with computationally inefficient algorithms, then running exponential weights on an exponentially large cover will attain optimal regret scaling like the square root of the horizon, so the more challenging question is what the oracle complexity of achieving average regret bounded by some epsilon is.  The primary cause for the poor regret is in our parameter recovery proof, specifically coming from our lower bound on the empirical covariance matrix of data on each mode, described in lines 339-345.  While the analysis is already fairly sophisticated in the proof of this result, one may hope that tighter bounds are possible, which would then translate into a significantly better regret bound.  We emphasize that it is already somewhat surprising that an oracle complexity that is polynomial in all parameters is possible in light of the general lower bound of Hazan & Koren (2016) cited in the paper.
>
> While we conjecture that an optimal algorithm should not argue directly through parameter recovery, an advantage of our analysis is that it exposes the various components and challenges in addressing our setting. Moreover, by ensuring parameter recovery, we obtain guarantees for simulation regret which may be of independent interest.

---

### Official Review · Reviewer_7mVW · 2023-07-11

**Soundness:** 2 fair
**Presentation:** 1 poor
**Contribution:** 2 fair
**Rating:** 4
**Confidence:** 2

**Summary:**

This paper studies the problem of prediction in piecewise affine systems over a finite horizon. The main results are a linear regret bound and a sublinear regret bound under the proposed metric of simulation regret. To obtain the regret bounds, the authors made many assumptions and assume the access to an empirical risk minimization oracle.

**Strengths:**

There is a good motivation to study piecewise-affine systems as an intermediate step between linear and nonlinear systems. This paper also presents many theoretical results.

**Weaknesses:**

It is hard for me to follow the paper from the problem setting of piece-wise affine regression. There are too many mathematical expressions without an appropriate amount of context and motivating examples. And many key definitions and explanations are deferred to the appendix.

The setting is confusing because some assumptions are not stated formally as assumptions. For example, in line 185, the authors assume $g_*$ to be an affine classifier, but they did not explain the implication or motivation of this assumption. It seems that taking argmax only contains a small subset of piece-wise affine systems. Besides, it is also unclear where the non-stochastic corruption $\delta_t$ comes from and if it is oblivious.

I also have some concern about whether the assumptions made to obtain the regret bound are too strong, especially for Assumption 1. I conjecture that the upper bound $\varepsilon_{orac}$ on the right hand side will increase with the horizon $s$ in practice if we want to solve this problem efficiently with unknown parameters. Assumption 6 also seems very strong and I feel it is almost equivalent to knowing the exact dynamics under each mode.


**Questions:**

Please see my comments in the previous section.

**Limitations:**

The authors discussed many future directions at the end of the paper, and I did not see any potential negative societal impact.

---

> ### Author Rebuttal · Authors · 2023-08-04
>
> Thank you for your feedback. We will be sure to provide more concrete examples and motivation for our setting in our revision. While we deferred several explanations to the appendix for the sake of space, we will better clarify the intuition for the mathematical expressions.
>
> We agree that there may be more settings for the boundaries that are not covered by the theory. For example, our bounds are poly in the the number of argmax regions $K$, whereas a classifier that, for example, parameterizes $K’$ hyperplanes can induce $2^{K’}$ regions. We will clarify this limitation in the revision; it is essentially for simplicity.   Specifically, the properties of the mode classifier we need are (1) it has a bounded disagreement cover and (2) the form of the classifier is amenable to online multi-class classification. We can envision several more general classifiers with this property, and will remark this when we revise.  It should be noted that one-vs-all classification is extremely common in classification and the problem is already very nontrivial in this case.
>
> Regarding the strength of the other assumptions, we note that Assumption 1 (i.e. that we have access to an optimization oracle over the function class) is the accepted standard in the online learning literature, as we discuss in the paragraph starting in line 145 in the related work.
> Moreover, one of the surprising implications of our result is that the general computational lower bounds of Hazan & Koren 2016, Block et al 2022, and Haghtalab et al 2022, cited in our paper, do not hold PWA functions in the smoothed online learning setting, paving the way for future analysis of computational tractability for function classes of practical interest.  While online learning is possible with stronger oracles (e.g. the FTPL oracle in Agarwal, Gonen, Hazan), we show that the standard ERM oracle suffices. Furthermore, we discuss in lines 57-64 the practicality of this oracle in the context under consideration; informally, this oracle should be viewed as “the best available heuristic”, which can be replaced with better algorithms as they are developed.
>
> Re:f Assumption 6, we acknowledge in line 298 that this is a strong assumption; this section is just a first application of our main results, emphasizing that the barrier to go from prediction regret to simulation regret is stability of the system (and the predictions), rather than long horizons per se.

---

### Decision · Program_Chairs · 2023-09-21

**Decision:**

Accept (spotlight)

**Comment:**

This work proposes an algorithm for prediction under PWA systems with sublinear regret that depends polynomially on system parameters. The general PWA setting is known to be difficult, and this work gives reasonable conditions under which good regret is achievable. This is a very interesting theoretical result, and even though it’s unclear if the bounds are optimal, this result is still significant for achieving vanishing regret and oracle efficiency. I recommend acceptance.